# Cysteine cross-linking in native membranes establishes the transmembrane architecture of Ire1

Kristina Väth[1,2]*, Carsten Mattes[1,2]*, John Reinhard[1,2]*, Roberto Covino[3], Heike Stumpf[1,2], Gerhard Hummer[4,5], and Robert Ernst[1,2]

**The ER is a key organelle of membrane biogenesis and crucial for the folding of both membrane and secretory proteins. Sensors of the unfolded protein response (UPR) monitor the unfolded protein load in the ER and convey effector functions for maintaining ER homeostasis. Aberrant compositions of the ER membrane, referred to as lipid bilayer stress, are equally potent activators of the UPR. How the distinct signals from lipid bilayer stress and unfolded proteins are processed by the conserved UPR transducer Ire1 remains unknown. Here, we have generated a functional, cysteine-less variant of Ire1 and performed systematic cysteine cross-linking experiments in native membranes to establish its transmembrane architecture in signaling-active clusters. We show that the transmembrane helices of two neighboring Ire1 molecules adopt an X-shaped configuration independent of the primary cause for ER stress. This suggests that different forms of stress converge in a common, signaling-active transmembrane architecture of Ire1.**

## Introduction

The ER marks the entry point to the secretory pathway for soluble and membrane proteins. Under adverse conditions, accumulation of unfolded proteins causes ER stress and initiates the unfolded protein response (UPR). The UPR is mediated by the inositol-requiring enzyme 1 (Ire1) in budding yeast, and by the troika of IRE1α, PKR-like ER kinase (PERK), and activating transcription factor 6 (ATF6) in vertebrates (Walter and Ron, 2011). Once activated, the UPR down-regulates the production of most proteins and initiates a wide transcriptional program to up-regulate ER chaperones, ER-associated degradation, and lipid biosynthesis (Travers et al., 2000). Through these mechanisms, the UPR is centrally involved in cell fate decisions between life, death, and differentiation (Hetz, 2012). Insulin-producing β-cells, for example, rely on UPR signals for their differentiation into professional secretory cells, while chronic ER stress caused by an excess of saturated fatty acids kills them (Fonseca et al., 2009). Consistent with its broad effector functions, the UPR is associated with numerous diseases including diabetes, cancer, and neurodegeneration (Kaufman, 2002).

Ire1 is highly conserved among eukaryotes and represents the only transducer of ER stress in budding yeast (Nikawa and Yamashita, 1992; Kimata and Kohno, 2011). It is a type I transmembrane protein equipped with an ER-luminal sensor domain and two cytosolic effector domains: a kinase and an RNase (Cox et al., 1993; Sidrauski and Walter, 1997; Mori et al., 1993). How exactly unfolded proteins activate the UPR via direct and indirect mechanisms is a matter of active debate (Karagöz et al., 2017; Gardner and Walter, 2011; Adams et al., 2019; Amin-Wetzel et al., 2017; Le and Kimata, 2021). ER stress caused by the accumulation of unfolded proteins leads to the oligomerization of Ire1 (Kimata et al., 2007), which activates the cytosolic effector kinase and RNase domains (Korennykh et al., 2009). The unconventional splicing of the *HAC1* precursor mRNA initiated by the RNase domain facilitates the production of an active transcription factor that controls a broad spectrum of genes with unfolded protein response elements in their promoter regions (Travers et al., 2000; Mori et al., 1992). A regulated *IRE1*-dependent decay of mRNA has been suggested as a parallel mechanism to reduce the folding load of the ER. However, regulated *IRE1*-dependent decay of mRNA does not seem to play the same important role in *Saccharomyces cerevisiae* as it does in *Saccharomyces pombe* or mammalian cells (Travers et al., 2000; Hollien and Weissman, 2006; Frost et al., 2012; Tam et al., 2014; Li et al., 2018).

Lipid bilayer stress due to aberrant compositions of the ER membrane is equally potent in activating the UPR (Promlek et al., 2011; Volmer et al., 2013; Surma et al., 2013). This membrane-based mechanism is conserved throughout evolution (Ho et al., 2018; Hou et al., 2014; Volmer et al., 2013) and has been associated with pathogenesis of type II diabetes and the lipotoxicity associated with obesity (Fonseca et al., 2009; Pineau and Ferreira, 2010). We have shown that Ire1 from baker's yeast

[1]Medical Biochemistry and Molecular Biology, Medical Faculty, Saarland University, Homburg, Germany; [2]Preclinical Center for Molecular Signaling, Medical Faculty, Saarland University, Homburg, Germany; [3]Frankfurt Institute of Advanced Sciences, Goethe-University, Frankfurt, Germany; [4]Department of Theoretical Biophysics, Max Planck Institute of Biophysics, Frankfurt, Germany; [5]Institute of Biophysics, Goethe-University, Frankfurt, Germany.

*K. Väth, C. Mattes, and J. Reinhard contributed equally to this paper; Correspondence to Robert Ernst: robert.ernst@uks.eu.

inserts an amphipathic helix (AH) into the luminal leaflet of the ER membrane, thereby forcing the short, adjacent transmembrane helix (TMH) to tilt, which locally squeezes the bilayer (Halbleib et al., 2017). Aberrant stiffening of the ER membrane during lipid bilayer stress increases the free energy penalty for membrane deformations, thereby stabilizing oligomeric assemblies of Ire1 via a membrane-based mechanism (Halbleib et al., 2017; Ernst et al., 2018). Even though it is well-established that proteotoxic and lipid bilayer stress leads to the formation of Ire1 clusters (Kimata et al., 2007; Halbleib et al., 2017; Li et al., 2010; Belyy et al., 2020), it remains unexplored if these forms of ER stress have a distinct impact on the architecture of Ire1 within these clusters. It has been speculated that different forms of ER stress might induce conformational changes in the transmembrane region, thereby allowing Ire1/IRE1α to mount custom-tailored adaptive programs (Hetz et al., 2020; Cho et al., 2019; Ho et al., 2020).

Here, we report on a systematic dissection of Ire1's TMH region in signaling-active clusters. We have engineered a cysteine-less variant for a genomic integration at the endogenous IRE1 locus and generated a series of constructs featuring single cysteines in the TMH region. This enabled us to develop a cross-linking approach and to study the transmembrane configuration of Ire1 in the natural environment of ER-derived membrane vesicles featuring a native complexity of lipids and proteins. This approach uncovers the overall transmembrane architecture of Ire1 and suggests an X-shaped configuration of the TMHs of neighboring Ire1 molecules. Our findings underscore the crucial importance of Ire1's highly bent configuration in the TMH region for stabilizing an oligomeric state via a membrane-mediated mechanism. Most importantly, we provide direct evidence that proteotoxic and lipid bilayer stress converge in common architecture of the TMH region in signaling-active Ire1.

## Results

We used systematic cysteine cross-linking in the TMH region of Ire1 to gain insight into the structural organization of signaling-active clusters during ER stress. Recognizing that Ire1 is activated by aberrant physicochemical membrane properties (Halbleib et al., 2017; Ernst et al., 2018), which are hard to mimic in vitro, we performed these experiments with microsomes exhibiting the natural complexity of ER proteins and lipids.

### Cysteine-less Ire1 is functional

We have generated a cysteine-less version of Ire1 that allows us to introduce single-cysteine residues in the TMH region for subsequent cross-linking using copper sulfate (CuSO$_4$). The cysteine-less construct is based on a previously established knock-in construct of IRE1 that provides homogeneous, near-endogenous expression (Halbleib et al., 2017) and encodes for a fully functional variant of Ire1 equipped with an 3xHA tag and a monomeric, yeast-enhanced GFP (yeGFP) inserted in a flexible loop at the position H875 (Fig. 1 A; van Anken et al., 2014; Halbleib et al., 2017). To generate a cysteine-less version, we substituted each of the 12 cysteines in the luminal,

transmembrane, and cytosolic domains with serine. Two cysteines in the signal sequence, which are cotranslationally removed, remained in the final construct to ensure correct ER targeting and membrane insertion (Fig. 1 A). Cysteine 48 of yeGFP (C48$^{yeGFP}$) was mutated to serine, while C70$^{yeGFP}$ is present in the cysteine-less construct to ensure correct folding of the fluorescent protein (Costantini et al., 2015). Notably, C70$^{yeGFP}$ is buried inside the GFP (Ormö et al., 1996) and thus inaccessible for cross-linking agents under nondenaturing conditions.

The steady-state levels of WT and cysteine-less Ire1 are comparable (Fig. S1 A). Cysteine-less Ire1 is properly integrated into the membrane as shown by subcellular fractionation (Fig. S1 B) and extraction assays (Fig. S1 C), thereby matching previous observations for WT Ire1 (Kimata et al., 2007; Halbleib et al., 2017). The functionality of cysteine-less Ire1 was analyzed using a sensitive assay scoring for the growth of cells exposed to inducers of ER stress (Halbleib et al., 2017). Liquid cultures in either minimal (synthetic complete dextrose [SCD]) or full (yeast peptone dextrose [YPD]) medium were exposed to different concentrations of the reducing agent DTT interfering with disulfide bridge formation in the ER. After 18 h of cultivation, the ODs of these cultures were determined. Cells producing either WT or cysteine-less Ire1 are phenotypically indistinguishable by this assay and substantially more resistant to DTT than cells lacking IRE1 (Fig. 1 B). This suggests that cysteine-less Ire1 is functional and capable of mounting an adaptive UPR.

The functionality of cysteine-less Ire1 was further validated by quantifying the mRNA levels of spliced HAC1 (Fig. 1 C) and the mRNA level of the UPR target gene PDI1 (Fig. S1 D) in both stressed and unstressed cells. We used either DTT or tunicamycin (TM), an inhibitor of N-linked glycosylation, to induce proteotoxic stress for 1 h and analyzed lysates from stressed and unstressed cells by RT-quantitative PCR (RT-qPCR). As expected, the level of the spliced HAC1 mRNA was several-fold higher in stressed versus unstressed cells, and this up-regulation is observed in both WT and cysteine-less Ire1-producing cells. (Fig. 1 C). Control experiments also validated a comparable degree of HAC1 mRNA splicing in WT or cysteine-less Ire1-producing cells stressed with DTT (Fig. S1 D). We also observed an up-regulation of the PDI1 mRNA in response to ER stress, albeit to slightly lower extent for the cysteine-less version compared with the WT construct (Fig. S1 E). Using confocal microscopy and by applying an automated pipeline to identify cells with and without fluorescent clusters, we show that both cysteine-less and WT Ire1 cluster under conditions of ER stress, but not in unstressed cells (Fig. 1 D). Notably, confocal microscopy can only identify large clusters of Ire1, while dimers and smaller assemblies escape our detection. Furthermore, the detection of Ire1 in unstressed cells is particularly challenging in our case, because our knock-in strategy aims to provide a close to endogenous level of IRE1 expression (Halbleib et al., 2017). This is important because even the mild degree of overexpression when using an endogenous promoter from a yeast centromere (CEN)-based plasmid (Karim et al., 2013) is likely to interfere with normal UPR function by favoring dimerization and oligomerization. Using our setup, we robustly detect GFP-positive clusters of Ire1 (Fig. 1 D) in stressed cells,

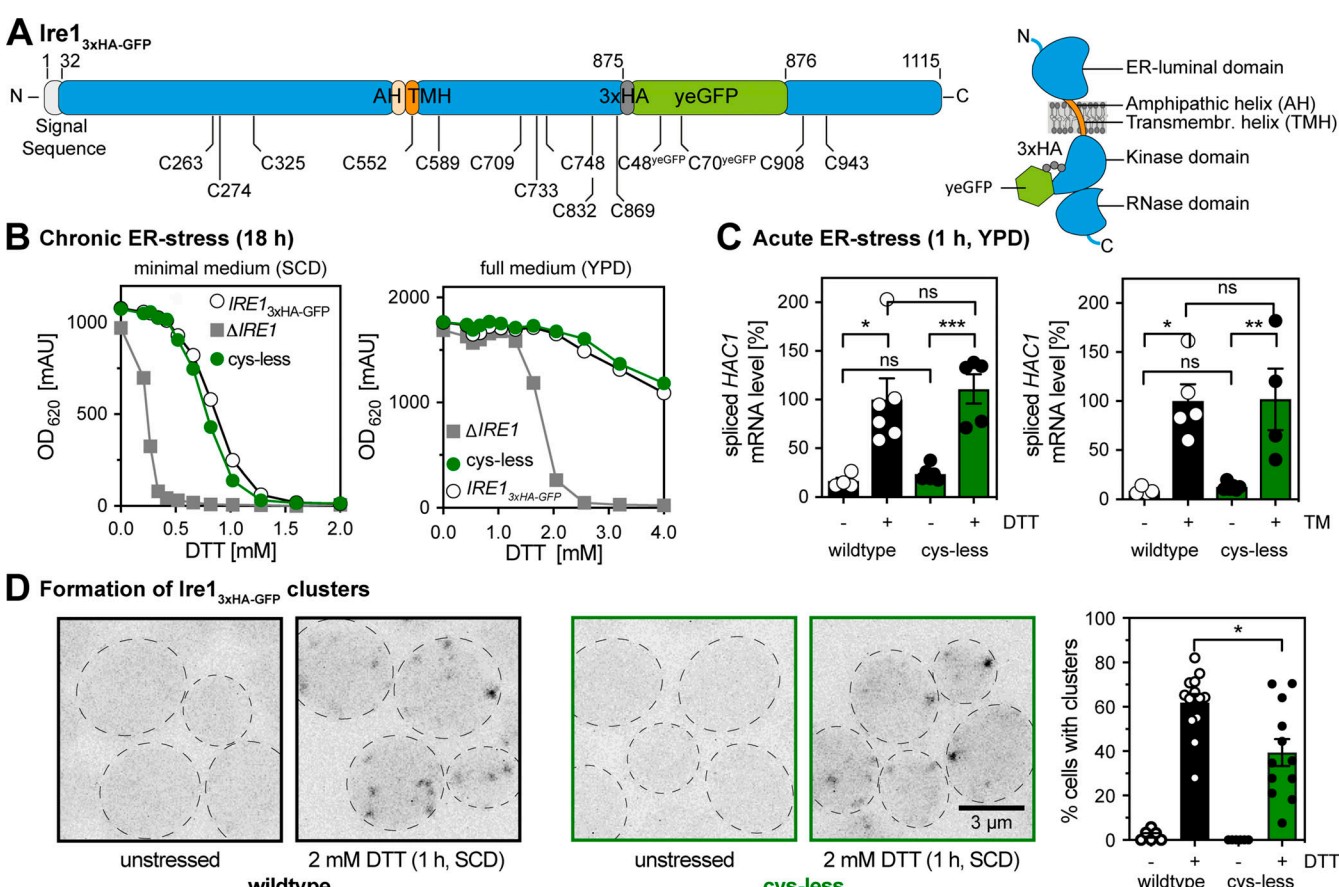

Figure 1. **Cysteine-less Ire1 expressed from its endogenous locus is functional. (A)** Schematic representations of the *IRE1*$_{3xHA-GFP}$ construct indicating the position of cysteine residues and topology. All 12 cysteines of Ire1 and C48$^{yeGFP}$ of yeGFP were substituted to serine to generate a cysteine-less variant. C70$^{yeGFP}$ remains in the final construct. Two cysteines in the signal sequence of Ire1 are removed upon ER translocation. **(B)** Resistance of the indicated strains to prolonged ER stress. Stationary overnight cultures of the indicated strains were used to inoculate a fresh culture in full or minimal media to an OD$_{600}$ of 0.2. After cultivation for 5–7 h at 30°C, the cells were diluted with prewarmed full or minimal media to an OD$_{600}$ of 0.01. Cells were cultivated for 18 h at 30°C in the indicated media and stressed with DTT. The density of the resulting culture was determined using the OD$_{620}$ or OD$_{600}$. Number of experiments in SCD (left): Δ*IRE1*, WT (n = 20), and cysteine-less (n = 12, each from four individual colonies). Number of experiments in YPD (right): Δ*IRE1* (n = 14 from three colonies), cysteine-less (n = 12 from four colonies), and WT (n = 9 from three colonies). **(C)** The relative level of the spliced *HAC1* mRNA was determined by RT-qPCR in unstressed and acutely stressed cells. Exponentially growing cells of the indicated strains were used to inoculate a fresh culture in YPD medium to an OD$_{600}$ of 0.2. After cultivation to an OD$_{600}$ of 0.7, the cells were stressed for 1 h with either 4 mM DTT (left) or 1.0 µg/ml TM (right). The data were normalized to the level of the spliced *HAC1* mRNA in DTT-stressed cells with the *IRE1*$_{3xHA-GFP}$ WT construct. Number of experiments (left): WT -DTT: n = 4; WT +DTT: n = 6; cysteine-less -DTT: n = 6; cysteine-less +DTT: n = 5. Number of experiments (right): WT -TM: n = 4; WT +TM: n = 5; cysteine-less -TM: n = 6; cysteine-less +TM: n = 4. **(D)** Cells were cultivated from OD$_{600}$ of 0.2 to OD$_{600}$ of 0.7 in SCD medium and then either left untreated or stressed with 2 mM DTT for 1 h. Live cells were mounted on agar slides, and z-stacks were recorded using confocal microscopy. Cells and clusters of Ire1 were automatically detected and quantified. All data are represented as the mean ± SEM of three independent experiments. WT -DTT: (n = 6 fields of view/172 cells); WT +DTT: (n = 13/302); cysteine-less -DTT: (n = 6/209); cysteine-less +DTT: (n = 12/326). Significance was tested by an unpaired, two-tailed Student's *t* test (data distribution was assumed to be normal, but this was not formally tested), except for C, which was analyzed using a Kolmogorov–Smirnov test. *, P < 0.05; **, P < 0.01; ***, P < 0.001.

while the tendency of clustering is somewhat lower for the cysteine-less Ire1 compared with the WT (Fig. 1 D). Colocalization of GFP-positive clusters with an ER-targeted variant of dsRed-HDEL confirms the ER localization of WT and cysteine-less Ire1 in DTT-stressed cells (Fig. S1 F). In line with the functional data (Fig. 1, B and C), we conclude that both WT and cysteine-less Ire1 can mount robust responses to acute and prolonged forms of ER stress.

### Cross-linking of Ire1's TMH in ER-derived microsomes
We established a strategy to cross-link single-cysteine variants of Ire1 via CuSO$_4$ in microsomes derived from the ER of stressed

cells (Fig. 2, A–C). Our approach has several advantages over previous attempts: Ire1 is studied (i) as a full-length protein, (ii) at the near-endogenous level, (iii) in its natural, complex membrane environment, (iv) with a spatial resolution of one residue, and (v) in a signaling-active state. In contrast to mercury chloride (HgCl$_2$), which cross-links by forming covalent bonds with two nearby cysteines (Soskine et al., 2002), CuSO$_4$ is "traceless" by catalyzing the oxygen-dependent formation of a disulfide bond (Bass et al., 2007). We performed the cross-linking experiments on ice and with CuSO$_4$ (instead of the more reactive Cu$^{2+}$-phenanthroline) to prevent the loss of signal from unspecific cross-linking and/or aggregation. Even though

## A  Overview of experimental setup

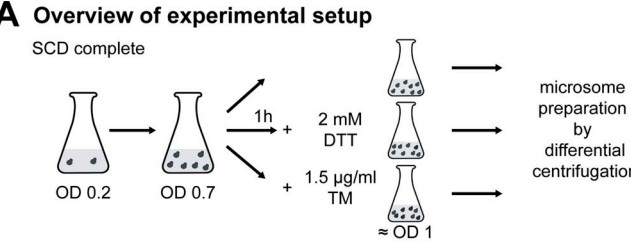

## B  Overview of prepared microsomes used for crosslinking

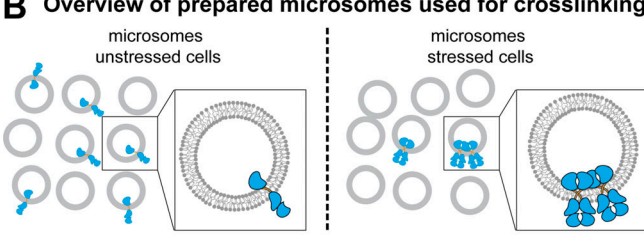

## C  Cysteine crosslinking in microsomes with CuSO₄

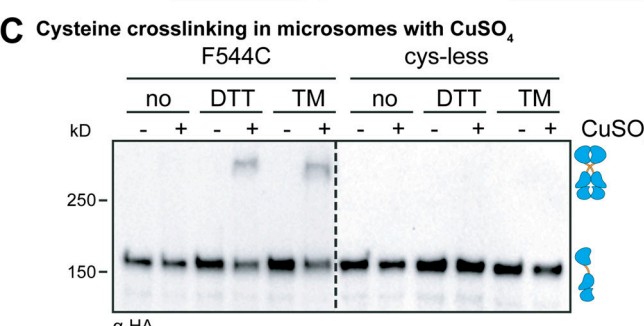

Figure 2. **The cross-linking of Ire1 via single cysteines in microsomes requires CuSO₄ and preformed clusters. (A)** Cultivation of yeast cells for cysteine cross-linking. A culture in SCD medium was inoculated with stationary cells to an $OD_{600}$ of 0.2. After cultivation at 30°C to an $OD_{600}$ of 0.7, the clustering of Ire1 was induced either by DTT (1 h, 2 mM, SCD) or TM (1 h, 1.5 µg/ml, SCD) as indicated. After harvesting, the cells were lysed and used to prepare microsomes. **(B)** Schematic representation of the cysteine cross-linking with CuSO₄. Only microsomes from stressed cells contain clusters of Ire1 clusters that can cross-link via cysteines using CuSO₄. **(C)** Cross-linking of a single-cysteine variant of Ire1 in microsomes. The indicated strains were cultivated in the presence and absence of ER stressors as described in A. 80 OD equivalents of cells were harvested, and microsomes were prepared. 8 µl microsomes (1 mg/ml protein) was mixed with 2 µl of 50 mM CuSO₄, and the sample was incubated on ice for 5 min to catalyze cysteine cross-linking. The reaction was stopped by the addition of 2 µl 1 M NEM, 2 µl 0.5 M EDTA, and 4 µl membrane sample buffer. The resulting samples were analyzed by SDS-PAGE and immunoblotting using anti-HA antibodies.

every cross-linking approach on membrane proteins faces the challenge of varying efficiencies at different depths in the membrane, $Cu^{2+}$-mediated cross-linking has been successfully used to interrogate and establish structure–function relationships of membrane proteins (Falke and Koshland, 1987; Bass et al., 2007; Matthews et al., 2011; Lopez-Redondo et al., 2018). Here we have studied the configuration of Ire1's TMH in UPR-signaling clusters, which are long-lived and stable for minutes (Kimata et al., 2007; Cohen et al., 2017). Because CuSO₄-mediated cross-linking occurs on the same time scale, it can provide useful structural information even though it leads to the formation of covalent disulfide bonds under our experimental conditions.

Cells expressing either a cysteine-less variant of Ire1 or a variant with a single cysteine in the TMH region (F544C) were

cultivated to the mid-exponential phase in minimal medium (Fig. 2 A). These cells were either left untreated or stressed for 1 h with either DTT (2 mM) or TM (1.5 µg/ml) to cause ER stress, which leads to the formation of Ire1 clusters (Kimata et al., 2007; Halbleib et al., 2017; Belyy et al., 2020). We used such an early time point to minimize the contribution of secondary effects from stress- and UPR-dependent reprogramming of the cell. We then isolated crude microsomes from these cells and incubated them on ice for 5 min either in the presence or absence of 10 mM CuSO₄ to catalyze the formation of disulfide bonds by oxidizing nearby sulfhydryl groups (Kobashi, 1968). Given the low copy number of ~260 for Ire1 (Ghaemmaghami et al., 2003) and the fragmentation of the ER during microsome preparation, we expected to detect cross-linking of single-cysteine variants of Ire1 only when it was clustered before the preparation (Fig. 2 B).

Immunoblotting of the resulting samples revealed a prominent, HA-positive signal corresponding to monomeric Ire1 and a less-pronounced HA-positive signal from a band with lower electrophoretic mobility that was only observed when (i) Ire1 contained a single cysteine in the TMH region (F544C), (ii) the microsomes were prepared from stressed cells (either DTT or TM), and (iii) cross-linking was facilitated by CuSO₄ (Fig. 2 C). This suggests a remarkably specific formation of covalent, disulfide bonds between two Ire1 molecules in the TMH region, despite the presence of numerous other, potentially competing membrane proteins with exposed cysteines in the ER. The observed degree of cross-linking was somewhat low considering that up to 70–85% of Ire1 may reside in signaling-active clusters under conditions of ER stress (Aragón et al., 2009). For our cross-linking approach, however, we used a slightly milder condition to induce ER stress (2 mM DTT instead of 10 mM) and performed all experiments with an *IRE1* knock-in strain that provides a more native-like expression level (Halbleib et al., 2017; Aragón et al., 2009). Notably, the signal from the cross-linked species was increased by neither the use of more reactive cross-linking agents (e.g., HgCl₂ or $Cu^{2+}$-phenanthroline) nor by harsher cross-linking conditions (higher temperatures or increased concentrations of the cross-linking agent). In fact, more reactive agents and harsher conditions only caused a loss of the total HA-positive signal, presumably due to an unspecific cross-linking and/or aggregation of Ire1 (data not shown). A co-immunoprecipitation analysis using Flag- and HA-tagged Ire1 variants produced in the same cell and cross-linked in microsomes via the native cysteine (C552) verified that the additional band with low electrophoretic mobility represents disulfide-linked, SDS-resistant dimers of Ire1 (Fig. S2 A). In fact, treating a cross-linked species of Ire1 with heat under reducing conditions revealed full reversibility of disulfide bond formation (Fig. S2 B). We conclude that CuSO₄ can catalyze the formation of disulfide bridges between two neighboring Ire1 molecules, when they are present in preformed clusters and isolated in microsomes from stressed cells.

### A cross-linking screen in the TMH region of Ire1

Next, we generated a set of 13 mutant variants of Ire1, each containing a single cysteine in the TMH region starting with E540C at the transition between the AH and the TMH (Fig. 3 A)

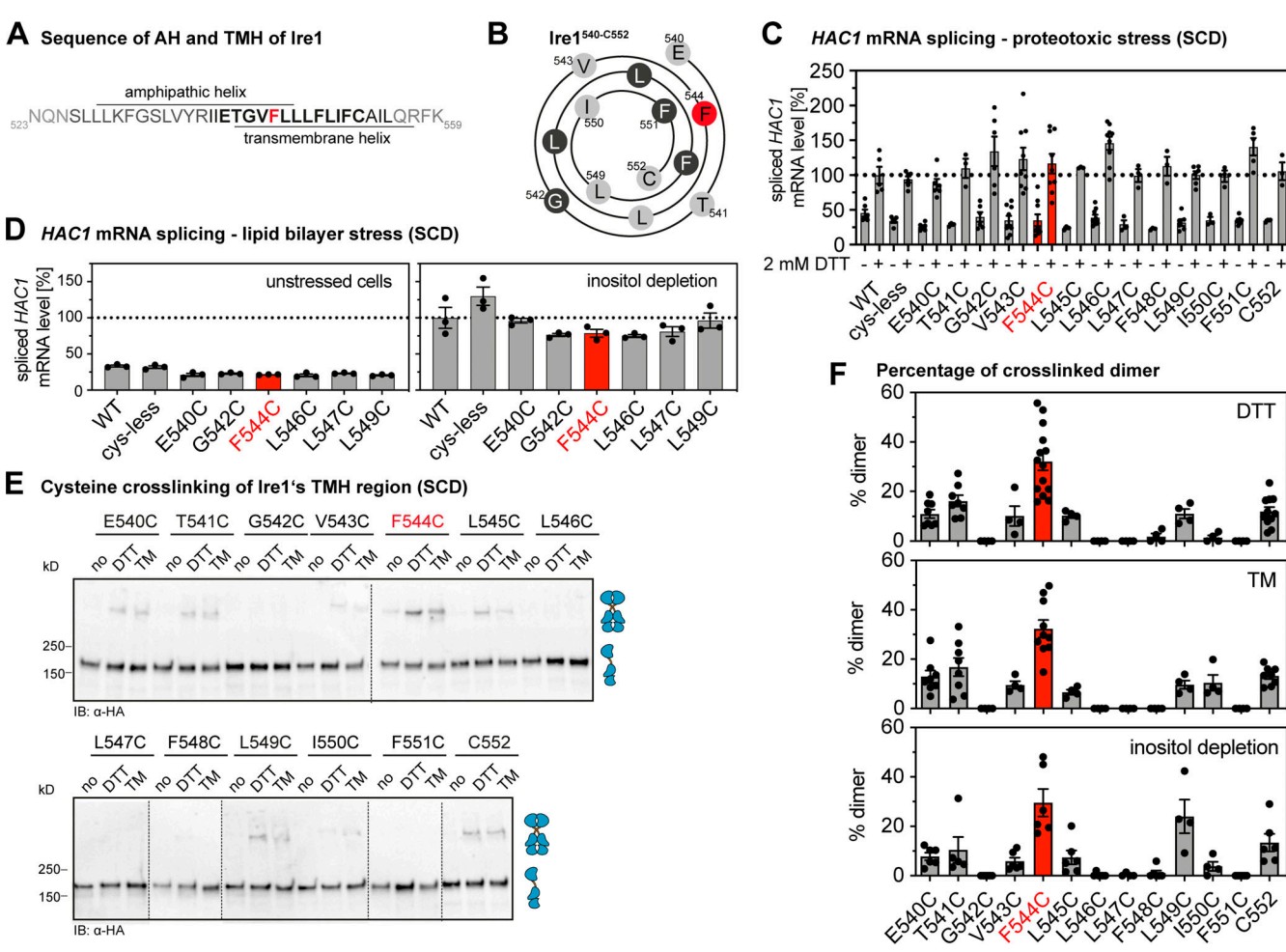

**Figure 3. Systematic cross-linking of cysteines in the TMH region of Ire1 reveals a specific configuration during ER stress. (A)** Primary structure of ER-luminal AH of Ire1 and the short TMH. Almost every residue of the short TMH (shown in bold) was substituted individually by cysteine for the cysteine cross-linking strategy. **(B)** Helical wheel representation of Ire1's TMH (Ire1$^{540-552}$). **(C)** The level of the spliced *HAC1* mRNA was determined from the indicated strains by RT-qPCR for either unstressed cells or cells stressed with 2 mM DTT for 1 h (for details, see E). The data are normalized to the level of the spliced *HAC1* mRNA in stressed cells with a tagged, WT variant of Ire1. Number of independent experiments with technical duplicates for +DTT condition: WT: $n = 5$; cysteine-less: $n = 6$; E540C: $n = 6$; T541C: $n = 3$; G542C: $n = 6$; V543C: $n = 9$; F544C: $n = 9$; L545C: $n = 3$; L546C: $n = 9$; L547C: $n = 3$; F548C: $n = 3$; L549C: $n = 6$; I550C: $n = 3$; F551C: $n = 5$; C552: $n = 3$. Number of experiments with technical duplicates for the unstressed, -DTT condition: WT: $n = 6$; cysteine-less: $n = 5$; E540C: $n = 6$; T541C: $n = 3$; G542C: $n = 6$; V543C: $n = 9$; F544C: $n = 9$; L545C: $n = 3$; L546C: $n = 9$; L547C: $n = 3$; F548C: $n = 3$; L549C: $n = 6$; I550C: $n = 3$; F551C: $n = 6$; C552: $n = 3$. **(D)** The level of the spliced *HAC1* mRNA was determined from the indicated strains by qPCR using stressed (inositol-depleted) and unstressed cells. The data are normalized to the level of the spliced *HAC1* mRNA splicing caused by 2 mM DTT, as determined in C. Number of independent experiments with technical duplicates for the condition of inositol depletion: WT: $n = 3$; cysteine-less: $n = 3$; E540C: $n = 3$; G542C: $n = 3$; F544C: $n = 3$; L546C: $n = 3$; L547C: $n = 3$; L549C: $n = 3$. Number of independent experiments with technical duplicates for the unstressed condition: WT: $n = 3$; cysteine-less: $n = 3$; E540C: $n = 3$; G542C: $n = 3$; F544C: $n = 3$; L546C: $n = 3$; L547C: $n = 3$; L549C: $n = 3$. **(E)** A culture in SCD medium was inoculated with stationary cells to an $OD_{600}$ of 0.2. After cultivation at 30°C to an $OD_{600}$ of 0.7, Ire1 clustering was induced by either DTT (1 h, 2 mM, SCD) or TM (1 h, 1.5 µg/ml, SCD). 8 µl microsomes (1 mg/ml protein) from unstressed (no) and stressed cells was mixed with 2 µl of 50 mM $CuSO_4$, and the sample was incubated on ice for 5 min to catalyze cysteine cross-linking. The reaction was stopped, and the sample was analyzed by SDS-PAGE and immunoblotting using anti-HA antibodies. **(F)** Quantification of cysteine cross-linking of the indicated variants of Ire1 in microsomes isolated cells stressed by DTT, TM, or inositol depletion. Cells were cultivated and treated as described in E. For inositol depletion, a culture was inoculated with exponentially growing cells to an $OD_{600}$ of 0.5 and cultivated for 3 h at 30°C in inositol-free medium (a representative immunoblot after cross-linking is shown in Fig. S3 B). The percentage of cross-linked species was determined by densitometry. Data are represented as the mean ± SEM of at least three independent experiments. IB, immunoblot.

and ending at the native C552, which is substituted to serine in cysteine-less Ire1. Our scanning approach covered more than three helical turns and almost the entire short TMH of Ire1 (Fig. 3, A and B). Systematic cross-linking of these variants can provide important insight into the organization of Ire1's TMH in signaling-active clusters. An important prerequisite for a structural interpretation is that the single-cysteine substitutions

required to form the cross-links affect neither the oligomerization nor the activity of Ire1.

We therefore subjected all Ire1 variants with engineered cysteine residues (E540C to F551C) to a sensitive, cell-based assay to ascertain the functionality of the UPR under conditions of prolonged ER stress (Fig. S3 A). Consistent with the functional role of the AH adjacent to the short TMH (Halbleib

et al., 2017), we found that the substitution of AH residues to cysteine (E540C, T541C, or G542C) impaired the response to ER stress, as evident from an increased sensitivity of the respective cells to DTT (Fig. S3 A). The substitution of TMH residues (V543C–F551C), by contrast, did not cause any apparent functional defect (Fig. S3 A). Hence, these TMH variants are suitable to map the transmembrane architecture via cysteine cross-linking. To validate the functionality of these variants with a more direct assay, we systematically quantified the level of the spliced *HAC1* mRNA in stressed and unstressed cells under conditions of both proteotoxic (Fig. 3 C) and lipid bilayer stress (Fig. 3 D), which is caused by inositol depletion (Promlek et al., 2011; Surma et al., 2013). Because these data are normalized to the level of the spliced *HAC1* mRNA in DTT-stressed cells, it is possible to compare the UPR activity between these conditions (Fig. 3, C and D). We find a similar level of the *HAC1* mRNA in stressed cells and, consistently, a comparable degree of *HAC1* mRNA splicing in cells by either DTT or inositol depletion (Fig. S3 B). All single-cysteine variants were functional and responsive to proteotoxic stress (Fig. 3 C). Likewise, the subset of variants tested under conditions of lipid bilayer stress showed robust activation of the UPR (Fig. 3 D). Because the steady-state level of all Ire1 variants was also comparable (Fig. S3 C), we could proceed with mapping the TMH region.

We subjected the entire set of single-cysteine variants to the cysteine cross-linking procedure (Fig. 3 E and Fig. S3 D) and determined the fraction of cross-linked Ire1 for construct (Fig. 3 F). While some variants (e.g., G542C or L546C) showed no detectable cross-linking, a significant portion of them (e.g., T541C or L549C) could be cross-linked under the given experimental conditions (Fig. 3, E and F). The F544C variant consistently exhibited the highest cross-linking efficiency (Fig. 3 F). Notably, the differences in cross-linking are not caused by an aberrant oligomerization of Ire1, because confocal microscopy experiments with cells cultivated and treated as in the cross-linking experiments demonstrate the same degree of cluster formation of all single-cysteine variants upon ER stress as judged by cluster size and intensity and compared with cysteine-less Ire1 (Fig. S3, D–F).

### Different forms of ER stress converge in a common architecture of the TMH region

Using the cross-linking assay, we could show that the overall pattern of cross-linking residues was independent of the condition of ER stress (Fig. 3 F). Lipid bilayer stress and proteotoxic stress induced by either DTT or TM show essentially the same cross-linking pattern (Fig. 3 F). These data strongly suggest that the overall structural organization of Ire1 is similar for different types of stress, at least in the TMH region. Notably, the L549C mutant showed significant cross-linking in cells stressed by DTT or TM, but even more during inositol depletion (Fig. 3 F). Because F544C, the best-cross-linking residue, and L549C seemingly lie on opposing sites of Ire1's TMH as judged from a helical wheel representation (Fig. 3 B), this raises the question of whether the corresponding residues in the native TMH can face each other at the same time. This point was addressed by molecular dynamics (MD) simulations further below.

Cysteine cross-linking can be used to infer structural models. The observed pattern of cross-linking residues in the TMH of Ire1 is very distinct from those observed in the TMH of the growth hormone receptor (Brooks et al., 2014) and the thrombopoietin receptor (Matthews et al., 2011), which form parallel dimers leading to a helical periodicity of cross-linking. Instead, our cross-linking data suggest an X-shaped configuration of the TMHs with the best cross-linking residue, F554, positioned at the crossing point. Intriguingly, such an arrangement would be consistent with the previously reported, highly tilted orientation of the monomeric TMH of Ire1, which is enforced by the adjacent, ER-luminal AH (Halbleib et al., 2017). However, it is important to realize that cross-links might occur either within dimers of Ire1 or across dimers in higher oligomeric assemblies.

To obtain a structural representation, we used an experimentally validated model of the monomeric TMH region of Ire1 (Halbleib et al., 2017), generated a model of the dimer based on extensive MD simulations in lipid membranes, and integrated the cross-linking data with a particular attention on the contact between the two F544 residues (Fig. 4, Video 1, and Video 2), which were restrained to face each other. The resulting model of the dimeric TMH region highlighted a highly bent configuration of each protomer leading to an X-shaped configuration of the dimer (Fig. 4, Video 1, and Video 2). A substantial membrane thinning (Fig. 4 B) and water penetration around the dimeric TMH region of Ire1 became apparent (Fig. S4 A and Video 1). It is tempting to speculate that this substantial degree of membrane deformation facilitates the access of $Cu^{2+}$ ions to F544C for mediating efficient cross-linking (Fig. 3, E and F). A thorough inspection of the trajectories revealed that the residues at positions F544 and L549 can face their counterpart in an X-shaped dimer at the same time (Fig. S4 C), thereby rendering the corresponding single-cysteine variants capable of cross-linking (Fig. 3 B). This would be unlikely if the TMHs associated in a strictly parallel fashion. Inspecting the dynamics of Ire1's TMH region in a MD simulation over a period of 1,000 ns (Video 1) underscored the stability of the overall X-shaped configuration, which nevertheless allowed for significant relative motions of the TMHs. In summary, our combined approach of biochemical cross-linking and MD simulations established a surprising configuration of Ire1's TMH region with a particularly small interface between the TMHs.

### Validating the structural model of the TMH region of Ire1

Our cross-linking approach indicates that the TMH residue F544 is part of a small interface between Ire1 protomers, which might stabilize the unusual X-shaped transmembrane configuration of Ire1. Aromatic residues TMH residues have been implicated in sensing lipid saturation by Mga2 (W1042; Covino et al., 2016; Ballweg et al., 2020) and lipid bilayer stress by the mammalian IRE1α (W547; Cho et al., 2019). Despite a different position within the ER membrane, we wanted to test a similar role for F544 in Ire1 from baker's yeast. We generated a F544A variant of Ire1, which contained the native C552 in the TMH as the only accessible residue for $Cu^{2+}$-mediated cross-linking. A cell-based assay revealed that the F544A mutant was phenotypically indistinguishable from cysteine-less Ire1 (Fig. 5 A) and the F544C

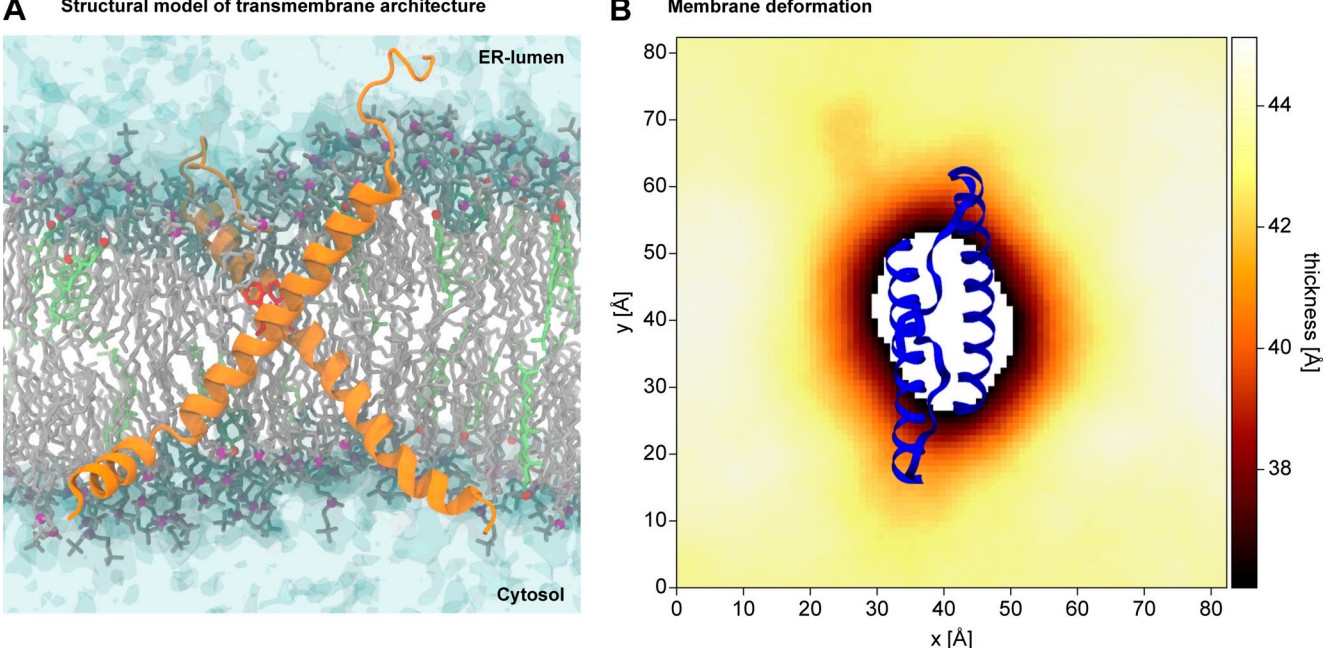

**Figure 4. Structural model of the TMH region of Ire1. (A)** Configuration of a model TMH dimer obtained from atomistic MD simulations. Protomers are shown as an orange ribbon with the two F544 residues highlighted in red. POPC lipids and their phosphate moieties are shown in gray and purple, respectively. Cholesterol molecules and their hydroxyl groups are shown in light green and red, respectively. Water is shown with a transparent surface representation. **(B)** Membrane thickness around the sensor peptide, defined as the average vertical distance between the two phosphate layers. A representative structure of the dimeric TMH region is shown in blue. For the SEM of the thickness profile, see Fig. S4 C.

mutant (Fig. 3 C and Fig. S1 A). This finding was corroborated by Cu²⁺-mediated cross-linking of C552 in microsomes isolated from stressed cells (DTT or TM). The intensity of the band corresponding to cross-linked Ire1 was unaffected by the F544A mutation (Fig. 5 B). Thus, F544 does not contribute to the stability of Ire1 dimers and oligomers even though it is placed near the equivalent residue on the opposing Ire1 protomer.

Previously, we have proposed that a tilted configuration of the monomeric TMH region, which is stabilized by a proximal AH, facilitates Ire1 to sense aberrant membrane properties (Halbleib et al., 2017; Covino et al., 2018). In fact, disrupting the amphipathic character of the AH by an F531R mutation increases the cellular sensitivity to ER stress (Fig. 5 C) and reduces the cross-linking propensity via the native C552 residue in the TMH (Fig. 5 D). These findings provide biochemical evidence that the AH contributes to the stability of either dimeric or oligomeric forms of Ire1, which are challenging to distinguish.

Similarly, when the AH-disrupting mutation F531R was combined with the F544C mutation (at the crossing-point of the X-shaped TMH dimer), we observed only a very mild, yet significant functional defect (Fig. S5 A) and a strongly reduced cross-linking propensity (Fig. S5 B). This robust resistance to DTT is somewhat surprising considering the strongly reduced cross-linking propensity. However, the disruption of the AH changes the placement of the TMH in the membrane and the degree of membrane thinning and water penetration (Halbleib et al., 2017). We speculate that these combined changes would place the polar F544C residue more deeply in the hydrophobic core of the membrane, thereby affecting its propensity to

undergo a Cu²⁺-catalyzed cross-linking, but at the same time favoring Ire1 dimerization. Notably, the F544C mutation alone does not lead to an increased UPR activity and ER stress resistance (Fig. 3, C and D; and Fig. S3 A). In fact, the primary sequence of Ire1's TMH can be systematically mutated (Fig. 3 C and Fig. S3 A), scrambled (in the case of the mammalian IRE1α), or exchanged altogether (Halbleib et al., 2017; Volmer et al., 2013) without causing a detectable functional defect. It therefore seems that a suitably placed polar residue in the TMH, here through the F544C mutation, becomes phenotypically relevant only when Ire1 is otherwise compromised. Beyond that, our data suggest that the overall architecture of the TMH region with an intact AH is relevant for normal UPR function.

### The TMH region of Ire1 makes dimer- and oligomer-specific contacts

Does the cross-linking of engineered cysteines in the TMH occur only within dimers of Ire1 or also across dimers in signaling-active clusters? The x-ray structure of the core ER-luminal domain of Ire1 revealed an interface-1 (IF1) required for dimerization, and an interface-2 (IF2) providing a platform for the back-to-back association of dimers in higher oligomeric assemblies (Credle et al., 2005; Korennykh and Walter, 2012). Consistent with a previous report (van Anken et al., 2014), the formation of microscopically visible clusters of Ire1 is abolished by disrupting either IF1 or IF2 by mutation (T226A/F247A and W426A for IF1 and IF2, respectively; Fig. 6 A). Expectedly, lack of clustering correlates with an increased cellular sensitivity to DTT (Fig. S6).

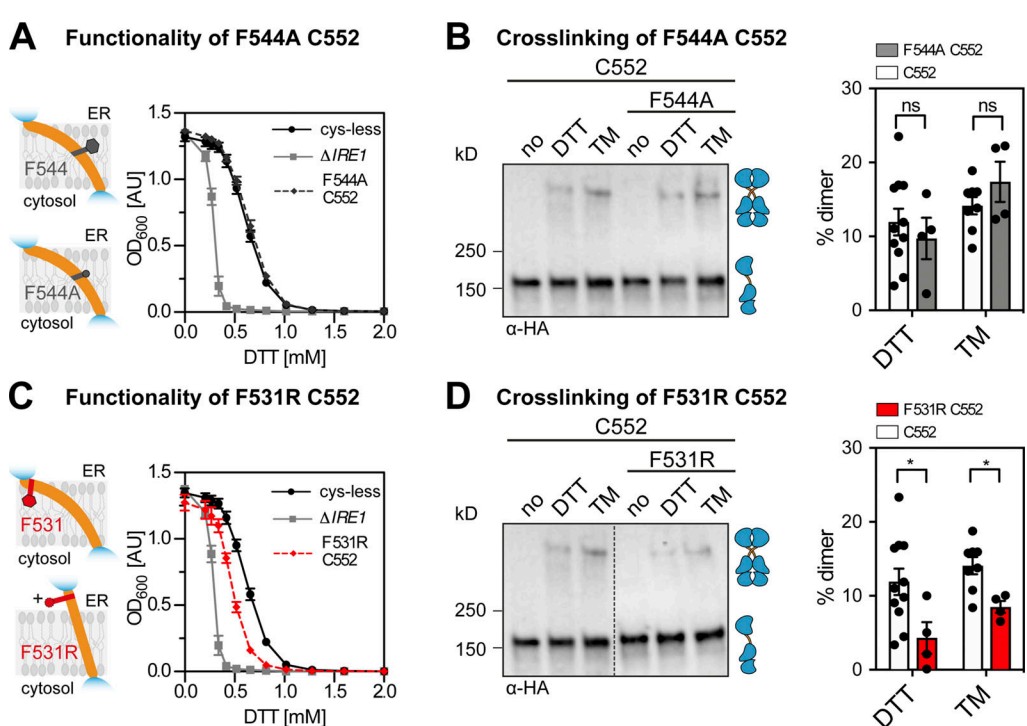

**Figure 5. The impact of mutations in the TMH and the AH of Ire1 on its functionality and cross-linking propensity. (A)** The ER stress resistance of cells expressing the F544A variant of $IRE1_{3xHA-GFP}$ containing the native cysteine 552 was determined. Stationary overnight cultures of the indicated strains were used to inoculate a fresh culture minimal medium to an $OD_{600}$ of 0.2. After cultivation for 5 to 7 h at 30°C, the cells were diluted in 96-well plates to an $OD_{600}$ = 0.01 with prewarmed minimal medium and cultivated in the presence of the indicated concentrations of DTT for 18 h at 30°C. The density of the resulting culture was determined using the $OD_{600}$. Number of experiments including technical replicates: $\Delta IRE1$ ($n$ = 12 from two colonies); cysteine-less ($n$ = 12 from two colonies); F544A/C552 ($n$ = 12 from four colonies). **(B)** The impact of the F544A mutation on Ire1 degree of cross-linking via cysteine 552 was analyzed. The indicated strains were subjected to the cross-linking procedure as outlined in Fig. 3 E. Data for the C552 variant are identical to the data in Fig. 3 F. Number of experiments from DTT-stressed cells including technical duplicates: C552 ($n$ = 11 from 6 colonies); F531R/C552 ($n$ = 4 from two colonies). Number of experiments from TM-stressed cells including technical duplicates: C552 ($n$ = 8 from four colonies); F544A/C552 ($n$ = 4 from two colonies). **(C)** ER stress resistance of indicated cells including a single-cysteine variant (C552) of $IRE1_{3xHA-GFP}$ with an AH-disrupting F531R mutation was determined. The cells were cultivated and treated as in A. The data for $\Delta IRE1$ and cysteine-less $IRE1$ are identical to the data in A. Number of experiments including technical triplicates for F531R/C552 ($n$ = 9 from three colonies). **(D)** The impact of the AH-disrupting F531R mutation on Ire1 cross-linking via cysteine 552 was determined. The indicated strains were subjected to the same cross-linking procedure used for Fig. 3 E. Data for the C552 single-cysteine variant are identical to the data in Fig. 3 F. The immunoblot for the C552 single-cysteine variant is identical to that in B. Number of experiments including technical duplicates for DTT-stressed F531R/C552 cells ($n$ = 4 from two colonies) and TM-stressed F531R/C552 cells ($n$ = 4 from two colonies). All data are represented as the mean ± SEM derived from at least three independent experiments. Significance was tested by an unpaired, two-tailed Student's $t$ test. *, P < 0.05. Data distribution was assumed to be normal, but this was not formally tested.

By disrupting IF2 and leaving IF1 intact, we sought to uncover the contribution of dimeric and oligomeric assemblies to the cross-linking propensity in the TMH region. We focused on F544C marking the crossing-point of the X-shaped TMH region in dimeric Ire1, and on E540C and T541C in the vicinity. Upon disruption of IF2 (W426A), these single-cysteine variants failed to form microscopically visible clusters in stressed cells (Fig. 6 B). The positioning of the engineered cysteine, however, had profound impact on the cellular resistance to DTT in rich medium. The F544C/IF2 double mutant rendered the respective cells more resistant than the IF2 mutant alone, while the T541C/IF2 and E540C/IF2 mutants were highly sensitive to DTT and indistinguishable from cells lacking $IRE1$ altogether (Fig. 6 C). Thus, the functional defect from the IF2 mutation can be alleviated or even aggravated by polar residues in the TMH region.

For interpreting these data, it is important to consider the time frame of the different assays. Cross-linking is performed with microsomes isolated from acutely stressed cells, which

were treated with either DTT or TM for only 1 h. Similarity, clustering of Ire1 is studied by confocal microscopy in acutely cells stressed after 1 h of treatment. The cellular resistance to DTT, however, is scored after 18 h of cultivation. The acute proteotoxic stress caused by DTT or TM treatments has barely any impact on the cellular lipid composition under given conditions (Reinhard et al., 2020). Prolonged treatments, however, cause membrane aberrancies, which can dominate Ire1 activation (Promlek et al., 2011) and which are likely to affect the resulting ER stress resistance phenotype.

To further characterize the impact of the single-cysteine variants on Ire1 function, we determined the level of the spliced $HAC1$ mRNA in time course experiments with DTT-stressed cells (Fig. 6 D). We find that the level of the spliced $HAC1$ mRNA is up-regulated in response to DTT-induced stress for cysteine-less Ire1 and F544C/IF2, but not for the E540C/IF2 and T541C/IF2 double mutants (Fig. 6 D). Notably, we find that UPR activation is delayed for the F544C/IF2 double mutant

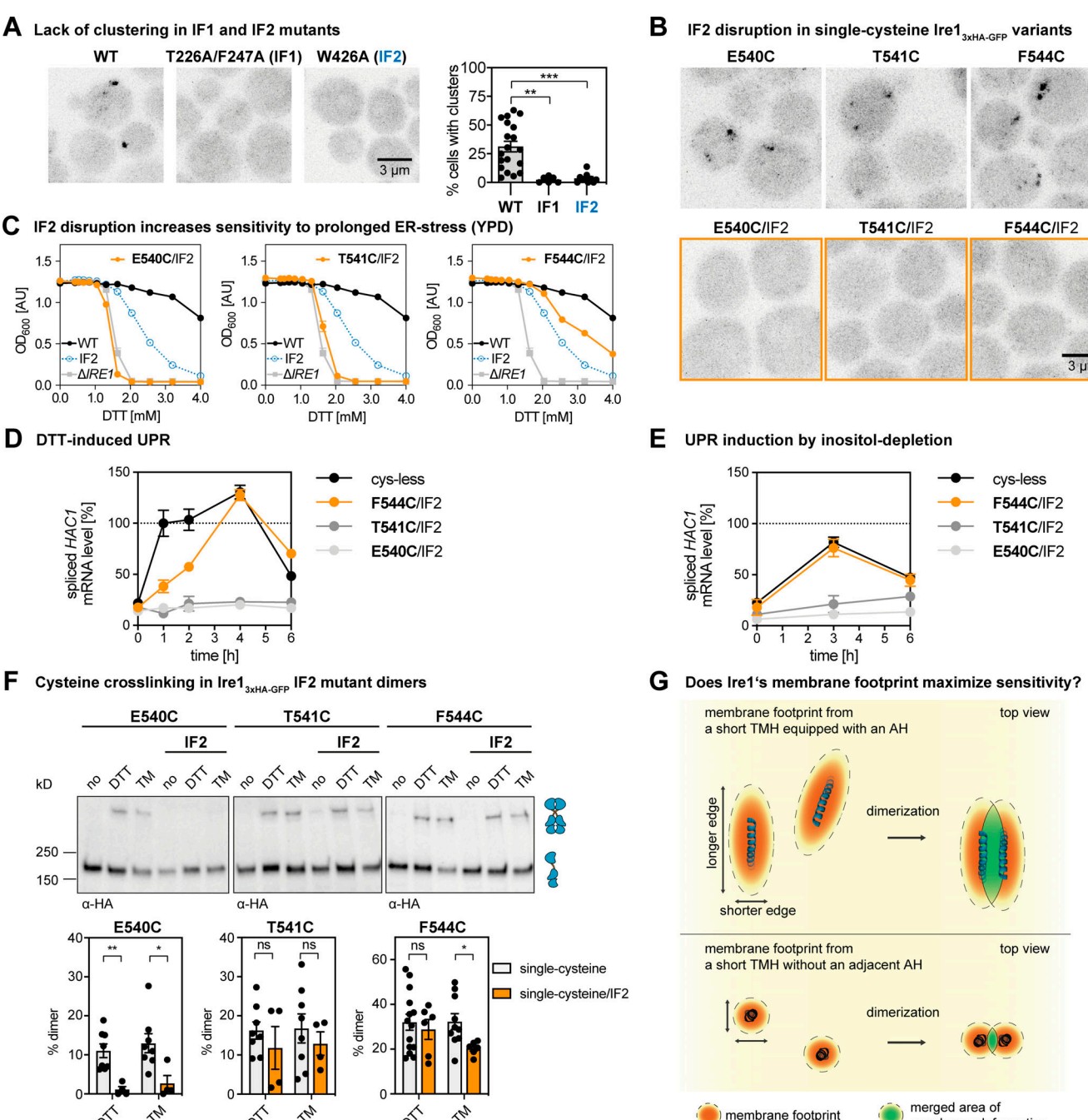

Figure 6.  **Cross-linking occurs within and across dimers of Ire1. (A)** Indicated variants of an *IRE1* knock-in construct (Halbleib et al., 2017) were cultivated and stressed with 2 mM DTT for 1 h as described in Fig. 1 D. A refined, automated counting of Ire1-containing clusters was performed as described in the Online supplemental material. Data from Fig. 1 D were reanalyzed and pooled with new data to yield the following: WT (*n* = 19 fields of view/441 cells); T226A/F247A (IF1; *n* = 6/154); W426A (IF2; *n* = 10/329). All data are represented as the mean ± SEM. Significance was tested by an unpaired, two-tailed Student's *t* test. **, P < 0.01; ***, P < 0.001. **(B)** Clustering in DTT-stressed cells was studied by confocal microscopy of the indicated single-cysteine variants with either an intact or disrupted IF2 (orange). Representative images from at least six independent fields of view are shown. For a quantitative analysis, see Fig. S6 B. **(C)** ER stress resistance of indicated cells was studied in rich medium containing different concentrations of DTT. Data from independent experiments for E540C/IF2 (*n* = 4), T541C/IF2 (*n* = 5), and F544C/IF2 (*n* = 6; IF2 indicates the W426A mutation) are plotted in orange. Reference datasets for ΔIRE1 (*n* = 6), Ire1$_{3xHA-GFP}$ WT (*n* = 6 from two individual colonies), and the IF2 variant (*n* = 12 from two individual colonies) are plotted in gray, black, and blue, respectively. All data are represented as the mean ± SEM. **(D)** The level of the spliced *HAC1* mRNA was determined from the indicated strains by RT-qPCR after treating the cells with 2 mM DTT for the indicated times. The data are normalized to the level of the spliced *HAC1* mRNA in cysteine-less Ire1 after 1 h of treatment and derive from three independent colonies with experimental duplicates (*n* = 6). **(E)** The level of the spliced *HAC1* mRNA was determined for the indicated strains cultivated under inositol-depletion conditions. The data are normalized to the level of the spliced *HAC1* mRNA splicing caused by 2 mM DTT, as determined in D. The data derive from three independent colonies with experimental duplicates (*n* = 6). **(F)** The impact of the IF2-disrupting W426A mutation on cross-linking via the indicated single cysteines was determined. Single- and double-mutant strains were subjected to the cross-linking procedure as in Fig. 3 E. Data for the single mutant variants are replotted from Fig. 3 F. All data are represented as the mean ± SEM. Number of experiments including technical duplicates for DTT-stressed cells:

E540C (*n* = 8 from four colonies), T541C (*n* = 8 from four colonies), F544C (*n* = 14 from seven colonies; identical to data in Fig. 3 F), E540C/IF2 (*n* = 4 from two colonies), T541C/IF2 (*n* = 4 from two colonies), and for F544C/IF2 (*n* = 6 from three colonies). Number of experiments including technical duplicates for TM-stressed cells: E540C (*n* = 8 from four colonies), T541C (*n* = 8 from four colonies), F544C (*n* = 10 from five colonies; identical to data in Fig. 3 F), E540C/IF2 (*n* = 4 from two colonies), T541C/IF2 (*n* = 4 from two colonies), and F544C/IF2 (*n* = 6 from three colonies). Significance was tested by an unpaired, two-tailed Student's *t* test. *, P < 0.05; **, P < 0.01. Data distribution was assumed to be normal, but this was not formally tested. **(G)** Hypothetical model for Ire1's exquisite sensitivity. The membrane-based oligomerization of Ire1 (blue) and unrelated single-pass membrane proteins (black) leads to the coalescence of deformed membrane regions (green). In the case of Ire1, a larger portion of the deformed membrane region can be shared upon dimerization due to the ellipsoid membrane "footprint" and an association via the longer edge of deformation (parallel to the major axis of the ellipse). According to this model, this maximizes the sensitivity of Ire1 to aberrant membrane properties when compared with unrelated single-pass membrane proteins, which can merge only a relatively small portion of their circular membrane "footprint" upon dimerization.

compared with the cysteine-less control strain. Because membrane aberrancies caused by DTT manifest over a time course of several hours (Promlek et al., 2011), this suggests that the F544C/IF2 double mutant may respond predominantly to such membrane-based stresses. This interpretation is further confounded by the observation that the two other double mutants, E540C/IF2 and T541C/IF2, with mutations in the functionally critical AH (S526-V543) cannot respond to this type of prolonged DTT stress (Fig. 6 D). In order cross-validate our interpretation, we studied the response of the same set of strains to lipid bilayer stress caused by inositol depletion (Fig. 6 E). While the F544C/IF2 mutation exhibited an almost identical response to inositol depletion as the control strain, the two E540/IF2 and T541/IF2 variants showed a massively impaired response (Fig. 6 E). Because E540 and T541 are part of the AH, these data underscore the central importance of the AH for sensing lipid bilayer stress. More importantly, however, these data suggest that membrane sensitivity of Ire1 may be particularly important for dealing with prolonged forms of ER stress caused by proteotoxic agents.

Next, we subjected these double mutant variants to the cross-linking procedure (Fig. 6 F). Cross-linking via E540C in DTT- and TM-stressed cells was abolished by the IF2 mutation, while the cross-links observed for the T541C and F544C variants were only marginally affected (Fig. 6 F). This suggests that the cross-links of T541C and F544C are formed within Ire1 dimers, while E540C cross-links across dimers. Importantly, these data not only validate the particular position of F544 at the crossing point of two TMHs in two adjacent Ire1 protomers, but also provide direct, biochemical evidence that the unusual X-shaped transmembrane architecture might laterally associate to form higher oligomers. Notably, such lateral "stacking" of the transmembrane domain in signaling-active clusters would be consistent with the complex, elongated organization of clusters as recently observed by super-resolution microscopy for IRE1α (Belyy et al., 2020). On the functional level, our data show that the dimerization of Ire1 is not sufficient to mediate resistance to ER stress: the T541C/IF2 variant forms dimers that can be cross-linked (Fig. 6 D), but it does not render cells more resistant to DTT than cells lacking Ire1 (Fig. 6 C), nor does it up-regulate the level of the *HAC1* mRNA in response to ER stress (Fig. 6, D and E). A suitably positioned polar residue (here F544C) leaves the membrane-sensitive AH intact and increases the cellular ER stress resistance in the IF2/F544C double mutant compared with a single IF2 mutant (Fig. 6 C). Thus, seemingly subtle changes in the TMH region can have substantial impact on the ER stress

resistance phenotype, especially when the normal function of Ire1 is compromised.

## Discussion

Here, we establish a structural model of Ire1's TMH region in signaling-active clusters (Fig. 4). In a previous study, we have established a model of Ire1's monomeric TMH region (Halbleib et al., 2017), but its organization in dimers and higher oligomers, especially in the complex environment of the ER membrane, remained unexplored. Predicting a dimeric structure based on a model for the monomer is not trivial, as the two protomers can be arranged in various ways and might undergo substantial conformational changes upon oligomerization. Based on a systematic cysteine cross-linking approach in native membranes and aided by MD simulations, we show that the neighboring TMHs in clusters of Ire1 organize in an X-shaped configuration.

Our model of the transmembrane organization provides intriguing insights into the membrane-deforming potential of Ire1 (Fig. 4, A and B; Fig. S4; and Video 1). Positively charged residues at the cytosolic end of the TMH (Fig. 3 A) and the previously identified ER-luminal AH (Halbleib et al., 2017) cooperate in squeezing the lipid bilayer (Fig. 4, A and B; and Fig. S4 A). This deformation is most prominent at the intersection of the two protomers reaching almost to the level of the lipid bilayer center (Fig. 4 B and Video 1). Membrane squeezing and the associated disordering of lipid acyl chains come at energetic costs, which are affected by the composition and collective physicochemical properties of the surrounding bilayer (Radanović et al., 2018; Covino et al., 2018). The higher this cost (e.g., due to increased lipid saturation, inositol depletion, or membrane aberrancies from prolonged proteotoxic stresses), the higher the free energy gain from coalescing these regions and thus the propensity of Ire1 to oligomerize.

The specific way each membrane protein locally deforms the bilayer, referred to as membrane "footprints" (Haselwandter and MacKinnon, 2018) or "fingerprints" (Corradi et al., 2018), could be at the origin of membrane sensitivity and, more generally, could control the organization of supramolecular assemblies (Corradi et al., 2018). Is it possible that the unusual TMH region of Ire1 and its resulting footprint serve a specific function? We speculate that the combination of a short TMH with an AH inserting deep into the bilayer contributes to Ire1's exquisite sensitivity to aberrant ER membrane stiffening. The region of membrane compression around monomeric Ire1 is, when viewed from the top, not of circular shape but ellipsoid

due to the membrane-inserted AH (Fig. 6 G; Halbleib et al., 2017). Based on simple geometric considerations, it is conceivable that the total extent of membrane deformation contributing to the free energy of dimerization depends on how precisely the two TMH regions are arranged toward each other. Our structural model of the dimeric TMH suggests that the two protomers associate via the longer edge of membrane deformation (parallel to the major axis of the ellipse; Fig. 4, A and B), thereby maximizing the area of coalescence (Fig. 6 G, top) and minimizing the free energy. We speculate that Ire1 is more responsive to aberrant membrane stiffening than other single-pass transmembrane proteins with short TMHs but without AHs. Because these proteins also lack the characteristic ellipsoid shape of membrane deformation (Kaiser et al., 2011), they coalesce only a smaller area of their footprints upon dimerization (Fig. 6 G, bottom). It will be intriguing to study the membrane-driven dimerization and oligomerization of Ire1 side by side with other single-pass membrane proteins exhibiting distinct membrane footprints using advanced microscopic tools such as single-molecule photobleaching (Chadda et al., 2016).

Our data also provide evidence that cross-linking can occur across dimers of Ire1 (Fig. 6 F), thereby suggesting that the X-shaped dimeric arrangements of the TMH region can laterally associate and "stack" in the plane of the membrane. We propose that it is the characteristic, ellipsoid shape of membrane deformation by monomeric Ire1 and the unusual mode of dimerization and oligomerization that maximize the sensitivity of Ire1 to aberrant membrane properties.

Our structural and functional analyses suggest that the oligomeric state of Ire1 is stabilized by the overall transmembrane architecture and the membrane-embedded AH, but not by specific interactions between residues in the TMH. Disrupting the AH, which also disrupts transmembrane architecture (Covino et al., 2018), increases the cellular sensitivity to ER stress (Fig. 5 C). In contrast, the F544A mutation at the intersection of neighboring TMHs causes no functional defect (Fig. 5, A and B). Instead of maximizing the interface between the TMHs for forming a more stable protein:protein interaction, they are kept in a configuration where only a few TMH residues can contact the opposing protomer. However, they are driven together via a membrane-based mechanism and thus are particularly sensitive to the properties of the surrounding membrane (Covino et al., 2018).

Strikingly, our data provide evidence that different forms of ER stress converge in a single overall transmembrane architecture of Ire1. We observed remarkably similar cross-linking patterns in the context of lipid bilayer stress and proteotoxic stress (Fig. 3 F). This suggests that the X-shaped configuration in the TMH region is maintained in the signaling-active clusters even under largely distinct conditions of ER stress. Neither the oligomerization of Ire1 per se nor lipid bilayer stress seems to cause major conformational changes in the TMH region of the individual protomers. Based on our data, we speculate that Ire1 mounts a single response to different types of ER stress, but with distinct temporal patterns of activation. Proteotoxic stress caused by DTT or TM is characterized by two phases: an early phase of a rapid UPR activation with little to no changes in the lipid composition, and a second, slower phase characterized by

a build-up of membrane aberrancies (Promlek et al., 2011; Reinhard et al., 2020). While these membrane aberrancies remain poorly characterized, they serve as a robust signal for Ire1 activation (Fig. 6 D; Promlek et al., 2011). The lipid bilayer stress caused from inositol depletion, in contrast, lacks the early phase of UPR activation. It manifests slowly and causes a distinct temporal pattern of UPR activation (Fig. 6, D and E). It will be interesting to study whether different temporal patterns of UPR activation are sufficient to give rise to largely distinct transcriptional programs or if, alternatively, Ire1 can custom tailor its output via yet unknown mechanisms (Hetz et al., 2020; Ho et al., 2020; Fun and Thibault, 2020).

Our cross-linking data suggest a similar transmembrane architecture in Ire1 in response to proteotoxic and lipid bilayer stress (Gardner and Walter, 2011; Halbleib et al., 2017). While we cannot formally exclude conformational changes in other parts of the protein, we do not find evidence that Ire1 custom-tailors its signaling output via conformational changes in the TMH region. Based on our cross-linking data and the observed temporal patterns of activation for different mutants of Ire1 (Fig. 6 E), we suggest that the complex metabolic, transcriptional, and nontranscriptional adaptations to different forms of ER stress do not reflect distinct functional modes of Ire1. Instead, we propose that different degrees of oligomerization and different rates of Ire1 activation and inactivation are sufficient to drive differently stressed cells into distinct physiological states.

Our combined results lead to the following model of UPR activation. Both accumulating unfolded proteins and lipid bilayer stress lead to the oligomerization of Ire1 and the formation of signaling-active clusters (Korennykh and Walter, 2012). Under these conditions, the cytosolic effector domains "follow" the oligomerization of the ER-luminal domain and the TMH region. A large diversity of ER-luminal and cytosolic interactors including chaperones can tune and specify the activity of mammalian UPR transducers (Sepulveda et al., 2018; Amin-Wetzel et al., 2017). This may reflect a way to custom tailor the globally acting UPR to different cell types with distinct protein folding requirements at steady-state and during differentiation. Lipid bilayer stress activates the UPR in both yeast and mammals via a membrane-based mechanism and does not require the binding of unfolded proteins to the ER-luminal domain and/or associated chaperones (Promlek et al., 2011; Halbleib et al., 2017; Volmer et al., 2013). Furthermore, our findings underscore the importance of Ire1's membrane sensitivity to deal with the stress caused by prolonged cellular treatments with proteotoxic agents (Promlek et al., 2011). Our data from direct, cross-linking experiments suggest that both proteotoxic and lipid bilayer stress converge in a single overall architecture of the TMH region. We propose that Ire1's distinct signaling outputs to different forms of ER stress reflect a different temporal pattern of Ire1 activation rather than different qualities of signaling.

## Materials and methods
### Reagents, antibodies, strains, and plasmids
All chemicals and reagents used in this study were purchased from Sigma-Aldrich, Carl Roth, or Millipore and are of analytical

or higher grade. The following antibodies were used: mouse anti-Flag monoclonal (M2; Santa Cruz), rat anti-HA monoclonal (3F19; Roche), mouse anti-Dpm1 monoclonal (5C5A7; Life Technologies), mouse anti-Pgk1 (22C5D8; Life Technologies), mouse anti-MBP monoclonal (NEB), anti-mouse-HRP (Dianova), and anti-rat-HRP (Dianova). All strains and plasmids used in this study are listed in Table S1 and Table S2, respectively.

## Generation of a cysteine-less construct and a Flag-tag variant of *IRE1*

A cysteine-less construct of *IRE1* was generated based on a previously described knock-in construct (Halbleib et al., 2017). This construct comprises the *IRE1* promoter (–1 to –551 bp), the *IRE1* gene including a coding sequence for a 3xHA tag and a monomeric version of yeGFP (A206R$^{yeGFP}$) inserted at the position of H875, and the *IRE1* endogenous 5′ terminator on the plasmid pcDNA3.1-IRE1$_{3xHA-GFP}$ (Halbleib et al., 2017). A cysteine-less variant was generated by site-directed mutagenesis. Cysteine 48 (C48$^{yeGFP}$) of the monomeric yeGFP was substituted to serine, while cysteine 70 (C70$^{yeGFP}$) remained in the final construct (Costantini et al., 2015; Ormö et al., 1996). Single-cysteine variants were generated by site-directed mutagenesis.

Plasmids encoding either single-cysteine variants or cysteine-less Ire1 (Table S2) were linearized using *Hind*III and *Xho*I restriction enzymes and used for transforming our previously established cloning strain lacking both the *IRE1* gene and its promoter. Strains used in this study are listed in Table S1. Additionally, a Flag-tagged cysteine-less Ire1 version based on the *CEN*-based Ire1 construct from the pPW1628/pEv200 plasmid was generated. The 3xHA epitope tag in the knock-in construct was replaced by a 3xFlag epitope tag using the Q5 site-directed mutagenesis kit (NEB). The newly generated knock in sequence was amplified in a multi-step PCR reaction adding the terminator sequence from the pEv200 plasmid and *Bss*HI and *Hind*III restriction site. The transfer of the *IRE1*$_{3xFlag-GFP}$ sequence in the *CEN*-based pPW1628/pEv200 plasmid was performed using *Bss*HI/*Hind*III restriction sites.

## Cultivation and live cell confocal microscopy

The yeast strains were cultivated at 30°C on agar plates containing SCD complete medium or selection medium. Liquid yeast cultures either in SCD or YPD (the pH of the medium was not adjusted) were inoculated with a single colony and typically cultivated at 30°C for a minimum of 18 h to reach the stationary phase. This overnight culture was used to inoculate a fresh culture to an OD$_{600}$ = 0.2, which was cultivated until the mid-exponential phase. For microsomal membrane preparation, stationary cells were used to inoculate a fresh culture in SCD complete medium to an OD$_{600}$ of 0.2. After cultivation at 30°C to an OD$_{600}$ of 0.7, the cells were either left untreated or stressed with either 2 mM DTT or 1.5 µg/ml TM for 1 h. For inositol depletion, exponentially growing cells were washed with SCD complete without inositol and then used to inoculate the main culture to an OD$_{600}$ of 0.5 in SCD complete without inositol, which was further cultivated for 3 h.

## Live cell confocal microscopy and image analysis

A fresh culture in SCD medium was inoculated to an OD$_{600}$ = 0.2 and cultivated for 5–5.5 h at 30°C and under constant agitation at 220 rpm. To induce ER stress, DTT was added to a final concentration of 2 mM followed by additional cultivation for 1 h. The cells were harvested by centrifugation and mounted on microscopic slides coated with a thin layer of SCD containing 1.5% agarose for immobilization. Microscopy was performed at 23 ± 2°C using a Zeiss LSM 780 confocal laser scanning microscope (Carl Zeiss Microscopy GmbH) with inverted stage and spectral detection, a Plan-Apochromat 63× 1.40 NA oil immersion objective with immersol 518 f, and the acquisition software ZEN2012 (Carl Zeiss Microscopy GmbH). GFP fluorescence was excited at 488 nm, and the emission was detected between 493 and 598 nm. Transmission images were simultaneously recorded using differential interference contrast optics. Z-stacks (450-nm step size, 62.1-µm pinhole size) were recorded. When multiple fluorophores were imaged (Fig. S6 F), GFP was excited at 488 nm and dsRed at 561 nm, and emission was detected at 493–557 nm and 592–704 nm, respectively. For multi-fluorophore images, a Z-stack step size of 372 nm with a pinhole diameter of 80.3 µm was used. Image stacks were corrected for potential x-y drift using the Fiji plugin StackReg (Thévenaz et al., 1998; Schindelin et al., 2012). Maximum intensity and sum projections were created, while the contrast was adjusted equally for all images using Fiji (Schindelin et al., 2012). Individual cells and clusters of Ire1 were identified by automated segmentation using CellProfiler (McQuin et al., 2018). In brief, the cellular areas were determined for each image based on sum projections of recorded z-stacks and the cellular auto-fluorescence. After smoothing with a median filter, potential cells were identified by global thresholding (minimum cross-entropy). Objects outside the diameter restraint of 1.9c6.3 µm were discarded. Cells being too bright (a high autofluorescence indicates cell death) were omitted from further analysis if the mean intensity of a potential cell exceeded the mean intensity of all potential cells within an image by >30%. Clusters of Ire1 within cells were identified in maximum intensity projections using a threshold of 1.5 times the mean intensity of the identified cells. Potential clusters outside the diameter range 0.3–0.9 µm were discarded. The strain RE773 IRE1$_{3xHA-yeGFP}$ E540C/IF2 showed substantial signs of cell death (increased auto-fluorescence) when challenged with DTT. Therefore, all microscopic images represented in and used for Fig. 6 and Fig. S6 B were reanalyzed and subjected to more stringent parameters to avoid false positive identifications of Ire1 clusters. Cells were not considered if their mean intensity was 10% above average. Structures with diameters from 0.3 to 1.2 µm were initially allowed as potential clusters, but only counted if their maximum intensity was at least 2.5 times higher than the mean intensity of the respective cell. Furthermore, if >3.5% of a cell area was covered by potential clusters, the cell was considered as unfit and counted as free of clusters.

## Assaying the resistance to ER stress

The cellular resistance to ER stress caused by DTT was assayed using a sensitive growth assay (Halbleib et al., 2017). Stationary

overnight cultures were used to inoculate a fresh culture to an $OD_{600}$ of 0.2. After cultivation for 5–7 h at 30°C, the cells were diluted with prewarmed medium to an $OD_{600}$ of 0.05. 50 µl of these diluted cultures was mixed in a 96-well plate with 180 µl of medium and 20 µl of a DTT dilution series leading to a final concentration of DTT between 0 and 2 mM and 0 and 4 mM, respectively. After incubation at 30°C for 18 h, the cultures were thoroughly mixed, and 200 µl of the cell suspension was transferred to a fresh 96-well plate for determining the density of the culture via spectrophotometers using the $OD_{600}/OD_{620}$.

### RNA preparation, cDNA synthesis, and qPCR analysis

The level of the spliced *HAC1* mRNA and the *PDI1* mRNA in stressed and unstressed cells was determined via RT-qPCR using Oligo(dT) primers, the Superscript II RT protocol (Invitrogen), the ORA qPCR Green ROX L Mix (HighQu), and a Piko Real PCR system (Thermo Fisher Scientific). The RNA was prepared from 5 OD equivalents of stressed and unstressed cells using the RNeasy Plus RNA Isolation Kit (Qiagen). 500 ng RNA of the total isolated RNA was used as a template for the synthesis of cDNA using Oligo(dT) primers and the Superscript II RT protocol (Invitrogen). qPCR was performed using ORA qPCR Green ROX L Mix (HighQu) in a Piko Real PCR system (Thermo Fisher Scientific). The following primers were used at a final concentration of 400 nM: *HAC1s* forward primer: 5′-CTTTGTCGCCCA AGAGTATGCG-3′; *HAC1s* reverse primer: 5′-ACTGCGCTTCTG GATTACGC-3′; *ACT1* forward primer: 5′-TGTCACCAACTGGGA CGATA-3′; *ACT1* reverse primer: 5′-AACCAGCGTAAATTGGAA CG-3′; *PDI1* forward primer: 5′-GATCGATTACGAGGGACCTAG A-3′; and *PDI1* reverse primer: 5′-GCGGAGGGCAAGTAAATA GAA-3′.

The qPCR program included the following steps: (1) 95°C, 15 min; (2) 95°C, 20 s; (3) 58°C, 20 s; (4) 72°C, 30 s; and (5) 72°C, 5 min; steps 2–4 were repeated 40 times. For quantifying the level of the *PDI1* mRNA and the spliced *HAC1* mRNA, we used the comparative ΔΔCT method using normalization to *ACT1* levels (StepOnePlus user Manual, Applied Biosystems).For amplifying both cDNAs generated from the spliced and unspliced *HAC1* mRNA, we used the following primers at a final concentration of 400 nM and previously established PCR conditions (Promlek et al., 2011): *HAC1* splicing forward primer: 5′-TACAGGGATTTC CAGAGCACG-3′; and *HAC1* splicing reverse primer: 5′-TGAAGT GATGAAGAAATCATTCAATTC-3′.

### Preparation of cell lysates and immunoblotting

Lysates were prepared from exponentially growing cells, which were harvested by centrifugation (3,000 × *g*, 5 min, 4°C) and then washed once with double-distilled water and once with PBS. During washing, the cells were transferred into 1.5-ml reaction tubes, allowing for a more rapid centrifugation (8,000 × *g*, 20 s, 4°C). The tubes with the washed cell pellet were placed in a –80°C freezer and stored until further use. For preparing the lysate, either 5 or 20 OD equivalents were resuspended in 400 µl or 1,000 µl lysis buffer (PBS containing 10 µg/ml chymostatin, 10 µg/ml antipain, and 10 µg/ml pepstatin), respectively. After addition of either 100 µl or 500 µl of zirconia beads, respectively, the cells were disrupted by bead beating for 5 min at 4°C.

Four parts of the resulting lysate were mixed with one part of 5× reducing sample buffer (8 M urea, 0.1 M Tris-HCl, pH 6.8, 5 mM EDTA, 3.2% [wt/vol] SDS, 0.15% [wt/vol] bromphenol blue, 4% [vol/vol] glycerol, and 4% [vol/vol] β-mercaptoethanol) and then incubated at 95°C for 10 min to fully unfold and solubilize the proteins therein. 0.1 OD equivalents of the resulting sample was subjected to SDS-PAGE, and the proteins were separated on 4–15% Mini-PROTEAN-TGX strain-free gels (BioRad). For subsequent immunoblotting, proteins were transferred from the gel to methanol-activated polyvinylidene difluoride membranes using semi-dry Western blotting. Specific proteins were detected using antigen-specific primary antibodies, HRP-coupled secondary antibodies, and chemiluminescence.

### Microsomal membrane preparation

80 $OD_{600}$ equivalents were harvested from a mid-exponential culture by centrifugation (3,000 ×*g*, 5 min, 4°C), washed with PBS, and stored at –80°C. All steps of membrane fractionation were performed on ice or at 4°C. Cells were resuspended in 1.5 ml lysis buffer (50 mM Hepes, pH 7.0, 150 mM NaCl, 1 mM EDTA, 10 µg/ml chymostatin, 10 µg/ml antipain, and 10 µg/ml pepstatin). For cysteine cross-linking experiments, a buffer without EDTA was used. After cell disruption using zirconia beads (Roth) and a bead beater (2 × 5 min), cell debris was removed by centrifugation (800 ×*g*, 5 min, 4°C; and 5,000 ×*g*, 10 min, 4°C). The supernatant was centrifuged (100,000 ×*g*, 45 min, 4°C) to obtain crude microsomes in the pellet. Microsomes were resuspended in 1.4 ml lysis buffer, sonicated for homogenization (50%, 5 × 1 s, MS72 tip on a sonifier cell disrupter from Branson Ultrasonic), snap-frozen in liquid $N_2$, and stored in aliquots at –80°C.

### Test of membrane integration

The cleared supernatant of a 5,000 ×*g* step was divided into equal parts, which were then mixed with an equal volume of lysis buffer supplemented with 0.2 M $Na_2CO_3$, resulting in a final pH of 11, 5 M urea, and 2% Triton X-100 or without additional additives. After incubation for 1 h on a rotator, these samples were centrifuged (100,000 ×*g*, 45 min, 4°C) to separate soluble from insoluble material. The supernatant and pellets from these fractions corresponding to 0.2 OD equivalents were further analyzed by SDS-PAGE and immunoblotting.

### CuSO₄-induced cysteine cross-linking

Microsomes were thawed on ice. 8 µl microsomes (1 ± 0.2 mg/ml protein) were mixed either with 2 µl of 50 mM $CuSO_4$ or 2 µl double-distilled water and then incubated for 5 min on ice. The reaction was stopped with 8 µl of membrane sample buffer (4 M urea, 50 mM Tris-HCl, pH 6.8, 1.6% [wt/vol] SDS, 0.01% [wt/vol] bromophenol blue, and 2% [vol/vol] glycerol) containing 125 mM EDTA and 250 mM *N*-Ethylmaleinimid (NEM). The samples were analyzed by SDS-PAGE and immunoblotting with chemiluminescence detection. The percentage of cross-linked dimer was determined via densitometry with Fiji (Schindelin et al., 2012) using the bands corresponding to the monomeric and covalently cross-linked protein.

## Immunoprecipitation from microsomes after CuSO₄-induced cysteine cross-linking

300 µl of microsomes with a typical protein concentration of 1 mg/ml were incubated with 12.5 µl 250 mM $CuSO_4$ (final concentration of 10 mM) for 5 min on ice. The reaction was stopped by adjusting the sample to a final concentration of 50 mM EDTA and 111 mM NEM by adding 30 µl of 0.5 M EDTA stock solution and 44 µl of 1 M NEM stock solution, respectively. The final volume was adjusted to 1.3 ml with lysis buffer with a final concentration of 5 mM EDTA. The $CuSO_4$ concentration was thus reduced to 2.4 mM and the NEM concentration to 33.6 mM, respectively.

After cross-linking, the microsomes were solubilized using 2% Triton X-100 and incubated for 1 h at 4°C under constant agitation. Insoluble material was removed by centrifugation (20,000 ×g, 10 min, 4°C). The resulting supernatant was incubated with 8 µl Flag beads (Sigma-Aldrich), equilibrated with immunoprecipitation (IP) wash buffer (lysis buffer + 5 mM EDTA + 0.2% Triton X-100), for 3 h under constant shaking. Flag beads were washed five times with IP wash buffer by centrifugation (8,000 ×g, 30 s, 4°C). For elution, the Flag beads were incubated with 10 µl IP-Wash and 10 µl 5× reducing sample buffer for 5 min at 95°C, which did not disrupt the disulfide bond formed between two protomers of Ire1. These samples were analyzed by SDS-PAGE and immunoblotting.

## Modeling of the transmembrane dimer of Ire1 and MD simulations

The dimeric TMH region of Ire1 was modeled using a 56–amino acid–long peptide, 516-SRELD EKNQNSLLLK FGSLVYRIIE TGVFLLLLFLI FCAILQRFKI LPPLYVLLSK I-571. We extracted an equilibrated, monomeric configuration of the peptide from a previously performed 10-µs-long equilibrium MD simulation. We duplicated the configuration in order to create a new system containing two identical protomers. We then rotated and translated one of the two protomers to form a dimer structure, such that the two F544s faced each other with the distance between their Cβ atoms at around 0.7 nm. A short energy minimization in solution resolved all steric clashes between side chains. The structure of the model dimer was prepared by using gromacs/2019.3 tools (Abraham et al., 2015) and VMD (Humphrey et al., 1996). We used Charmm-GUI (Wu et al., 2014; Lee et al., 2016) to reconstitute the dimer in a bilayer containing 248 POPC and 62 cholesterol molecules modeled in the Charmm36m force field (Klauda et al., 2010; Best et al., 2012). We solvated the system with 24813 TIP3P water molecules and 72 chloride and 66 sodium ions, corresponding to a salt concentration of 150 mM.

## Equilibrium and restrained simulations of the dimer model

After an initial energy minimization and quick relaxation, we equilibrated the dimer model in the bilayer. We first ran a 50-ns-long simulation restraining the position of protein atoms by using harmonic potentials with force-constants (in units of kJ mol⁻¹ nm⁻²) of 500 for backbone atoms and 200 for side chain atoms. We then ran further 50 ns lowering the force-constants to 200 and 50, respectively. After this equilibration, we relieved

all restraints and ran a 1,000-ns-long MD simulation, where the system evolved according to its unbiased dynamics. We ran both the restrained equilibration and unbiased production simulation in gromacs/2019.3 using a time step of 2 fs. Electrostatic interactions were evaluated with the Particle-Mesh-Ewald method (Essmann et al., 1995). We maintained a constant temperature of 303 K (Bussi et al., 2007), applying separate thermostats on the protein, membrane, and solvent with a characteristic time of 1 ps. We applied the semi-isotropic Berendsden barostat (Berendsen et al., 1984) for the restrained equilibration, and the Parrinello–Rahman barostat (Parrinello and Rahman, 1981) for the production runs, acting separately on the x-y plane and z direction to maintain a constant pressure of 1 atm, and with a characteristic time of 5 ps. We constrained all hydrogen bonds with the LINCS algorithm (Hess et al., 1998). Molecular visualizations were obtained with VMD and rendered with Tachyon.

## Data representation and replicates

All data are represented as the average ± SEM if not stated otherwise. The number of the biological and technical replicates are provided in the Online supplemental material. Statistical tests were performed with Prism 8 for macOS Version 8.4.0.

## Online supplemental material

Fig. S1 shows protein levels of cysteine-less Ire1 and characterization of its membrane association. Fig. S2 shows validation of a covalent, reversible cross-linking of Ire1 homodimers via disulfide bridges. Fig. S3 shows functionality of cysteine mutants and their cross-linking potential in lipid bilayer stress conditions. Fig. S4 shows that the dimeric TMH region of Ire1 deforms the membrane. Fig. S5 shows that a mutation of the AH affects Ire1 function and cross-linking propensity. Fig. S6 shows that disrupting ER-luminal interfaces for dimerization (IF1) and oligomerization (IF2) of Ire1 impairs cellular ER-stress resistance and the formation of Ire1 clusters. In Video 1, a structural model of the TMH region of Ire1 highlights membrane thinning and water penetration into the bilayer. Video 2 shows dynamics of the TMH region of Ire1 dimers over a period of 600 ns. Table S1 lists yeast strains used in this study. Table S2 lists plasmids used in this study.

## Data availability

All data discussed in the paper are included in this published article and in the online supplemental material. Additional materials including qPCR data, microscopy data, and the immunoblots contributing to the bar diagrams in Fig. 3 F; Fig. 5, B and D; and Fig. 6 F have been deposited to Mendeley Data (DOI: 10.17632/s52vt8spmc.1).

## Acknowledgments

We thank Kristina Halbleib for her important contributions during the early phase of the project and David Ron for critically reading the manuscript and helpful discussions. We thank Sebastian Schuck and Dimitrios Papagiannidis (ZMBH, Heidelberg University, Germany) for providing the pSS455 plasmid.

This work was supported by the Deutsche Forschungsgemeinschaft (SFB807 "Transport and Communication

across Biological Membranes" to R. Ernst and G. Hummer; SFB894 "Ca²⁺-Signals: Molecular Mechanisms and Integrative Functions" to R. Ernst). R. Ernst was supported by the Volkswagen Foundation (Life?, 93089). This project has received funding from the European Research Council under the European Union's Horizon 2020 research and innovation program (grant agreement no. 866011). R. Covino and G. Hummer were supported by the Max Planck Society and by the LOEWE CMMS program of the state of Hesse. R. Covino acknowledges the support of the Frankfurt Institute for Advanced Studies.

The authors declare no competing financial interests.

Author contributions: Conceptualization: R. Ernst and R. Covino; experimental design: K. Väth, J. Reinhard, C. Mattes, and R. Ernst; experiment performance: K. Väth, J. Reinhard, C. Mattes, and H. Stumpf; modeling, MD simulations, and rendering: R. Covino; microscopy and Segmentation: J. Reinhard; writing – original draft: K. Väth, R. Covino, and R. Ernst; writing – revised draft: K. Väth, R. Covino, J. Reinhard, C. Mattes, G. Hummer, and R. Ernst; figure design: K. Väth, J. Reinhard, C. Mattes, and R. Ernst; funding acquisition: R. Ernst and G. Hummer; supervision: R. Ernst, R. Covino, and G. Hummer.

Submitted: 12 November 2020

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

# Supplemental material

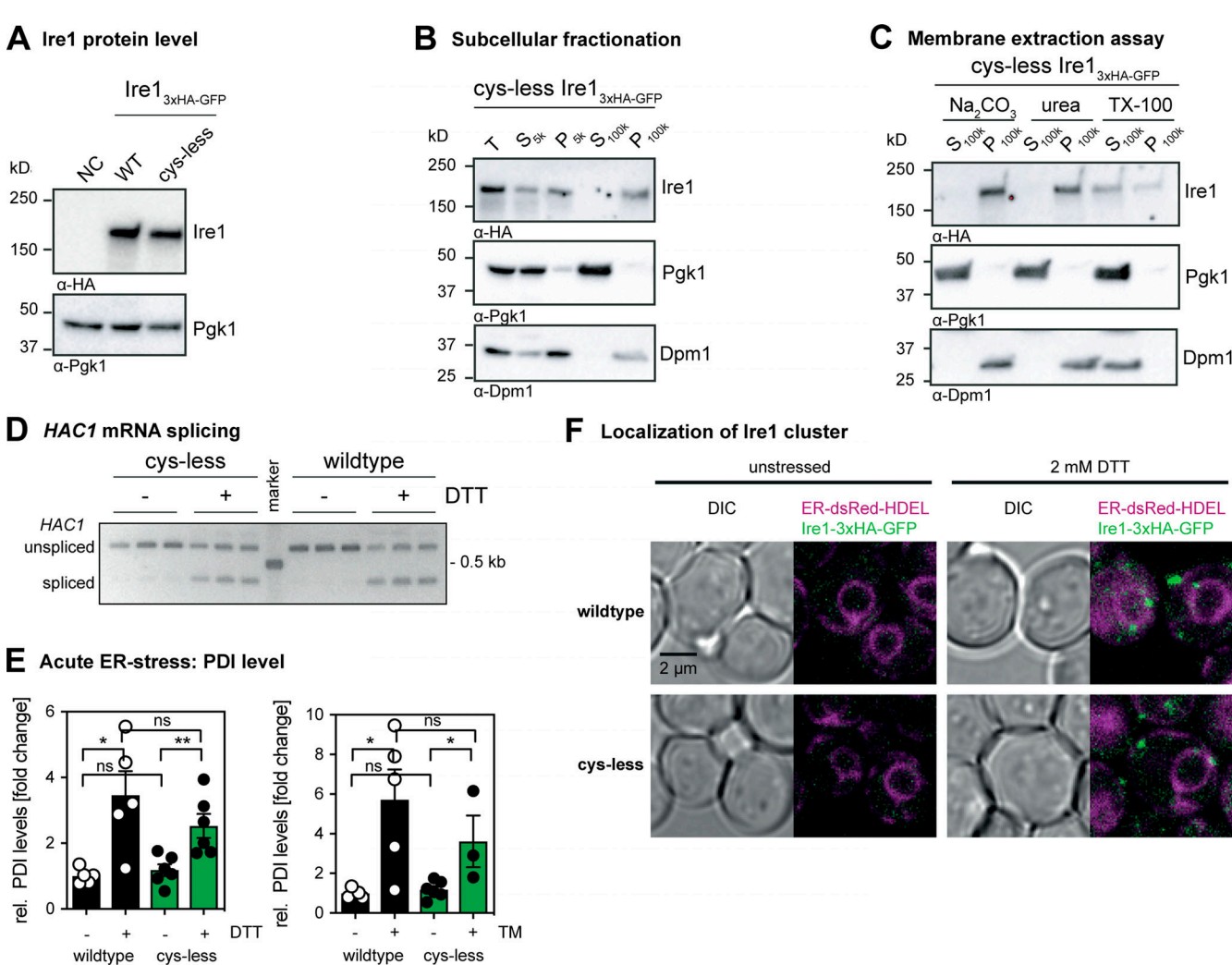

**Figure S1.** **Protein levels of cysteine-less Ire1 and characterization of its membrane association. (A)** Protein levels of cells expressing either $IRE1_{3xHA-GFP}$ WT or the cysteine-less (cys-less) variant. The isogenic WT strain BY4741 that does not express a HA-tagged variant of $IRE1$ was used as a negative control (NC). Stationary overnight cultures were used to inoculate a fresh culture in SCD complete to an $OD_{600}$ of 0.2 and cultivated until an $OD_{600}$ of 1 was reached. 0.1 OD equivalents of cell lysates were immunoblotted using anti-HA and anti-Pgk1 antibodies. **(B)** Subcellular fractionation of exponentially growing cells expressing cysteine-less $IRE1_{3xHA-GFP}$ by differential centrifugation at 5,000 ×g and 100,000 ×g. Stationary overnight cultures were used to inoculate a fresh culture in SCD complete to an $OD_{600}$ of 0.2 and cultivated until an $OD_{600}$ of 1 was reached. 80 $OD_{600}$ equivalents were harvested, lysed, and served as total input (T) for microsomal membrane preparation. The total lysate (T), and the individual supernatant (S) and pellet (P) fractions from centrifugation steps at 5,000 ×g (5k) and 100,000 ×g (100k) were analyzed separately by immunoblotting using anti-HA, anti-Pgk1, and anti-Dpm1 antibodies. 0.4 OD equivalents were loaded per lane. **(C)** Extraction assay of microsomes. Carbonate and urea extraction validate proper membrane integration of cysteine-less $IRE1_{3xHA-GFP}$ (cys-less). Samples of each step corresponding to 0.2 OD equivalents were analyzed by immunoblotting using anti-HA, anti-Pgk1, and anti-Dpm1 antibodies. **(D)** The indicated strains from a stationary culture were used to inoculate fresh culture in SCD to an $OD_{600}$ of 0.2. After cultivation at 30°C to an $OD_{600}$ of 0.7, cells were either left untreated or stressed with DTT (1 h, 2 mM, SCD). The level of the cDNA obtained from the spliced and unspliced HAC1 mRNA was amplified and separated by a 2% agarose gel. **(E)** PDI1 mRNA levels in acutely stressed cells normalized to the fold change of unstressed cells expressing $IRE1_{3xHA-GFP}$ WT. Exponentially growing cells of the indicated strains were used to inoculate fresh YPD media to an $OD_{600}$ of 0.2, cultivated in YPD, and acutely stressed with either 4 mM DTT (left) or 1.0 μg/ml TM (right) for 1 h. The relative level of PDI1 in these cells was analyzed by RT-qPCR and quantitated using the comparative ΔΔCT method using normalization to ACT1 levels. The data were normalized to the PDI1 level in unstressed cells carrying the $IRE1_{3xHA-GFP}$ WT construct. All error bars in this figure represent the mean ± SEM. Number of independent experiments: (left) -DTT: WT ($n$ = 6); -DTT: cysteine-less ($n$ = 6); +DTT: WT ($n$ = 5); cysteine-less ($n$ = 6); (right) -TM: WT ($n$ = 5); cysteine-less ($n$ = 6); +TM: WT ($n$ = 5); cysteine-less ($n$ = 3). Significance was tested by an unpaired, two-tailed Student's t test. *, $P < 0.05$; **, $P < 0.01$. Data distribution was assumed to be normal, but this was not formally tested. **(F)** Cells were cultivated from $OD_{600}$ of 0.2 to $OD_{600}$ of 0.7 in SCD medium and then either left untreated or stressed with 2 mM DTT for 1 h. Live cells were mounted on agar slides, and z-stacks were recorded using confocal microscopy. Images show the center plane of indicated channels. DIC, differential interference contrast; rel., relative; TX-100, Triton X-100.

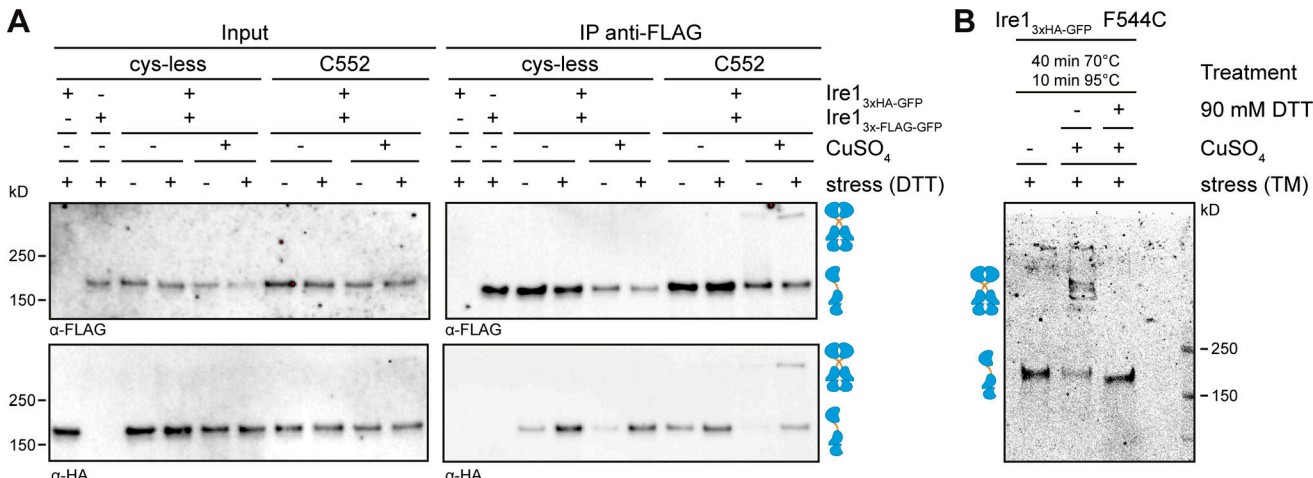

Figure S2. **Validation of a covalent, reversible cross-linking of Ire1 homodimers via disulfide bridges. (A)** A cross-linking experiment using $CuSO_4$ was performed with microsomes prepared from cells expressing a HA-tagged variant of Ire1 from endogenous locus ($IRE1_{3xHA-GFP}$) and a Flag-tagged variant ($IRE1_{3xFlag-GFP}$) from a CEN-based plasmid. A yeast culture in selective medium without leucine was inoculated to an $OD_{600}$ of 0.2 from a stationary overnight culture and cultivated at 30°C until an $OD_{600}$ of 0.7 was reached. The cells were either stressed with 2 mM DTT or left untreated and were further cultivated for 1 h. 80 $OD_{600}$ equivalents from these cultures were harvested by centrifugation. Microsomal membranes were isolated by differential centrifugation. Microsomes prepared from cells expressing only one of the two tagged variants of Ire1 served as controls. Both constructs contained a single cysteine in the TMH region at the position 552 (C552). After incubation of the microsomes with 10 mM $CuSO_4$ on ice for 5 min, the cross-linking reaction was stopped by the addition of NEM in a final concentration of 111 mM and EDTA in a final concentration of 50 mM. The microsomes were then solubilized using 2% Triton X-100 and subjected to an IP using anti-Flag beads. Both the input and IP samples were analyzed by immunoblotting using anti-Flag and anti-HA antibodies. **(B)** The reversibility of the cysteine-mediated cross-link was validated using the indicated F544C variant of Ire13xHA-GFP. The cross-link was induced by $CuSO_4$ in microsomes prepared from cells stressed with TM as described in Fig. 3. The cross-link was reverted by treating the sample with 90 mM DTT and incubating at 70° and 95° as indicated. The monomeric and dimeric species of Ire1 are indicated by symbols.

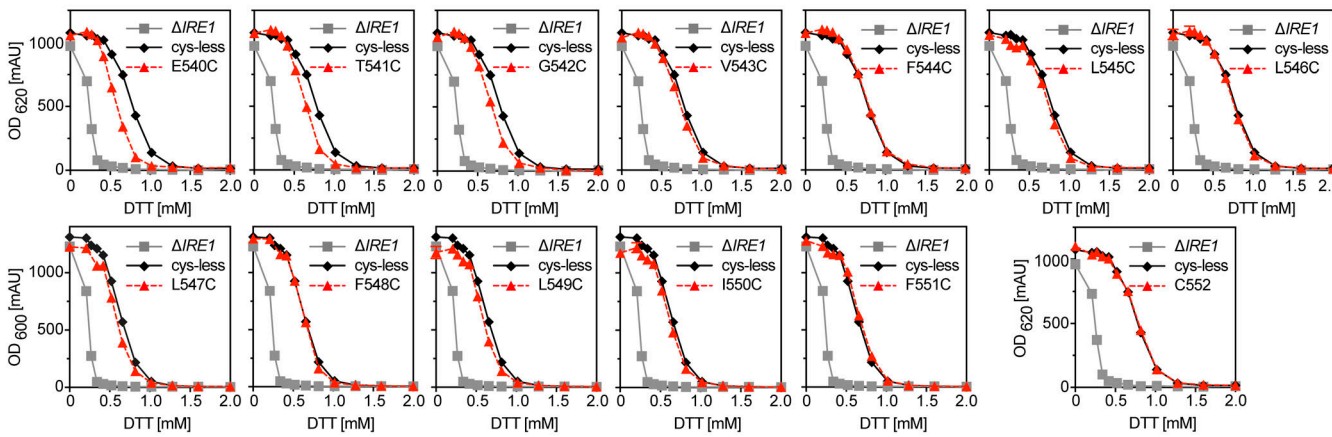

**A** Functionality of cysteine mutants

**C** Expression levels

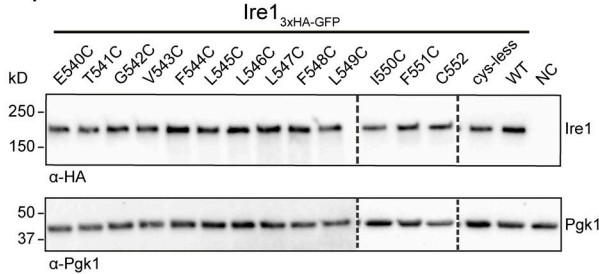

**E** Formation of Ire1_{3xHA-GFP} clusters

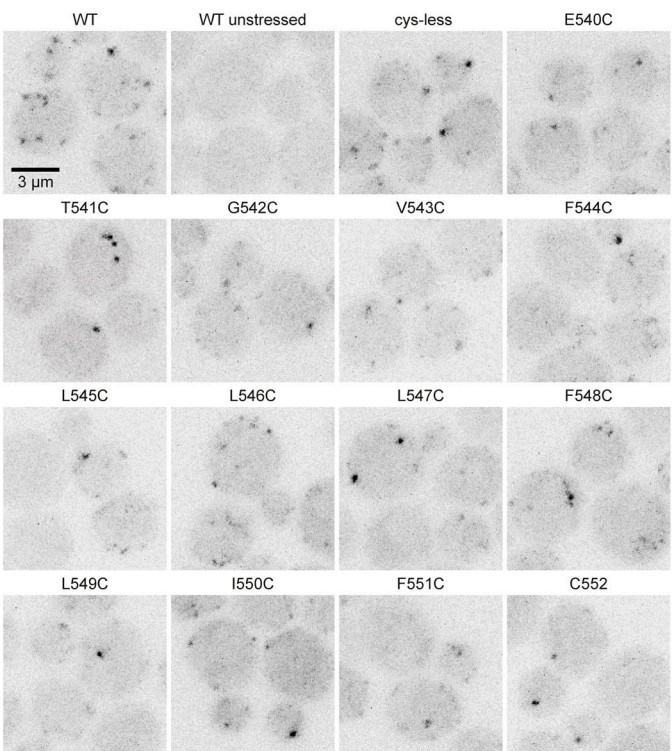

**B** *HAC1* mRNA splicing

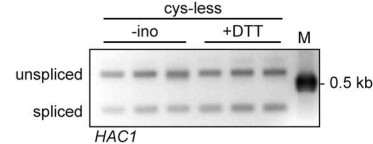

**D** Inositol depletion crosslinks

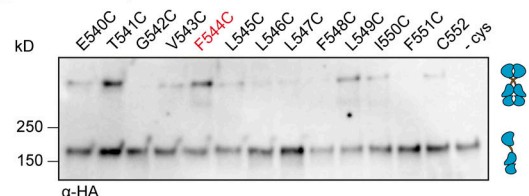

**F** Quantification of cells with Ire1_{3xHA-GFP} clusters

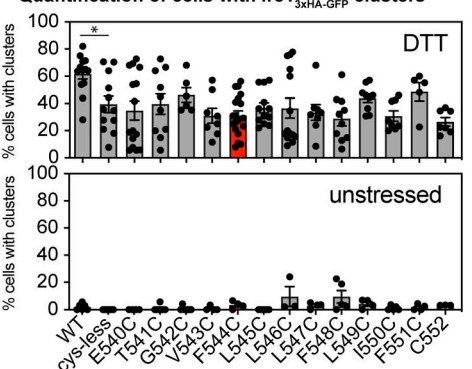

**G** Properties of DTT-induced Ire1_{3xHA-GFP} clusters

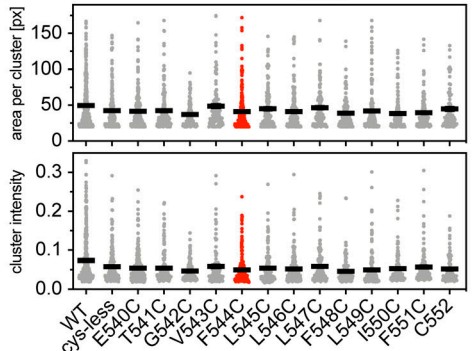

Figure S3.   **Functionality of cysteine mutants and their cross-linking potential in lipid bilayer stress conditions. (A)** The resistance to ER stress was investigated for the indicated yeast strains. Stationary overnight cultures of the indicated yeast strains were used to inoculate a fresh culture in full or minimal media to an $OD_{600}$ of 0.2. After cultivation for 5–7 h at 30°C, the cells were diluted with fresh minimal media to an $OD_{600}$ of 0.1. Cells were cultivated for 18 h at 30°C and stressed with DTT. The density of the resulting culture was determined using the $OD_{620}$ or $OD_{600}$. The error bars represent the mean ± SEM of at least two independent clones. Number of experiments to yield $OD_{620}$: $\Delta IRE1$ ($n$ = 20; identical to Fig. 1 B); cysteine-less ($n$ = 12; identical to Fig. 1 B); E540C to L546C ($n$ = 6; triplicates from two individual colonies); and C552 ($n$ = 12; triplicates from two individual colonies). Number of experiments to yield $OD_{600}$: $\Delta IRE1$ ($n$ = 10; replicates from three individual colonies); cysteine-less ($n$ = 22; replicates from four individual colonies); L547C, L549C, I550C, and F551C ($n$ = 6; triplicates from two individual colonies); and F548C ($n$ = 5; replicates from two individual colonies). **(B)** The indicated strains were cultivated and treated as described in Fig. 3, C and D, using conditions of proteotoxic (+DD) and lipid bilayer stress (-ino), respectively. The level of the cDNA obtained from the spliced and unspliced $HAC1$ mRNA was amplified and separated by a 2% agarose gel using a DNA ladder as size marker (M). **(C)** Protein levels of cells expressing different $IRE1_{3xHA-GFP}$ variants. The lysates of exponentially growing cells were immunoblotted using anti-HA and anti-Pgk1 antibodies. **(D)** Cross-linking of single cysteine variants of Ire1 in microsomes derived from cells grown in lipid bilayer stress conditions. Exponentially growing cells in SCD complete media were washed and used to inoculate a fresh culture in SCD complete to an $OD_{600}$ of 0.5. To induce lipid bilayer stress, the cells were washed and then cultivated in pre-warmed SCD complete without inositol medium for 3 h. 80 OD equivalents were harvested and used for microsomal membrane preparation. $CuSO_4$-induced cross-link was performed by incubating 8 µl of microsomes with 2 µl of 50 mM $CuSO_4$ for 5 min on ice. After stopping the reaction with NEM and EDTA, samples were subjected to SDS-PAGE with a subsequent immunoblotting with anti-HA antibody. Notably, all samples subjected to SDS-PAGE underwent a cross-linking procedure. Differences in specific and unspecific cross-linking may falsely suggest differences in loading. **(E)** Cells were cultivated to the early exponential phase in SCD and either treated with 2 mM DTT for 1 h or left untreated. Representative images (maximum projections of z-stacks) recorded by confocal microscopy. **(F)** The percentage of cluster-containing cells was determined for stressed (2 mM DTT, 1 h) and unstressed cells using a custom-made CellProfiler pipeline. The percentage of cluster-containing cells with the cysteine-less variant of Ire1 is not significantly different from any of the cells with single-cysteine variants. All data for the WT and cysteine-less variant are identical to the data from Fig. 1 C and plotted as a reference. For stressed cells: E540C ($n$ = 12 fields of view/359 cells); T541C ($n$ = 10/206); G542C ($n$ = 6/124); V543C ($n$ = 8/223); F544C ($n$ = 19/439); L545C ($n$ = 12/181); L546C ($n$ = 14/399); L547C ($n$ = 8/203); F548C ($n$ = 10/279); L549C ($n$ = 9/212); I550C ($n$ = 8/232); F551C ($n$ = 5/152); and C552 ($n$ = 7/188). For unstressed cells: E540C ($n$ = 4 fields of view/130 cells); T541C ($n$ = 7/153); G542C ($n$ = 7/188); V543C ($n$ = 4/121); F544C ($n$ = 4/108); L545C ($n$ = 5/101); L546C ($n$ = 3/111); L547C ($n$ = 4/106); F548C ($n$ = 5/143); L549C ($n$ = 4/103); I550C ($n$ = 5/165); F551C ($n$ = 4/108); and C552 ($n$ = 3/90). **(G)** The area of the detected clusters in the z-projection was determined and plotted. It was 49.9 px for the WT variant, 42.6 px for the cysteine-less variant, and ranged from a minimum of 37.2 px (G542C) to maximum of 48.9 px (V543C) for the single-cysteine variants. The integrated fluorescent intensity of detected clusters was 0.074 (arbitrary units) for the WT, 0.059 for the cysteine-less construct, and ranged from a minimum of 0.046 for the F548C variant to a maximum of 0.059 for the L547C variant. Significance was tested using a Kolmogorov–Smirnov test (*, P < 0.05). The segmented and analyzed number of clusters for each construct was as follows: WT: $n$ = 395 (raw data in Fig. 1 D); cysteine-less: $n$ = 211 (raw data in Fig. 1 D); E540C: $n$ = 215; T541C: $n$ = 158; G542C: $n$ = 95; V543C: $n$ = 101; F544C: $n$ = 224; L545C: $n$ = 131; L546C: $n$ = 191; L547: $n$ = 121; F548C: $n$ = 127; L549C: $n$ = 168; I550C: $n$ = 121; F551C: $n$ = 113; and C552: $n$ = 75. px, pixels.

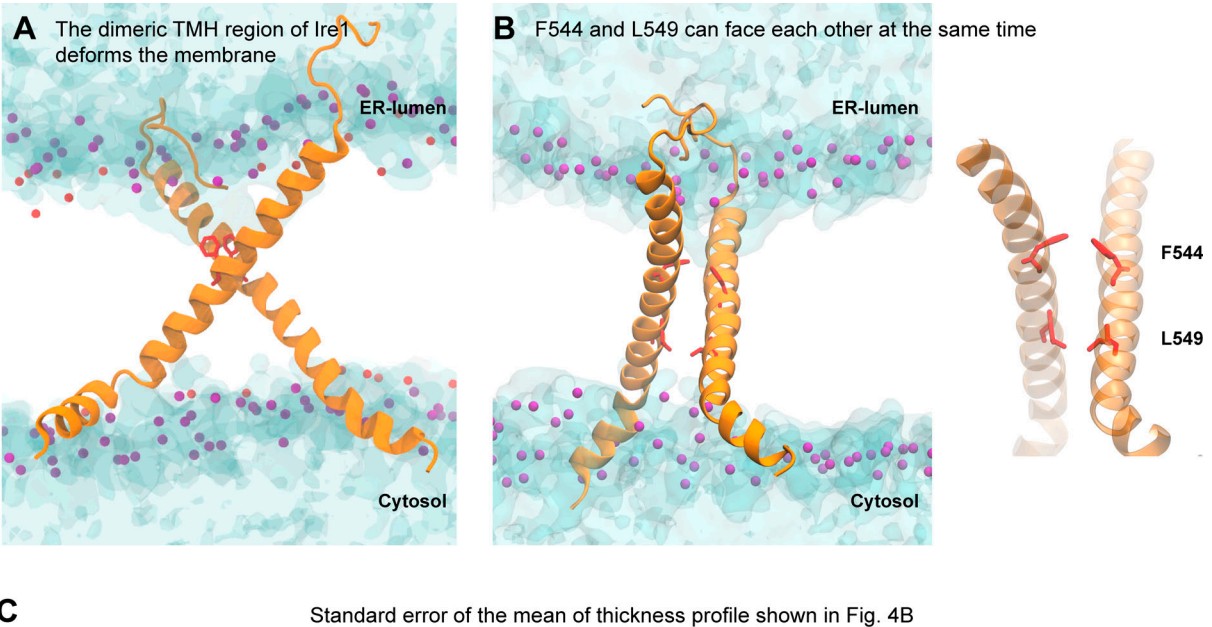

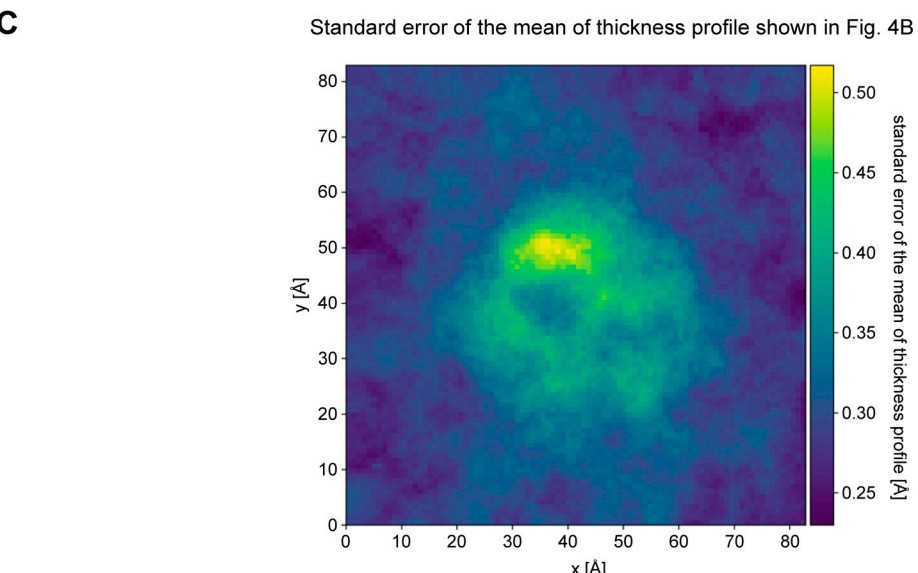

Figure S4. **The dimeric TMH region of Ire1 deforms the membrane. (A)** Membrane deformation by the modeled, dimeric TMH region of Ire1. Water is shown in blue tones with a transparent surface representation. The phosphate moieties of POPC are shown as purple beads. **(B)** Configuration of a model TMH dimer obtained from atomistic molecular dynamics simulations. Protomers are shown as an orange ribbon, with the residues F544 and L549 highlighted in red. The phosphate moieties of POPC are shown as purple beads. The hydroxyl groups of cholesterol molecules are shown as red beads. Water is shown with a transparent surface representation. Right: Lipid and water are not shown for clarity. **(C)** The SEM of the thickness profile represented in Fig. 4 B. The thickness fluctuations in the proximity of the TMH dimer (not shown, centered in the middle of the box) give rise to a locally increased SEM of the thickness profile, but is much lower than the actual degree of membrane deformation as plotted in Fig. 4 B.

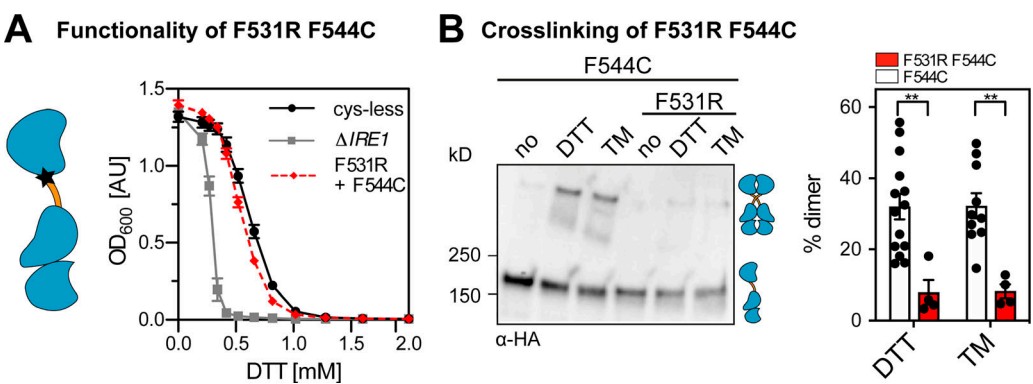

**Figure S5.   A mutation of the AH affects Ire1 function and cross-linking propensity. (A)** The ER-stress resistance of cells expressing the AH-disrupting F531R variant of *IRE1*₃ₓHA-GFP containing the F544C single-cysteine was scored using an ER-stress resistance assay. The indicated cells were cultivated and treated as in Fig. 5, A and C. Data for Δ*IRE1* and the cysteine-less construct are identical to data in Fig. 5 A. F531R/F544C (*n* = 12 independent experiments from two individual colonies). **(B)** The impact of the AH-disrupting F531R mutation of Ire1 on the degree of cross-linking via the single-cysteine variant F544C was determined using the microsome-based cross-linking assay. Cells were cultivated and further treated as described in Fig. 5, B and D. The data are represented as the mean ± SEM. All data related to the F544C variant are identical to the data in Fig. 3 F. F531R/F544C (*n* = 4 independent experiments from two individual colonies). Significance was tested by an unpaired, two-tailed Student's t test. **, P < 0.01.

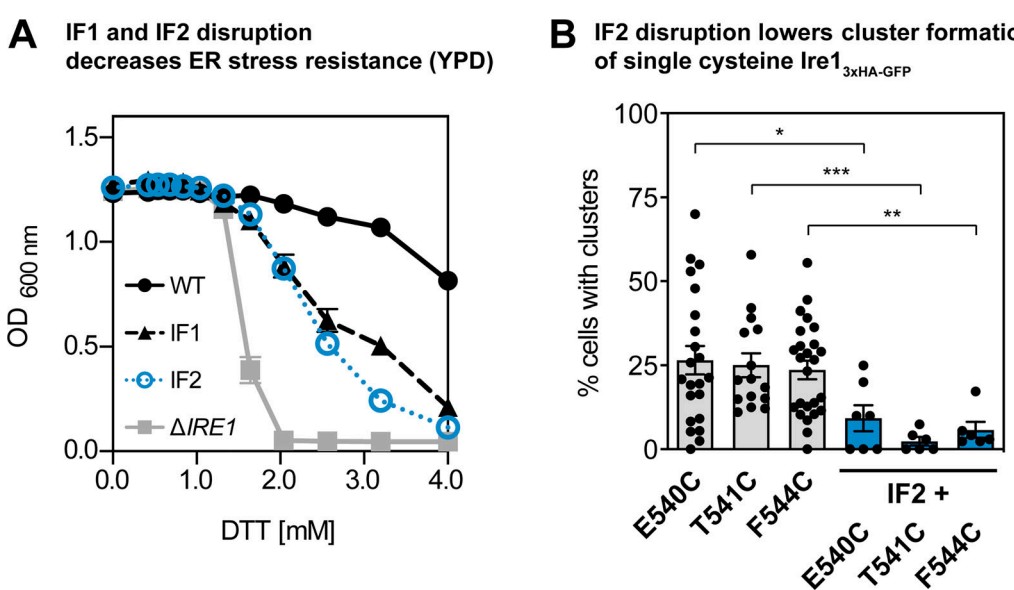

**Figure S6.   Disrupting ER-luminal interfaces for dimerization (IF1) and oligomerization (IF2) of Ire1 impairs cellular ER-stress resistance and the formation of Ire1 clusters. (A)** The resistance to DTT of cells expressing the IF1 (T226A/F247A) or IF2 (W426A) variants of *IRE1* with all native cysteines was analyzed in rich medium. The indicated cells were cultivated and treated as in Fig. 5, A and C. Data for Δ*IRE1* (gray squares) and *IRE1* WT knock-in construct with all native cysteines (black circles) are plotted as a reference. WT (*n* = 6) and IF2 (*n* = 12) data were from two individual colonies. IF1 (*n* = 4) and Δ*IRE1* (*n* = 6) data are from a single colony. **(B)** The percentage of cluster-containing cells was determined for the indicated strains cultivated in SCD medium and stressed with 2 mM DTT for 1 h. We re-used the raw microscopic data from Fig. S3 F for E540C (*n* = 15 fields of view/374 cells), T541C (*n* = 9/181), and F544C (*n* = 19/440), re-analyzed them as described in the Materials and methods, and pooled those with additional data for E540C (*n* = 7/281), T541 (*n* = 6/172), and F544C (*n* = 6/213). We also studied the clustering of E540C/IF2 (*n* = 7/98), T541C/IF2 (*n* = 6/150), and F544C/IF2 (*n* = 6/208). Microscopic images were analyzed using a customized CellProfiler pipeline. The percentage of cluster-containing cells with single cysteine variants of Ire1 is significantly different from any of the cells where the ER-luminal IF2 was disrupted by mutation (W426A). The data are represented as the mean ± SEM. Significance was tested using a Kolmogorov–Smirnov test (*, P < 0.05; **, P < 0.01; ***, P < 0.001).

Video 1.    **A structural model of the TMH region of Ire1 highlights membrane thinning and water penetration into the bilayer.** The two protomers of Ire1 TMH region are shown as orange ribbons. The residue corresponding to F544 is highlighted in red. The phosphate moieties of the lipid headgroups are shown as red/purple spheres. Water is indicated as shaded region to highlight membrane thinning. Lipid acyl chains are not shown for clarity.

Video 2.    **Dynamics of the TMH region of Ire1 dimers over a period of 600 ns.** The two TMH regions are shown as orange ribbons. The residue corresponding to F544 is highlighted in red, while residues T541 to F551 are shown in blue. The phosphate moieties of the lipid headgroups are shown as purple spheres. Water, ions, and lipid acyl chains are omitted for clarity.

**Tables S1 and S2 are provided online as separate files. Table S1 shows yeast strains used in this study. Table S2 shows plasmids used in this study.**

