## [Peer Review File · The Journal of Cell Biology]

Cysteine crosslinking in native membranes establishes the transmembrane architecture of Ire1

Kristina V  th, Carsten Mattes, John Reinhard, Roberto Covino, Heike Stumpf, Gerhard Hummer, and Robert Ernst

Corresponding Author(s): Robert Ernst, Saarland University

Review Timeline:

Submission Date:	2020-11-12
Editorial Decision:	2020-12-21
Revision Received:	2021-04-28
Editorial Decision:	2021-05-13
Revision Received:	2021-05-17

Monitoring Editor: Jodi Nunnari

Scientific Editor: Andrea Marat

Transaction Report:

DOI: <https://doi.org/10.1083/jcb.202011078>

December 21, 2020

Re: JCB manuscript #202011078

Prof. Robert Ernst
Saarland University
PZMS and Medical Biochemistry and Molecular Biology
Kirrberger Str. 100 Geb. 61.4
Homburg 66421
Germany

Dear Prof. Ernst,

Thank you for submitting your manuscript entitled "Systematic crosslinking in native membranes establishes the transmembrane architecture in Ire1 clusters". Your manuscript has been assessed by expert reviewers, whose comments are appended below. As you will read, the reviewers expressed interest in this work, however, significant concerns were also raised, which preclude publication of the current version of the manuscript in JCB.

All three reviewers raised important concerns, which are partially overlapping. In our opinion all of the points raised are valid and thus must be addressed in their entirety, which will include new experimental data where requested.

Please let us know if you are able to address the major issues outlined above and wish to submit a revised manuscript to JCB. Note that a substantial amount of additional experimental data likely would be needed to satisfactorily address the concerns of the reviewers. As you may know, the typical timeframe for revisions is three to four months. However, we at JCB realize that the implementation of social distancing and shelter in place measures that limit spread of COVID-19 also pose challenges to scientific researchers. Lab closures especially are preventing scientists from conducting experiments to further their research. Therefore, JCB has waived the revision time limit. We recommend that you reach out to the editors to decide on an appropriate time frame for resubmission. Please note that papers are generally considered through only one revision cycle, so any revised manuscript will likely be either accepted or rejected.

If you choose to revise and resubmit your manuscript, please also attend to the following editorial points. Please direct any editorial questions to the journal office.

GENERAL GUIDELINES:

Text limits: Character count is < 40,000, not including spaces. Count includes title page, abstract, introduction, results, discussion, acknowledgments, and figure legends. Count does not include materials and methods, references, tables, or supplemental legends.

Figures: Your manuscript may have up to 10 main text figures. To avoid delays in production, figures must be prepared according to the policies outlined in our Instructions to Authors, under Data Presentation, <https://jcb.rupress.org/site/misc/ifora.xhtml>. All figures in accepted manuscripts will be screened prior to publication.

IMPORTANT: It is JCB policy that if requested, original data images must be made available. Failure to provide original images upon request will result in unavoidable delays in publication. Please ensure that you have access to all original microscopy and blot data images before submitting your revision.

Supplemental information: There are strict limits on the allowable amount of supplemental data. Your manuscript may have up to 5 supplemental figures. Up to 10 supplemental videos or flash animations are allowed. A summary of all supplemental material should appear at the end of the Materials and methods section.

If you choose to resubmit, please include a cover letter addressing the reviewers' comments point by point. Please also highlight all changes in the text of the manuscript.

Regardless of how you choose to proceed, we hope that the comments below will prove constructive as your work progresses. We would be happy to discuss them further once you've had a chance to consider the points raised. You can contact the journal office with any questions, cellbio@rockefeller.edu or call (212) 327-8588.

Thank you for thinking of JCB as an appropriate place to publish your work.

Sincerely,

Jodi Nunnari, Ph.D.
Editor-in-Chief

Andrea L. Marat, Ph.D.
Senior Scientific Editor

Journal of Cell Biology

Reviewer #1 (Comments to the Authors (Required)):

In this manuscript, the authors sought to determine the structural and biochemical aspects of Ire1 activation upon endoplasmic reticulum (ER) stress. Using a systematic cysteine-crosslinking assay at the transmembrane helix (TMH) region of Ire1, the authors identified amino acid F544 that stabilized Ire1 dimers when substituted to a cysteine residue. Based on experimental data and simulations, the authors hypothesized that Ire1 TMH region forms an 'X'-shaped structure where F544 is at the interface between two Ire1 TMH protomers. Lastly, using an Ire1 variant, IF2, which disrupts Ire1 oligomeric association through the luminal domain, the authors revealed the significance of several amino acid residues at the TMH region in the dimerization/oligomerization process of Ire1.

Overall, the experiments were well designed, executed, and included detailed analysis. The conclusions of the manuscript offer important perspectives on the mechanism of Ire1 to sense ER stress during proteotoxic stress and lipid bilayer stress (LBS). However, there are several weaknesses that should be addressed to strengthen the authors' hypothesis. I recommend the authors to perform additional experiments, to consider alternative explanations to rationalize their data, and/or to tone down some of their conclusions.

Major points to address

(1) The fluorescence signal to monitor Ire1 clusters in Fig. 1D is weak. Usually, a weak signal of unstressed Ire1 overlapping with the ER network should be observed. Unstressed Ire1 WT and Cys-less in Fig. 1D exhibited no signal. It would have been a good control to show that the distribution of Ire1 Cys-less at the ER is similar to WT in unstressed cells especially that Fig. 1D lack any ER marker. Currently, we need to have faith that Ire1 clusters at the ER.

(2) Through the manuscript, HAC1 mRNA splicing is reported as relative HAC1 splicing to the stress condition. Therefore, it is impossible to compare the splicing efficiency between dithiothreitol (DTT), tunicamycin (Tm), and inositol depletion for the Ire1 variants. Someone can only compare HAC1 splicing efficiency between Ire1 variants within the same stress condition. As HAC1 splicing is normalized to ACT1 mRNA levels instead of being normalized to total HAC1 mRNA levels (total included spliced and unspliced HAC1 mRNA), it is impossible to appreciate the differences in HAC1 mRNA expression levels and splicing efficiencies across the different conditions. In other words, HAC1 splicing could be reported in a more transparent way. Alternative assays to demonstrate Ire1 activation would also strengthen their conclusions.

(3) In the first two figures, the authors characterized the functionality and dimerization of Ire1 Cys-less and F544C variants during proteotoxic stress (DTT and Tm). The authors concluded that these variants are activated by ER stress and that they form clusters as well as Ire1 WT (Fig. S3D). Based on these authors' conclusion, someone should expect most of Ire1, if not all, to form dimers in the crosslinking experiments of Ire1 as well as being able to immunoprecipitate the majority of Ire1 Cys variants. However, the fractions of dimerized Ire1 are rather small. How representative are the in vitro crosslinking experiments perform on ice to the actual Ire1 dimeric states in vivo? Does it mean that only the fractions of dimerized Ire1 are activated and thus the soles contributors to HAC1 slicing?

(4) I would like the authors to comments on the reasons that could explain the different of Ire1 F544C or Cys-less expression levels in unstressed conditions. It might be due to loading variation, but a loading control is not included in Fig. 2C. Alternatively, should we assume that each lane contains between 6.4 and 9.6 µg microsomal proteins (based on 1 {plus minus} 0.2 mg/ml protein)?

(5) In Figure 3 and S3, it is apparent that mutations in Ire1 AH region will increase sensitivity to ER stress in cells, but not for TMH variants. Yet, it is puzzling that at a comparable level of HAC1 mRNA splicing (HAC1s), AH-cysteine-variants (E540C, T541C, and G542C) displayed sensitivity against ER stress despite having comparable levels of HAC1s relative to WT.

(6) The authors stated "Because F544C, the best-crosslinking residue, and L549C seemingly lie on opposing sites of Ire1's TMH as judged from a helical wheel representation (Fig. 3B), this raises the question if the corresponding residues in the native TMH can face each other at a low distance and at the same time" (page 6). However, I find this argumentation rather weak as with DTT and Tm, variants E540C, T541C, I550C, and C552 exhibit similar dimerization when compared to L549C levels.

(7) The authors also stated that they have performed an "unbiased MD [molecular dynamics] simulation" (page 6). In my view they have been biased as their MD simulation is based on yet another MD simulation from their previous publication (PMID 28689662). They used their previous MD simulation model to generate an Ire1 dimeric MD simulation. In my view their proposed "X-

shaped" dimer is very speculative, and it lacks experimental evidence to support it. I can appreciate that it is a difficult weakness to address. As this model is central to their conclusions, additional experiments should be performed in vitro and in vivo. For example, the authors could get NMR structures of Ire1 TMH in liposomes. Otherwise, I strongly suggest the authors to tone down their conclusions.

(8) In light of comment # 7 above, I disagree with the authors statement "Our crosslinking approach indicates that the TMH residue F544 is part of a small interface between Ire1 protomers, which might stabilize the unusual X-shaped transmembrane configuration of Ire1. Interestingly, an aromatic residue in the TMH region of the mammalian IRE1 α (W547) was reported to stabilize the oligomeric state of IRE1 α (Cho et al., 2019). Hence, we embarked on testing a similar role for F544 in Ire1 from baker's yeast". The positioning of mammalian W547 within the ER membrane is very different and should be closer to the cytosol when compared to yeast F544.

(9) The authors hypothesized that the stability of Ire1 dimers is likely dependent on the biochemical properties of AH rather than that of TMH. However, the increased sensitivity toward ER stress and the weakened crosslinking propensity observed in Ire1 F531R variant (Figure 5C,D) may be a result of destabilization of the luminal domain of Ire1 per se. Moreover, Ire1 Cys-less mutant and the other cysteine mutants, Ire1 F531R has a lower propensity to form clusters (reduction of ~50% compared with WT with 2 mM DTT) and is less sensitive to proteotoxic stress (~45% relative spliced HAC1 mRNA compared to WT with 4 mM DTT).

(10) In Figure 6D, the authors show that Ire1 T541C/IF2 variant forms CuSO₄-mediated dimers while Ire1 T541C/IF2 variant is sensitive to ER stress in Figure 6C. UPR activation reporter assays may be good indications to address the discrepancy between Ire1 dimerization and sensitivity to ER stress.

(11) The authors concluded that the different forms of ER stress (DTT, Tm, and inositol depletion) converge in a single, signaling-active conformation of Ire1. The crosslinking of Ire1 variants have been tested during inositol depletion (LBS) in Fig. S3C. However, the authors only focus on proteotoxic stress for the remaining experiments (Fig. 5 and 6). The authors would need to systematically performs the experiments of Fig. 5 and 6 during LBS to support their conclusion. I am wondering why the expression of some of the Ire1 variants are much lower compared to others during LBS (Fig. S3C). Are these destabilized by LBS?

Minor points to address (I refer to the page numbers of the generated PDF and the authors didn't make the effort to number the pages nor the lines of their manuscript)

(1) Define TMH (page 3)

(2) I find the labelling of amino acid numbers around AH and TMH confusing in Figure 1A. The authors might want to align the aa numbers and the domains or to indicate them clearly with a line.

(3) Page 5, "We conclude that CuSO₄ can catalyze the formation disulfide bridges between two neighboring Ire1 copies" should be replaced by "We conclude that CuSO₄ can catalyze the formation of disulfide bridges between two neighboring Ire1 copies"

Reviewer #2 (Comments to the Authors (Required)):

This study focuses on the architecture and function of transmembrane helix (TMH) and amphipathic helix (AH) of the yeast ER stress sensor Ire1. The authors employed a crosslinking-based strategy to pinpoint the contacts between transmembrane and amphipathic helices within Ire1 oligomers. To this end, they engineered a functional cysteine-less mutant of Ire1 and systematically replaced residues in the TMH and AH with cysteines. Contacts between helices were then identified via CuSO₄-mediated crosslinking in isolated ER microsomes. It is known that Ire1 signals through dimerization and/or high-order oligomerization, but the TMH conformation in dimeric and oligomeric Ire1 assemblies formed by different activators are poorly understood.

This approach revealed a consistent profile of crosslinked residues that are in close contact in response to IRE1 activation. Out of the tested interface residues, the F544C mutation resulted in the highest level of crosslinked dimers. Therefore, the authors conclude F544 is the main point of contact between the TM helices of an Ire1 dimer. When crosslinking was tested under proteotoxic stress (+DTT or Tunicamycin) or lipid stress (inositol depletion), crosslinking profiles were similar, suggesting that both types of stresses result in similar oligomeric arrangements of the TMH. Combining the crosslinking and functional studies together with simulation results, the authors conclude that the active form of Ire1 has X-shaped TMH conformation using F544 as the main interface.

The authors also studied the impacts of AH, IF1, and IF2 mutations on Ire1 TMH crosslinking propensity and functions. Consistent with their prior studies, disrupting the AH can lead to less crosslinked products and more stress sensitive phenotype. IF1 and IF2 (interfaces 1 and 2) were previously shown to play key roles in dimerization and oligomerization of Ire1, respectively. By disrupting IF2 and leaving IF1 intact, the authors are able to distinguish the TMH contacts are dimer-specific or oligomer-specific. They identified that the F544-F544' and T541-T541' contacts are most likely within dimers, whereas the E540-E540' contact is across dimers.

To summarize, this is a strong paper that adds important details to our understanding of the structural organization of Ire1 oligomers. It clearly reveals several residues at the interface between adjacent Ire1 transmembrane helices and provides a plausible model of the relative arrangement of the helices in Ire1 oligomers. The paper is well written with clear descriptions of the experimental approaches and an easily understandable discussion of the results.

Before publication, please consider and where appropriate address the following issues:

Major points:

1. How efficient is CuSO₄, an ionic water-soluble compound, at catalyzing cysteine crosslinking inside the lipid bilayer? Is there any published literature assessing the efficiency of CuSO₄ or other oxidizing reagents in the membrane context? It is important to reference and address if the findings can be significantly biased by the relative "depth" of each residue inside the membrane.
2. Can the authors comment on why crosslinking was performed on ice and not closer to yeast growth temperature and MD simulation temperature? How does this temperature difference affect membrane mobility and protein conformational dynamics?
3. The statement 'Ire1 does not seem to custom-tailor the signaling output to the specific input signals' is not supported by the data in this manuscript. Exhibiting a similar crosslinking profile in one specific region of the protein does not necessarily translate to identical downstream results. There could be a number of additional factors for tuning the outcome of downstream signaling (later discussed in manuscript), such as the relative abundance of dimers and oligomers, association of additional proteins, or targeting of the Hac1 mRNA to active Ire1.

4. Fig S3C. The gel shown seems to be an anomaly rather than a representative gel quantified in Fig. 3F (crosslinking under inositol depletion). Please show a more representative gel or address the high crosslinking for T541C.
5. Fig 6C. The authors do not explain clearly why F544C/IF2 mutant can partially rescue the stress sensitivity resulting from the IF2 mutation whereas other double mutants were more sensitive. Is that because F544C can lock the dimer in a conformation that can enhance Ire1 activity? Please discuss/provide a plausible explanation.
6. Fig 6D, last panel. Why do F544C/IF2 and F544C alone mutants have different crosslinking levels with tunicamycin treatment but similar level with DTT treatment? Is this result still consistent with the authors' conclusion that TMH of Ire1 has similar structural output to both proteotoxic and lipid stress?
7. The interpretation and presentation of simulation data can be greatly improved. What is the population of the X-shaped conformation that was shown in Figure 4 in 600 ns simulations? Why is the simulation length of 600 ns chosen? The authors may also consider calculating the free energy of different conformations throughout the 600 ns and map where the X-shaped conformation is in the energy diagram. Does the X-shaped conformation represent the minimum energy state?
8. The authors postulate that some of the residues in the transmembrane helices are important for intra-dimer contacts and others are important for inter-dimer contacts. One potential way to further validate this exciting notion would be to generate double-cysteine mutants, where one each of the intra- and inter- dimer residues are mutated to cysteines. If these residues do in fact bridge distinct interfaces, one would then expect to see the appearance of higher-order oligomeric species on the crosslinking gel (unless some technical hurdles impede this experiment; these hurdles could be addressed in the manuscript).

Minor points:

1. Typographical error in Figure 1. It should be 'amphipathic helix' not 'ampipathic helix'.
2. In the imaging experiments, the outline of the ER is not visible at all in the no-stress conditions. In published works, yeast Ire1 signal is clearly present in the ER network without stress, and coalesces into foci when stress is added. This seems to be due to sensitivity or expression level. Please address this different appearance in manuscript.
3. 'Upon dimerization of an unrelated single-pass transmembrane protein with a circular 'footprint', only a smaller portion of the deformed membrane can coalesce': are there any references to support this statement?
4. We would suggest having a better labeling on the cartoon representations in Figure 5A and 5C. Showing the ER lumen and cytosol sides will make it easier to read.
5. In term of clarity, showing the crosslinking gels only with globally applied contrast adjustment makes it difficult to see the much fainter crosslinked product relative to monomeric Ire1. In some instances, the figures could benefit from adding a cropped panel showing just the crosslinked product, contrast optimized for this gel region.
6. Page 7. "...polar residue F544C...alters he dimerization/oligomerization propensity": In future work, this interpretation can be experimentally tested by substituting F544 with non-polar residues.

Reviewer #3 (Comments to the Authors (Required)):

Recent studies of the unfolded protein response (UPR) signaling pathway have started to dissect the molecular mechanisms of how UPR-initiating components of the ER membrane recognize and respond to the increased demands of ER proteotoxic stress and lipid bilayer stress ("lipotoxic" stress). Specifically, for Ire1, one of the UPR inducers on the ER, the transmembrane (TM) domain

was thought to hold a key to responding to lipotoxic stress, as Ire1 retains the ability to respond to lipotoxic stress even with a truncated ER luminal domain, as shown by the Kimata (2011) and Ron (2013) groups. Interestingly, swapping the Ire1 TM domain with that from calnexin, an unrelated protein or TM domain sequence was scrambled did not affect Ire1's ability to respond to lipotoxic stress, revealing the lack of specific amino acid requirements for the Ire1 TM domain. Furthermore, the Ernst group published an intriguing finding that an amphipathic helix (AH) adjacent to and overlapping with the Ire1 TM domain plays a critical role in the response of yeast Ire1 to proteotoxic stress and lipids stress.

In this manuscript, Vaeth et al attempts to extend the Ernst study by dissecting how the AH and the IRE1 TM domain respond to lipid bilayer stress (induced by inositol depletion) in addition to proteotoxic stress. Ire1 contains multiple cysteine residues throughout the protein and thus, the authors generated a cysteine-less Ire1, making it possible to introduce a single cysteine within the AH and/or TM domains of Ire1 to perform CuSO₄-cysteine cross-linking experiments. The data in this manuscript represents ample work and should be published. Specifically, the generation of a cysteine-less Ire1 will likely provide a useful tool for future studies of Ire1. This approach allowed the authors to introduce single cysteine substitutions for CuSO₄-Cys crosslinking experiments. Together with simulation, the authors uncovered that F544 within the TM domain is a key residue for dimer formation in response to both proteotoxic and lipotoxic stress. Unfortunately, the conclusions presented here do not significantly extend the conclusions from the previous study. In this reviewer's opinion, papers published in Journal of Cell Biology should provide a significant advancement in our understanding of a specific topic(s) from prior knowledge. Therefore, in its current form, this manuscript would be more suitable for other journals.

In particular, the extents of 'Ire1 dimer' seem very low (most Ire1 remained monomer). The significance of this cross-linking is unclear for a few reasons. First, the assay was performed in microsomes prepared after ER stress induction of cells. During the preparation of microsomes, it is unclear if the ER membrane retains any potential changes imposed by the ER proteotoxic or lipotoxic stress. Furthermore, the effects of different amino acid changes were measured by (1) cell density test results across increasing concentrations of DTT (Fig S3A), (2) HAC1 splicing (Fig 3C), (3) dimer formation in microsomes (Fig 3E-F), and (4) Ire1 foci formation in vivo (Fig S3D-E). If a specific residue plays an important role in functional dimerization or oligomerization, these values should correlate with each other. However, among mutants displayed some Ire1 activities, these values did not show significant agreement in their assays; thus, the functional significance of Ire1 residues such as F544 is unclear.

Based on previously published studies, additional issues include the following:

(1) Crosslinking experiment: In addition to the issues described above, another issue is that even if the microsomes retain the ER stress-induced changes on the ER membrane and Ire1, it is unclear how the authors ensured that Ire1 crosslinking was not impacted by the Ire1-Ire1 interaction in trans between the different vesicles. (Ex. HAC1 mRNA splicing value for G542C is high (Figure 3C), but almost no Ire1 dimerization (Figure 3E-F)).

(2) HAC1 mRNA splicing: How was the HAC1 mRNA splicing calculation performed? What was calculated as 100% splicing? Was the HAC1 splicing in ER-stressed WT cells calculated as 100%? Was HAC1 splicing of mutant cells compared to that value? In addition, unstressed cells with WT Ire1 showed closed to 50% HAC1 mRNA splicing according to the presented values. While the basal HAC1 splicing levels can fluctuate to some degree between YPD and SCD, the splicing values differ significantly (in Fig 1C, the value seems to be <5%). The authors should show the actual

quantitative RT/CPR data.

(3) Ire1 foci in ER-stressed cells: Based on previous studies, Ire1 is thought to undergo dimerization and oligomerization in response to ER stress. The sizes of the Ire1 foci shown in this manuscript appear to be heterogeneous (for example, in Fig 1D), consistent with this idea. Why are only dimers considered? In addition, Ire1 oligomerization or dimerization status has been shown to switch the IRE1 RNase between HAC1 RNA splicing and regulated Ire1-dependent degradation (RIDD) of the ER-associated mRNA. The authors should measure the RIDD activity of the mutant Ire1. Moreover, Ire1 dimerization and oligomerization are known to be facilitated by the cytoplasmic portion of the Ire1 kinase and RNase domains, in addition to the Ire1 ER luminal and TM domains. Why were only the ER luminal domain interfaces, IF1 and 2, considered? Similar to the ER luminal domain, Ire1 interfaces through the cytoplasmic portion of Ire1.

(4) Subcellular fractionation: In Figure S3, differential centrifugation is used to fractionate cysteine-less Ire1 in order to confirm that it is similar to WT Ire1. It is unclear why significant levels of Ire1 TM protein are found in the low speed spin supernatant, suggesting that some Ire1 was solubilized. In contrast, re-probing with anti-Dpn1 antibody showed that the majority of Dpn1, another membrane protein, stayed in the low speed spin pellet fraction. Don't these results suggest that cysteine-less Ire1 acts differently than WT Ire1 in vivo? Similarly, results in Figure S2C suggest that cysteine-less Ire1 is not solubilized even after TX-100 treatment, while essentially 100% of Dpn1 is in the solubilized fraction upon TX-100 treatment. Taken together, these results show that property of Ire1 is altered in cysteine-less Ire1, or unfolded/aggregated. On the other hand, it is unlikely that this is the simple unfolding of cysteine-less Ire1 since levels of PDI upon DTT or TM treatments increase significantly at an extent similar to WT Ire1. Since single cysteine Ire1 mutants were generated on the idea that cysteine-less Ire1 behaved similar to WT Ire1, it needs to be clarified.

(5) Overall conclusions: The authors should discuss how their conclusions, specifically the significance of F544 within the TM domain of Ire1, fit in with previous results where a complete swap or scramble of Ire1 TM residues remains active. In this case, what happens to the proposed Ire1 TM conformation? Furthermore, in the discussion, the authors argue that "based on our crosslinking data, we suggest that the complex metabolic, transcriptional, and non-transcriptional adaptations to different forms of ER-stress do not reflect distinct functional modes of Ire1 (Fun and Thibault, 2020)." However, the current study does not address any contributions from the kinase or RNase domains of Ire1, which hold enzymatic activities of Ire1. Clearly, metabolic outcomes are largely impacted by the enzymatic activities of Ire1 and, thus, the authors should modify this statement. Finally, the authors state that "Based on a systematic cysteine crosslinking approach in native membranes and aided by MD simulations, we show that the neighboring TMHs in clusters of Ire1 are organized in an X-shaped configuration." Beyond the simulation and F554 TM residue involvement, additional configurations of the Ire1 TM domain may be possible.

Minor points:

(1) The authors stated "Nevertheless, seemingly subtle changes in the TMH region, however, can modulate the cellular resistance to ER stress when being placed. In light of these findings, we are convinced that studying UPR transducers at their endogenous level is absolutely required to understand their role in physiology." This has been described in many previous studies. The authors should reference earlier publications.

(2) In Figures 6C-6D, the authors probed the relationships between the F554 residue within the TM domain and IF2 residues.

(3) In Figure 5, the authors stated that a mutation within the AH, F531R, causes unfolding of AH and distorts the X and the placement of F544C for efficient crosslinking. However, Figure 5D shows that this mutation caused only a slight sensitivity change (15% to ~8%).

(4) pH concerns: Additionally, throughout the manuscript the authors describe experiments performed both in YPD and SGD. The effect of DTT is altered depending upon the pH of the media. For example, YPD is normally pH 7 unless adjusted, while SGD is pH 4-5. Has the pH of the media been adjusted? If so, this should be mentioned.

Reviewer #1 (Comments to the Authors (Required)):

In this manuscript, the authors sought to determine the structural and biochemical aspects of Ire1 activation upon endoplasmic reticulum (ER) stress. Using a systematic cysteine-crosslinking assay at the transmembrane helix (TMH) region of Ire1, the authors identified amino acid F544 that stabilized Ire1 dimers when substituted to a cysteine residue. Based on experimental data and simulations, the authors hypothesized that Ire1 TMH region forms an 'X'-shaped structure where F544 is at the interface between two Ire1 TMH protomers. Lastly, using an Ire1 variant, IF2, which disrupts Ire1 oligomeric association through the luminal domain, the authors revealed the significance of several amino acid residues at the TMH region in the dimerization/oligomerization process of Ire1.

Overall, the experiments were well designed, executed, and included detailed analysis. The conclusions of the manuscript offer important perspectives on the mechanism of Ire1 to sense ER stress during proteotoxic stress and lipid bilayer stress (LBS). However, there are several weaknesses that should be addressed to strengthen the authors' hypothesis. I recommend the authors to perform additional experiments, to consider alternative explanations to rationalize their data, and/or to tone down some of their conclusions.

We thank all three reviewers for their efforts and constructive criticism. We are convinced that the revision process has substantially improved the quality of our manuscript.

Major points to address

(1) The fluorescence signal to monitor Ire1 clusters in Fig. 1D is weak. Usually, a weak signal of unstressed Ire1 overlapping with the ER network should be observed. Unstressed Ire1 WT and Cys-less in Fig. 1D exhibited no signal. It would have been a good control to show that the distribution of Ire1 Cys-less at the ER is similar to WT in unstressed cells especially that Fig. 1D lack any ER marker. Currently, we need to have faith that Ire1 clusters at the ER.

We agree. It is important to have faith in the 'normal' localization of Ire1 clusters. We have provided additional data to the manuscript as described further below.

The main reason for the low fluorescence signal of GFP-tagged Ire1 in our hands is a lower expression level compared to previous attempts. Most previous studies on Ire1 localization, have expressed GFP/mCherry-tagged variants of *IRE1* from the endogenous promoter on *CEN*-based plasmids. It is known that *CEN*-based plasmids have a copy number of ~2 to ~4 in the BY4741 strain (PMID: 23107142), which - consequently- should result in a two- to four-fold overexpression of *IRE1* under these conditions. Because Ire1 is activated by oligomerization, we do our best to avoid such overexpression.

We have engineered a knock-in construct (Halbleib et al., 2017), which expresses *IRE1* with an internal GFP from the original genomic locus and using the endogenous promoter. Previously, we have shown that the expression level is ~2.7-fold lower than the level of *IRE1* expressed from *CEN*-based plasmids (Halbleib et al., 2017). Such mild degree of overexpression would have a substantial impact on the concentration of dimers and higher oligomers thereby leading to aberrantly high UPR signaling. In fact, the number and size of Ire1 clusters is significantly higher in cells expressing *IRE1* from *CEN*-based constructs compared to cells with the *IRE1* knock-in construct expressed from the endogenous, genomic locus:

Screenshot (from the Supplementary Material in Halbleib et al., 2017). GFP-tagged *IRE1* expressed from the endogenous promoter and a *CEN*-based plasmid is overexpressed, shows a significant degree of cell-to-cell variation of expression, and also an aberrantly high number and size of Ire1 clusters in stressed cells. The knock-in construct shows clustering in DTT-stressed cells, but the clusters are smaller and less frequent. The knock-in construct is not detectable above the autofluorescent background in unstressed cells.

In order to address the concern of the reviewer, we performed more experiments (new Figure S1F):

Using dsRed targeted to the ER-lumen and GFP-labeled Ire1 (WT and cysteine-less generated from the endogenous locus and promoter as used throughout our study), we provide supporting evidence by confocal microscopy that the clusters of Ire1 are associated with the ER, as expected. We see no difference in the localization between WT and cysteine-less Ire1.

(2) Through the manuscript, HAC1 mRNA splicing is reported as relative HAC1 splicing to the stress condition. Therefore, it is impossible to compare the splicing efficiency between dithiothreitol (DTT), tunicamycin (Tm), and inositol depletion for the Ire1 variants. Someone can only compare HAC1 splicing efficiency between Ire1 variants within the same stress condition. As HAC1 splicing is normalized to ACT1 mRNA levels instead of being normalized to total HAC1 mRNA levels (total included spliced and unspliced HAC1 mRNA), it is impossible to appreciate the differences in HAC1 mRNA expression levels and splicing efficiencies across the different conditions. In other words, HAC1 splicing could be reported in a more transparent way. Alternative assays to demonstrate Ire1 activation would also strengthen their conclusions.

The reviewer raises an important point regarding the normalization of qPCR data, which was also mentioned in point 2 by reviewer 3. In fact, we represent our data differently to what is common in the field. However, we do that for a good reason (see below). In the revised manuscript, we provide additional data on HAC1 mRNA splicing, which should help clarifying our point (new Figure S1D, S3B): We performed additional experiments (new Figure S1D, S3B) to show the degree of HAC1 splicing after amplifying both the spliced and the unspliced form HAC1 via RT-PCR. We also changed the labeling of the Y-axis in all relevant figures related to HAC1 splicing: Instead of “relative HAC1 splicing” we now label the Y-axis with “level of spliced HAC1 mRNA [%]”.

We are aware that it is most common in the field to plot the degree of HAC1 mRNA splicing as % spliced HAC1 mRNA in relation to the total HAC1 mRNA level. This is a reasonable way to normalize. It focusses on the *efficiency* of HAC1 mRNA splicing, but not on the *amount* of the spliced HAC1 mRNA generated in the respective cells. However, this (commonly used) way of normalization is sensitive to differences in the expression level of the HAC1 mRNA, while our approach is not. Normalizing the HAC1 mRNA to the ACT1 mRNA level (as we have done) focusses on the actual level of the spliced HAC1 mRNA, which -in our view- is reflecting the signaling output of the UPR better than the degree of HAC1 mRNA splicing. In fact, our way of normalization allows for a comparison between different experimental conditions.

Our additional data and rephrasing of the relevant sections/figures should help preventing any sort of confusion of the readers.

(3) In the first two figures, the authors characterized the functionality and dimerization of Ire1 Cys-less and F544C variants during proteotoxic stress (DTT and Tm). The authors concluded that these variants are activated by ER stress and that they form clusters as well as Ire1 WT (Fig. S3D). Based on these authors' conclusion, someone should expect most of Ire1, if not all, to form dimers in the crosslinking experiments of Ire1 as well as being able to immunoprecipitate the majority of Ire1 Cys variants. However, the fractions of dimerized Ire1 are rather small.

The reviewer raises important points. Why is the maximum degree of crosslinking relatively low?

In the revised manuscript, we directly address this concern: “The observed degree of crosslinking was somewhat low considering that up to 70-85% of Ire1 may reside in signaling-active clusters under conditions of ER stress (Aragón et al., 2009). For our crosslinking approach, however, we used a slightly milder condition to induce ER stress (2 mM DTT instead of 10 mM) and performed all experiments with an IRE1 knock-in strain that provides a more native-like expression level (Halbleib et al., 2017; Aragón et al., 2009). Notably, the signal from the crosslinked species was neither increased by the use of more reactive crosslinking agents (e.g. HgCl₂ or Cu²⁺-phenanthroline) nor by harsher crosslinking conditions (higher

temperatures or increased concentrations of the crosslinking agent). In fact, more reactive agents and harsher conditions only caused a loss of the total HA-positive signal presumably due to an unspecific crosslinking and/or aggregation of Ire1 (data not shown)."

To extend on this point: From a careful review of the available literature, it remains somewhat unclear, which fraction of Ire1 is found in clusters even under severe forms of ER stress. While the fraction of clustered Ire1 has been previously estimated to be ~70-85% (Aragón et al., 2009 (PMID: 19079237)), this estimate relied on microscopic data using Ire1-mCherry expressed from a *CEN*-based plasmid and a number of reasonable, yet critically important assumptions for image analysis. It is possible that the fraction of clustered Ire1 is lower than originally determined. This would be consistent also with more recent findings from the Peter Walter lab for IRE1alpha (PMID: 31871156) stating that only a small fraction (~5%) of IRE1alpha is found in clusters even under severe ER stress.

Related to our answer to point 1 by this reviewer, we want to highlight an important point: The impact of even mild overexpression must not be underestimated. The original study reporting 70-85% of Ire1 in clusters, used a plasmid-based expression of *IRE1* (PMID: 19079237). These conditions lead to a mild, yet significant overexpression, which should also increase the likelihood for Ire1 dimerization and oligomerization. For the sake of simplicity, let's consider only the case of homo-dimerization, where two monomeric Ire1 molecules come together to form a dimer. The law of mass action predicts that a 2.7-fold overexpression of Ire1 (the difference in the expression level from a plasmid-based expression versus a knock-in approach) would already lead to a >7-fold-increased concentration of the dimer! Taking these considerations into account, we have used a knock-in approach for the expression of IRE1, used milder conditions of ER stress and established particularly mild crosslinking reactions to study structure-function relationships of signaling-active Ire1.

We have extended our discussion to explain our experimental strategy.

How representative are the in vitro crosslinking experiments performed on ice to the actual Ire1 dimeric states in vivo?

X-linking can be used to dissect the configuration of the transmembrane helix region in dimeric and/or clustered Ire1 by scoring the reactivity of pairs of cysteines. Working on ice should lower the dynamics of the system, but it is unlikely to change the transmembrane architecture altogether. We used mild crosslinking conditions because at higher temperatures and/or higher concentrations of Cu²⁺ we observed unspecific protein aggregation and/or loss-of-Ire1-specific-signals in immunoblots. Our mild X-linking conditions provide full recovery of the Ire1-specific signal. Notably, cysteine-crosslinking (often performed on ice) has been very successfully used in the past to study the configuration of transmembrane helices in signaling-active proteins (references in the manuscript).

However, X-linking is not a suitable method to determine the fraction of dimerized/clustered Ire1. X-linking is irreversible and cannot reliably dissect the dynamic equilibrium between monomers, dimers, and clustered Ire1.

We discuss our choice for the crosslinking assay in greater detail in the revised manuscript.

Does it mean that only the fractions of dimerized Ire1 are activated and thus the sole contributors to HAC1 splicing?

As indicated in our answer to point 1, the fraction of dimerized/clustered Ire1 is a matter of active debate. We are convinced that X-linking experiments cannot solve this open question. However, our data and observations may help to fuel a new discussion. We feel that sophisticated microscopic approaches with single molecule sensitivity bear the potential to address this important unresolved question. Based on the reviewer's comments, we extended our discussion on the X-linking data.

(4) I would like the authors to comment on the reasons that could explain the difference of Ire1 F544C or Cys-less expression levels in unstressed conditions. It might be due to loading variation, but a loading control is not included in Fig. 2C. Alternatively, should we assume that each lane contains between 6.4 and 9.6 µg microsomal proteins (based on 1 {plus minus} 0.2 mg/ml protein)? #

From many experiments with these variants, we have no evidence for a significant difference in the expression level for the different mutants of Ire1 (stressed or unstressed). Even though there may be some degree of variations in loading (as indicated), we report the X-linking propensity as the ratio $\%X\text{-link} = \text{Intensity}_{\text{Dimer}} / (\text{Intensity}_{\text{Dimer}} + \text{Intensity}_{\text{Monomer}})$ in Figure 3, 5, and 6. Hence, the reported X-linking propensity should be independent of possible loading variations.

(5) In Figure 3 and S3, it is apparent that mutations in Ire1 AH region will increase sensitivity to ER stress in cells, but not for TMH variants. Yet, it is puzzling that at a comparable level of HAC1 mRNA splicing (HAC1s), AH-

cysteine-variants (E540C, T541C, and G542C) displayed sensitivity against ER stress despite having comparable levels of HAC1s relative to WT.

This is indeed a puzzling, yet important observation. The E540C, T541C and G542C mutations do not seem to affect *HAC1* mRNA splicing during acute (1h) proteotoxic stress caused by 2 mM DTT. However, there is a significantly increased sensitivity to DTT of the respective yeast strains upon overnight cultivation (18 hrs) in the presence of DTT. Why is that? We decided to address this important point at a later stage in the manuscript with more experiments and an extended discussion.

To directly answer the reviewer's point: We think that E540C, T541C and G542C (all mutations are in the amphipathic helix (AH)) are defective in sensing lipid bilayer stress and the membrane-based stresses caused by prolonged DTT-induced ER stress. However, they show a full, normal response to an acute proteotoxic stress, because under these conditions the stress from unfolded proteins in the lumen of the ER dominates.

We have added more data and an entire paragraph to the discussion related to this important point. Notably, the T541C and G542C mutations (in the AH) cannot respond to any form of ER stress when combined with IF2 mutation (new Figure 6D,E). In contrast to that, the F544C mutation (in the TMH leaving the AH of Ire1 intact) can still respond to lipid bilayer stress and membrane-based stresses caused by prolonged DTT treatments (Promlek et al., 2011) (new Figure 6D, E).

In the revised discussion, we state:

"Proteotoxic stress caused by DTT or TM is characterized by two phases: An early phase of a rapid UPR activation with little to know changes in the lipid composition and a second, slower phase characterized by a build-up of membrane-aberrancies (Promlek et al., 2011; Reinhard et al., 2020). While these membrane aberrancies remain poorly characterized, they serve as a robust signal for Ire1 activation (Fig. 6D) (Promlek et al., 2011). The lipid bilayer stress caused from inositol-depletion, in contrast, lacks the early phase of UPR activation. It manifests slowly and causes a distinct temporal pattern of UPR activation (Fig. 6E). It will be interesting to study, if different temporal patterns of UPR activation are sufficient to give rise to largely distinct transcriptional programs or if -alternatively- Ire1 can custom-tailor its output via yet unknown mechanisms (Hetz et al., 2020; Ho et al., 2020; Fun and Thibault, 2020)."

We are convinced that the newly added data and the extended discussion of those seemingly puzzling observations represent an important new aspect of our revised manuscript.

(6) The authors stated "Because F544C, the best-crosslinking residue, and L549C seemingly lie on opposing sites of Ire1's TMH as judged from a helical wheel representation (Fig. 3B), this raises the question if the corresponding residues in the native TMH can face each other at a low distance and at the same time" (page 6). However, I find this argumentation rather weak as with DTT and Tm, variants E540C, T541C, I550C, and C552 exhibit similar dimerization when compared to L549C levels.

We understand the point. Originally, we have added this argument in response to a reviewer's suggestion in the revision phase at a different journal. We agree that this statement is not particularly strong. However, we also feel this statement might help those readers expecting a perfect helical pattern for the crosslinking propensity in the TMH. Such perfect helical pattern can be expected if two helices stably associate in a strictly parallel fashion. This, however, is clearly not the case for Ire1, whose TMHs are highly bend and associated as an 'X'. We decided to leave this sentence in the revised manuscript.

(7) The authors also stated that they have performed an "unbiased MD [molecular dynamics] simulation" (page 6). In my view they have been biased as their MD simulation is based on yet another MD simulation from their previous publication (PMID 28689662). They used their previous MD simulation model to generate an Ire1 dimeric MD simulation. In my view their proposed "X-shaped" dimer is very speculative, and it lacks experimental evidence to support it. I can appreciate that it is a difficult weakness to address. As this model is central to their conclusions, additional experiments should be performed in vitro and in vivo. For example, the authors could get NMR structures of Ire1 TMH in liposomes. Otherwise, I strongly suggest the authors to tone down their conclusions.

We thank the Reviewer for giving us the opportunity to clarify our writing. In this context, we used 'unbiased' as a technical term that refers to molecular dynamics simulations in which the system evolves according to its spontaneous dynamics without the action of any external biasing (driving) force. We reckon that the term created confusion and rephased all sections mentioning it. We originally performed two independent sets of simulations: in the first the initial dimer model sampled its 'unbiased' dynamics; in the second, we enforced an external bias—a harmonic potential restraint—to keep the two protomers close. However, since the dimeric model maintained the overall X-shaped configuration in the first simulation, we did not use data coming from the externally biased one in the current manuscript. Hence, we have removed those sections in the methods parts.

We thank the Reviewer for the interesting suggestions on how to further validate our structural model. Indeed, we have undertaken first steps to determine the orientation/structure of Ire1's transmembrane

domain by solid-state NMR. In the current manuscript, we do claim to report a structure of Ire1's transmembrane domain and we have carefully rephrased (toned down) our conclusions.

Furthermore, we extended our analysis of the MD simulation data, which indicate that the modeled and simulated Ire1 dimer 'squeezes' the surrounding lipid bilayer (new Figure 4B).

New Figure 4B: Membrane thickness around the sensor peptide, defined as the average vertical distance between the two phosphate layers, averaged over MD simulations in POPE. A representative structure of the dimeric TMH region is shown in blue.

(8) In light of comment # 7 above, I disagree with the authors statement "Our crosslinking approach indicates that the TMH residue F544 is part of a small interface between Ire1 protomers, which might stabilize the unusual X-shaped transmembrane configuration of Ire1. Interestingly, an aromatic residue in the TMH region of the mammalian IRE1 α (W547) was reported to stabilize the oligomeric state of IRE1 α (Cho et al., 2019). Hence, we embarked on testing a similar role for F544 in Ire1 from baker's yeast". The positioning of mammalian W547 within the ER membrane is very different and should be closer to the cytosol when compared to yeast F544.

We agree and have carefully rephrased this section. We now state: "Our crosslinking approach indicates that the TMH residue F544 is part of a small interface between Ire1 protomers, which might stabilize the unusual X-shaped transmembrane configuration of Ire1. Aromatic residues TMH residues have been implicated in sensing both lipid saturation by Mga2 (W1042) (Covino et al., 2016; Ballweg et al., 2020) and lipid bilayer stress by the mammalian IRE1 α (W547) (Cho et al., 2019). Despite a different position within the ER membrane, we wanted to test a similar role for F544 in Ire1 from baker's yeast."

(9) The authors hypothesized that the stability of Ire1 dimers is likely dependent on the biochemical properties of AH rather than that of TMH. However, the increased sensitivity toward ER stress and the weakened crosslinking propensity observed in Ire1 F531R variant (Figure 5C,D) may be a result of destabilization of the luminal domain of Ire1 per se.

While it is impossible at this moment to formally exclude any possible impact of the F531R mutation on the stability of the ER-luminal domain, we consider this possibility as highly unlikely. In fact, we have performed a series of controls in our previous paper, which first established the F531R mutation (Halbleib et al. 2017). We do not think that the F531R mutation destabilizes the ER-luminal domain, because

- the linker between the transmembrane domain and the ER-luminal domain is predicted to be flexible
- the F531R mutation does not impair the interaction between Ire1 and Sec63 (in stressed and unstressed cells)
- the F531R mutation does not impair the interaction with Ire1 and Kar2 (in stressed and unstressed cells)
- The F531R mutation shows genetic interactions with the IF1 and IF2 mutations in ER stress resistance assays. If the ER-luminal domain would be disrupted by the F531R mutation, these interfaces (IF1 and IF2) should already be disrupted and should not show additive effects with the F531R mutation).

Furthermore, we have previously shown that the AH *alone* (in the absence of a ER luminal domain) can undergo a membrane-controlled oligomerization, which is disrupted by the F531R mutation.

For these reasons, we do not think that the F531R mutation has a major impact on the folding/stability of the ER -luminal domain of Ire1, but instead that it contributes to a membrane-controlled oligomerization of Ire1. For a detailed discussion of this point, we refer to our previous paper (Halbleib et al., 2017) in our revised manuscript.

Moreover, Ire1 Cys-less mutant and the other cysteine mutants, Ire1 F531R has a lower propensity to form clusters (reduction of ~50% compared with WT with 2 mM DTT) and is less sensitive to proteotoxic stress (~45% relative spliced HAC1 mRNA compared to WT with 4 mM DTT).

We hope that we understand this point correctly. The reviewer seems to find the functional consequences of the F531R mutation as rather too weak, especially when compared to the minor functional differences already observed between cysteine-less Ire1 and WT Ire1. While there are some minor functional differences between Ire1 WT and cysteine-less Ire1 (and the single-cysteine mutants), we clearly show that cells expressing cysteine-less Ire1 are resistant to prolonged ER-stress caused by DTT (Figure 1B). Our data show that cysteine-less Ire1 responds efficiently to both proteotoxic and lipid bilayer stress.

We extended our discussion on the results regarding the F531R mutation in Fig. 5 and Fig. S5 by adding an entire paragraph. The F531R mutation sensitizes cells to DTT (Fig. 5C), while the cysteine-less variant barely affects the cellular resistance, if at all (Fig. 1B). For a more detailed phenotypic characterization of the F531R mutant, we also refer to our previous publication (Halbleib et al., 2017). Meanwhile, an important role of the AH-region for normal Ire1 function has been independently confirmed by two other laboratories (PMID: 30319071; PMID: 32349127).

At the same time, our revised manuscript provides evidence that an intact AH in Ire1 is required for the response to prolonged forms of ER-stress caused by DTT and to lipid bilayer stress (new Fig. 6D,E). The T541C and E540C mutations (in the AH region) exhibit strong synthetic defects with the IF2 mutation (Fig. 6C) and new Fig. 6D,E).

New Fig. 6D:

New Fig. 6E:

Thus, our revised manuscript underscores the importance of the AH for normal Ire1 function.

(10) In Figure 6D, the authors show that Ire1 T541C/IF2 variant forms CuSO₄-mediated dimers while Ire1 T541C/IF2 variant is sensitive to ER stress in Figure 6C. UPR activation reporter assays may be good indications to address the discrepancy between Ire1 dimerization and sensitivity to ER stress.

Indeed, X-linking suggests the formation of T541C/IF2 dimers (Figure 6F in the revised manuscript). Microscopy suggests a lack of cluster formation for F541C/IF2 (Figure 6B). Thus, our data suggest that dimer formation is not sufficient for mounting a UPR. In our view, this is consistent with previous reports and current models regarding the activation of Ire1. Following the reviewer's advice, we performed more functional assays (though we did not use UPR activation reporter assays, which rather report on the average degree UPR activation over a much longer period of time) (new Fig. 6D,E).

Consistent with the functional assays on ER stress sensitivity (Fig 6C in the revised manuscript), we find that neither the T541C/IF2 nor the E540C/IF2 variant mediates HAC1 mRNA splicing in response to lipid bilayer stress caused either by inositol depletion or prolonged DTT treatments. However, the F544C/IF2 mutant featuring a non-disrupted, functional AH responds normal to inositol-depletion (despite a disrupted IF2) and it also responds to efficiently to DTT after prolonged periods of time (presumably because the oligomerization and activation of Ire1 is dominantly driven by a membrane-based mechanism under these

conditions, which is sensed by the AH). We provide an extended discussion on these exciting observations in the revised manuscript.

(11) The authors concluded that the different forms of ER stress (DTT, Tm, and inositol depletion) converge in a single, signaling-active conformation of Ire1.

In the revised manuscript, we toned down our statements on the structural organization of the TMH region. We now state that different forms of ER stress converge in a similar, overall X-shaped organization of the transmembrane region.

The crosslinking of Ire1 variants have been tested during inositol depletion (LBS) in Fig. S3C. However, the authors only focus on proteotoxic stress for the remaining experiments (Fig. 5 and 6). The authors would need to systematically perform the experiments of Fig. 5 and 6 during LBS to support their conclusion.

We feel that the data in Figure 3 (supported by extensive MD simulations in Figure 4) are sufficient to suggest a similar X-shaped organization of the neighboring transmembrane domains. The experiments in Figure 5 and 6 were not designed to validate the structural organization of neighboring transmembrane helices under different forms of ER stress (as it is done in Figure 3).

I am wondering why the expression of some of the Ire1 variants are much lower compared to others during LBS (Fig. S3C). Are these destabilized by LBS?

As of now, we have no evidence that specific Ire1 variants are destabilized by LBS. After carefully re-checking all available data, we fail to see a selective destabilization of specific Ire1 variants by LBS as it may be implied from the variable band intensities in the immunoblot analysis shown in Figure S3C.

Figure S3C shows an immunoblot after microsomes preparation and after treating the microsomes with Cu^{2+} as a X-link promoting agent. All our crosslinking data show some degree of variability (each datapoint in Figure 3F indicates a specific experiment). This variability seems to reflect minor experimental variations in the preparation of microsomes, the subsequent Cu^{2+} mediated X-linking, and the quantification of immunoblots (which is not trivial when working at an endogenous protein level).

Notably, crosslinking experiments are technically challenging even when working with purified proteins. We perform the crosslinking experiments with microsomal preparations because we want to study Ire1 in its natural context. In fact, we are extremely happy with the quality and reproducibility of our crosslinking data.

We addressed the potential issues arising from a variable microsomes preparations by performing three to five independent preparations and experiments.

Reporting the X-linking propensity as a ratio $\%X\text{-link} = \text{Intensity}_{\text{Dimer}} / (\text{Intensity}_{\text{Dimer}} + \text{Intensity}_{\text{Monomer}})$ should make all data independent from differences in the preparation and in gel loading.

Minor points to address (I refer to the page numbers of the generated PDF and the authors didn't make the effort to number the pages nor the lines of their manuscript)

(1) Define TMH (page 3)

Corrected.

(2) I find the labelling of amino acid numbers around AH and TMH confusing in Figure 1A. The authors might want to align the aa numbers and the domains or to indicate them clearly with a line.

We have simplified the aa numbering and aligned the numbers.

(3) Page 5, "We conclude that CuSO_4 can catalyze the formation of disulfide bridges between two neighboring Ire1 copies" should be replaced by "We conclude that CuSO_4 can catalyze the formation of disulfide bridges between two neighboring Ire1 copies"

Corrected.

Thank you for your extensive efforts and review!

Reviewer #2 (Comments to the Authors (Required)):

This study focuses on the architecture and function of transmembrane helix (TMH) and amphipathic helix (AH) of the yeast ER stress sensor Ire1. The authors employed a crosslinking-based strategy to pinpoint the contacts between transmembrane and amphipathic helices within Ire1 oligomers. To this end, they engineered a functional cysteine-less mutant of Ire1 and systematically replaced residues in the TMH and AH with cysteines. Contacts

between helices were then identified via CuSO₄-mediated crosslinking in isolated ER microsomes. It is known that Ire1 signals through dimerization and/or high-order oligomerization, but the TMH conformation in dimeric and oligomeric Ire1 assemblies formed by different activators are poorly understood.

This approach revealed a consistent profile of crosslinked residues that are in close contact in response to IRE1 activation. Out of the tested interface residues, the F544C mutation resulted in the highest level of crosslinked dimers. Therefore, the authors conclude F544 is the main point of contact between the TM helices of an Ire1 dimer. When crosslinking was tested under proteotoxic stress (+DTT or Tunicamycin) or lipid stress (inositol depletion), crosslinking profiles were similar, suggesting that both types of stresses result in similar oligomeric arrangements of the TMH. Combining the crosslinking and functional studies together with simulation results, the authors conclude that the active form of Ire1 has X-shaped TMH conformation using F544 as the main interface.

The authors also studied the impacts of AH, IF1, and IF2 mutations on Ire1 TMH crosslinking propensity and functions. Consistent with their prior studies, disrupting the AH can lead to less crosslinked products and more stress sensitive phenotype. IF1 and IF2 (interfaces 1 and 2) were previously shown to play key roles in dimerization and oligomerization of Ire1, respectively. By disrupting IF2 and leaving IF1 intact, the authors are able to distinguish the TMH contacts are dimer-specific or oligomer-specific. They identified that the F544-F544' and T541-T541' contacts are most likely within dimers, whereas the E540-E540' contact is across dimers.

To summarize, this is a strong paper that adds important details to our understanding of the structural organization of Ire1 oligomers. It clearly reveals several residues at the interface between adjacent Ire1 transmembrane helices and provides a plausible model of the relative arrangement of the helices in Ire1 oligomers. The paper is well written with clear descriptions of the experimental approaches and an easily understandable discussion of the results.

Before publication, please consider and where appropriate address the following issues:

Major points:

1. How efficient is CuSO₄, an ionic water-soluble compound, at catalyzing cysteine crosslinking inside the lipid bilayer? Is there any published literature assessing the efficiency of CuSO₄ or other oxidizing reagents in the membrane context? It is important to reference and address if the findings can be significantly biased by the relative "depth" of each residue inside the membrane.

The reviewer is right. We extended our discussion on the crosslinking conditions and referenced additional manuscripts using Cu²⁺ (and other crosslinking agents). In the revised manuscript, we discuss the choice of the crosslinking conditions more extensively. We also discuss a potential 'bias' of the assay from the relative 'depth' of each residue in the membrane.

2. Can the authors comment on why crosslinking was performed on ice and not closer to yeast growth temperature and MD simulation temperature? How does this temperature difference affect membrane mobility and protein conformational dynamics?

The reviewer addresses important, technical points. We have carefully, optimized the crosslinking conditions. Our major goal was to establish mild X-linking conditions in order detect the crosslink of only perfectly spaced and oriented cysteine residues. It was necessary to perform the X-linking experiments at 4°C. The MD simulation was performed at 30°C. Nevertheless, the overall X-shaped architecture of the dimeric TMH region remained remarkably stable throughout the entire simulation.

Working at higher temperatures for the X-linking, the use of more reactive catalysts, or at higher concentrations of Cu²⁺ led to much-increased unspecific X-linking and an unspecific aggregation of all proteins (not only of Ire1). Together, this led to a loss-of-signal (a phenomenon not uncommon to any X-linking approach). In the revised manuscript, we highlight the potential caveats associated with performing the assays on ice, which -however- was necessary.

3. The statement 'Ire1 does not seem to custom-tailor the signaling output to the specific input signals' is not supported by the data in this manuscript. Exhibiting a similar crosslinking profile in one specific region of the protein does not necessarily translate to identical downstream results. There could be a number of additional factors for tuning the outcome of downstream signaling (later discussed in manuscript), such as the relative abundance of dimers and oligomers, association of additional proteins, or targeting of the Hac1 mRNA to active Ire1.

We have carefully rephrased our discussion. We have toned down our statements, especially those regarding the signaling outcome of Ire1.

4. Fig S3C. The gel shown seems to be an anomaly rather than a representative gel quantified in Fig. 3F (crosslinking under inositol depletion). Please show a more representative gel or address the high crosslinking for T541C.

The reviewer is correct. There are variations in the data as also indicated by the data points from individual experiments in Figure 3F. We decided against stitching together a gel with the most representative data. However, we will provide all relevant raw data via the online Mendeley Data platform (DOI:10.17632/s52vt8spmc.1).

5. Fig 6C. The authors do not explain clearly why F544C/IF2 mutant can partially rescue the stress sensitivity resulting from the IF2 mutation whereas other double mutants were more sensitive. Is that because F544C can lock the dimer in a conformation that can enhance Ire1 activity? Please discuss/provide a plausible explanation.

We have substantially extended our discussion and added new data regarding this intriguing point (Fig. S5A,B).

We now state: “Similarly, when the AH-disrupting mutation F531R was combined with the F544C mutation (at the crossing-point of the X-shaped TMH-dimer), we observed only a very mild, yet significant functional defect (Suppl. Materials Fig. S5A) and a strongly reduced crosslinking propensity (Suppl. Materials Fig. S5B). This robust resistance to DTT is somewhat surprising considering the strongly reduced crosslinking propensity. However, the disruption of the AH changes the placement of the TMH in the membrane and the degree of membrane thinning and water penetration (Halbleib et al., 2017). We speculate that these combined changes would place the polar F544C residue more deeply in the hydrophobic core of the membrane, thereby affecting its propensity to undergo a Cu²⁺-catalyzed crosslinking, but at the same time favoring Ire1 dimerization. Notably, the F544C mutation alone does not lead to an increased UPR activity and ER stress resistance (Fig. 3C, D; Suppl. Materials Fig. S3A). In fact, the primary sequence of Ire1’s TMH can be systematically mutated (Fig. 3C; Suppl. Materials Fig. S3A), scrambled (in the case of the mammalian IRE1 α) or exchanged altogether (Halbleib et al., 2017; Volmer et al., 2013) without causing a detectable functional defect. It therefore seems that a suitably placed polar residue in the TMH, here through the F544C mutation, becomes phenotypically relevant only when Ire1 is otherwise compromised. Beyond that, our data suggest that the overall architecture of the TMH region with an intact AH is particularly relevant for normal UPR function.”

6. Fig 6D, last panel. Why do F544C/IF2 and F544C alone mutants have different crosslinking levels with tunicamycin treatment but similar level with DTT treatment? Is this result still consistent with the authors' conclusion that TMH of Ire1 has similar structural output to both proteotoxic and lipid stress?

While there are some minor differences in the degree of crosslinking observed in DTT- versus TM-treated cells, we do not dare to interpret those due to the inherent, substantial variability of the X-linking data. The difference in crosslinking observed between DTT- and TM-treated cells in Fig. 6D is -in our view- too low to draw a firm mechanistic conclusion. We feel that minor differences in the crosslinking propensity observed between DTT and TM do not affect our conclusion that Ire1 has a similar overall transmembrane architecture during proteotoxic and lipid bilayer stress.

The most important findings from this Figure (now Fig. 6F in the revised version) is that the IF2 mutation does barely (if at all) reduces the crosslinking via T541C and F544C, even though it massively disrupts the formation of higher oligomers. This is also the point, which we discuss most prominently in the paper. These observations suggest that the crosslinking of these residues occurs between protomers in a dimer.

7. The interpretation and presentation of simulation data can be greatly improved. What is the population of the X-shaped conformation that was shown in Figure 4 in 600 ns simulations? Why is the simulation length of 600 ns chosen? The authors may also consider calculating the free energy of different conformations throughout the 600 ns and map where the X-shaped conformation is in the energy diagram. Does the X-shaped conformation represent the minimum energy state?

We thank the Reviewer for the interesting suggestions. In our intentions, our modelling and molecular dynamics simulation has two goals. First, we wanted to create a structural model that integrates what we know about the Ire1’s transmembrane monomer from previous simulations and EPR experiments (Halbleib et al., 2017) with the new experimental evidence reported in the current manuscript. If we consider cross-linking data as distance restraints, where the ‘X-shaped’ model is the only one that satisfies all restraints at once. Second, we ran 1000 ns atomistic molecular dynamics to test whether the proposed model is reasonable from a structural point of view. A deeply flawed initial model would rearrange or even fall apart within few tens of nanoseconds. In our case, instead, we observe only local rearrangements—for instance side chains changing their orientation—but the overall dimeric arrangement remains the same throughout the whole simulation. Since we do not sample any alternative dimer conformations, we cannot estimate a free energy. However, the Reviewer is right in suggesting that the dimeric model we proposed should represent a free energy minimum conformation, in particular with respect to the two unbound protomers. Calculating the free energy of dimerization from atomistic simulations for requires developing new methods and approaches, which is what we are currently focusing on.

8. The authors postulate that some of the residues in the transmembrane helices are important for intra-dimer contacts and others are important for inter-dimer contacts. One potential way to further validate this exciting notion

would be to generate double-cysteine mutants, where one each of the intra- and inter- dimer residues are mutated to cysteines. If these residues do in fact bridge distinct interfaces, one would then expect to see the appearance of higher-order oligomeric species on the crosslinking gel (unless some technical hurdles impede this experiment; these hurdles could be addressed in the manuscript).

Indeed, such experiments with double mutants would be very informative, but they are quite challenging. While we have tried to X-link a few double cysteine mutants, the expected signals for such double-crosslinked species was too low for a robust, reliable detection. Given the fact that we are performing the X-links in native membranes, we are very competing with a background from X-linking of Ire1 with unrelated proteins exposing cysteine-residues.

Minor points:

1. Typographical error in Figure 1. It should be 'amphipathic helix' not 'ampipathic helix'.

We corrected this typo.

2. In the imaging experiments, the outline of the ER is not visible at all in the no-stress conditions. In published works, yeast Ire1 signal is clearly present in the ER network without stress, and coalesces into foci when stress is added. This seems to be due to sensitivity or expression level. Please address this different appearance in manuscript.

Following the advice of the reviewer, we have extended the discussion regarding the microscopy data and the role of the expression level. We also added additional data (new Fig. S1F) to support a 'normal' localization of Ire1-clusters in the ER. See also our answer to point 1 by reviewer 1.

3. 'Upon dimerization of an unrelated single-pass transmembrane protein with a circular 'footprint', only a smaller portion of the deformed membrane can coalesce': are there any references to support this statement?

We have added more literature to support his statement and extended our discussion. We now state: "The specific way each membrane protein locally deforms the bilayer, referred to as membrane 'footprints' (Haselwandter and Mackinnon, 2018) or 'fingerprints' (Corradi et al., 2018), could be at the origin of membrane-sensitivity and, more generally, control the organization of supramolecular assemblies (Corradi et al., 2018). Is it possible that the unusual TMH region of Ire1 and its resulting footprint serves a specific function? We speculate that the combination of a short TMH with an AH inserting deep into the bilayer contributes to Ire1's exquisite sensitivity to aberrant ER membrane stiffening. The region of membrane compression around monomeric Ire1 is, when viewed from the top, not of circular shape but ellipsoid due to the membrane-inserted AH (Fig. 6G) (Halbleib et al., 2017). Based on simple geometric considerations, it is conceivable that the total extent of membrane deformation contributing to the free energy of dimerization depends on how precisely the two TMH regions are arranged towards each other. Our structural model of the dimeric TMH suggests that the two protomers associate via the longer edge of membrane deformation (parallel to the major axis of the ellipse) (Fig. 4A, B) thereby maximizing the area of coalescence (Fig. 6G, top) and minimizing the free energy. We speculate that Ire1 is more responsive to aberrant membrane stiffening than other single-pass transmembrane proteins with short TMHs but without AHs. Because these proteins also lack the characteristic ellipsoid shape of membrane deformation (Kaiser et al., 2011), they coalesce only a smaller area of their footprints upon dimerization (Fig. 6G, bottom). It will be intriguing to study the membrane-driven dimerization and oligomerization of Ire1 side-by-side with other single-pass membrane proteins exhibiting distinct membrane footprints using advanced microscopic tools such as single-molecules photobleaching (Chadda et al., 2016)."

4. We would suggest having a better labeling on the cartoon representations in Figure 5A and 5C. Showing the ER lumen and cytosol sides will make it easier to read.

As suggested, we added labels to Figure 5A and 5C.

5. In term of clarity, showing the crosslinking gels only with globally applied contrast adjustment makes it difficult to see the much fainter crosslinked product relative to monomeric Ire1. In some instances, the figures could benefit from adding a cropped panel showing just the crosslinked product, contrast optimized for this gel region.

Considering the discussion on the efficiency of crosslinks (see point 3 by reviewer 1), we decided against the use of 'overexposed' gels. We hope that the contrast in the raw data is sufficient.

6. Page 7. "...polar residue F544C...alters the dimerization/oligomerization propensity": In future work, this interpretation can be experimentally tested by substituting F544 with non-polar residues.

Thank you for this suggestion. For the revised manuscript, we have extended our discussion on the role of a polar residue at the position of F544C. Indeed, we plan further experiments to tune the sensitivity of Ire1 by mutations in the entire TMH region.

Reviewer #3 (Comments to the Authors (Required)):

Recent studies of the unfolded protein response (UPR) signaling pathway have started to dissect the molecular mechanisms of how UPR-initiating components of the ER membrane recognize and respond to the increased demands of ER proteotoxic stress and lipid bilayer stress ("lipotoxic" stress). Specifically, for Ire1, one of the UPR inducers on the ER, the transmembrane (TM) domain was thought to hold a key to responding to lipotoxic stress, as Ire1 retains the ability to respond to lipotoxic stress even with a truncated ER luminal domain, as shown by the Kimata (2011) and Ron (2013) groups. Interestingly, swapping the Ire1 TM domain with that from calnexin, an unrelated protein or TM domain sequence was scrambled did not affect Ire1's ability to respond to lipotoxic stress, revealing the lack of specific amino acid requirements for the Ire1 TM domain. Furthermore, the Ernst group published an intriguing finding that an amphipathic helix (AH) adjacent to and overlapping with the Ire1 TM domain plays a critical role in the response of yeast Ire1 to proteotoxic stress and lipids stress.

In this manuscript, Vaeth et al attempts to extend the Ernst study by dissecting how the AH and the IRE1 TM domain respond to lipid bilayer stress (induced by inositol depletion) in addition to proteotoxic stress. Ire1 contains multiple cysteine residues throughout the protein and thus, the authors generated a cysteine-less Ire1, making it possible to introduce a single cysteine within the AH and/or TM domains of Ire1 to perform CuSO₄-cysteine cross-linking experiments. The data in this manuscript represents ample work and should be published. Specifically, the generation of a cysteine-less Ire1 will likely provide a useful tool for future studies of Ire1. This approach allowed the authors to introduce single cysteine substitutions for CuSO₄-Cys crosslinking experiments.

We thank the reviewer for this positive assessment of our study.

Together with simulation, the authors uncovered that F544 within the TM domain is a key residue for dimer formation in response to both proteotoxic and lipotoxic stress. Unfortunately, the conclusions presented here do not significantly extend the conclusions from the previous study. In this reviewer's opinion, papers published in Journal of Cell Biology should provide a significant advancement in our understanding of a specific topic(s) from prior knowledge. Therefore, in its current form, this manuscript would be more suitable for other journals.

We respectfully disagree. This is the first study investigating the transmembrane architecture of Ire1 in stressed cells. No previous study has addressed the configuration of neighboring transmembrane regions of Ire1. Our previous study shows the potential of the transmembrane helix region to oligomerize in membrane-dependent manner, but it did not establish a model for dimeric Ire1. Notably, a recent cryo-EM study (doi.org/10.1101/2021.02.24.432779) resolved the ER-luminal domain of mammalian IRE1alpha, but did not resolve its transmembrane region.

Given the importance of the transmembrane helix region in sensing aberrant ER membrane properties, there is an urgent need for structure-function analyses focusing on processes in and at the ER membrane.

In particular, the extents of 'Ire1 dimer' seem very low (most Ire1 remained monomer). The significance of this cross-linking is unclear for a few reasons. First, the assay was performed in microsomes prepared after ER stress induction of cells. During the preparation of microsomes, it is unclear if the ER membrane retains any potential changes imposed by the ER proteotoxic or lipotoxic stress.

Some of the concerns regarding the microsome preparation are addressed in our answer to the major point 4 by this reviewer. We have no indication that the ER membrane composition/properties change during the isolation of microsomes.

Furthermore, the effects of different amino acid changes were measured by (1) cell density test results across increasing concentrations of DTT (Fig S3A), (2) HAC1 splicing (Fig 3C), (3) dimer formation in microsomes (Fig 3E-F), and (4) Ire1 foci formation in vivo (Fig S3D-E). If a specific residue plays an important role in functional dimerization or oligomerization, these values should correlate with each other. However, among mutants displayed some Ire1 activities, these values did not show significant agreement in their assays; thus, the functional significance of Ire1 residues such as F544 is unclear.

We respectfully disagree. Different types of assays report on different time scales. A defect observed for a specific Ire1 mutant by a particular assay at a certain time, does not automatically mean this mutant should show functional defects in another assay.

We understand that a lack-of-correlation between functional assays may seem puzzling to the readership. We have added an entire paragraph to the discussion to address this point. Furthermore, we have performed additional experiments (new Fig. 6D,E) to better understand the role of F544 and the role of an intact AH for Ire1's membrane-sensitivity. We hope that our efforts and extended discussion suitably addresses the reviewer's concerns.

Based on previously published studies, additional issues include the following:

(1) Crosslinking experiment: In addition to the issues described above, another issue is that even if the microsomes

retain the ER stress-induced changes on the ER membrane and Ire1, it is unclear how the authors ensured that Ire1 crosslinking was not impacted by the Ire1-Ire1 interaction *in trans* between the different vesicles.

We respectfully disagree. Microsomes have been used in the identification and dissection of key cellular processes including protein translocation, membrane protein folding, and COPII vesicle formation. We are convinced that microsomes represent an invaluable tool to also dissect mechanistic questions related to the UPR. The technical concerns of the reviewer regarding the isolation of microsomes will be addressed in response to the point 4 raised by this reviewer.

The engineered cysteine variants are in the transmembrane region of Ire1 and are thus -to a large part- buried in the hydrophobic core of the bilayer. Even if there would be considerable Ire1-Ire1 interactions *in trans* and between vesicles, our assay would only report on the proximity of two cysteines located within the same membrane.

Our data do neither confirm nor exclude the possibility of an Ire1-Ire1 interaction *in trans*. The focus of our study is the organization of Ire1 transmembrane domain. When submitting our manuscript to JCB, there was -at least to our knowledge- no study on Ire1-Ire1 interactions *in trans* and between vesicles (or across the ER lumen) available. Only after receiving the reviews from JCB, a manuscript appeared on bioarchives (doi.org/10.1101/2021.02.24.432779) that reports on Ire1-Ire1 interactions *in trans* (however, for the mammalian system).

(Ex. HAC1 mRNA splicing value for G542C is high (Figure 3C), but almost no Ire1 dimerization (Figure 3E-F)).

Indeed, G542C is functional (Figure S3A), it mediates HAC1 mRNA splicing (Figure 3C), and it forms higher-order oligomers (Figure S3D), while it is not efficiently crosslinked. This does not mean that Ire1 does not dimerize. It means that the two cysteines in neighboring Ire1 protomers are not in a suitable distance and/or not suitably oriented to form a disulfide bond. In other words: Lack of X-linking does not mean a lack of dimerization.

(2) HAC1 mRNA splicing: How was the HAC1 mRNA splicing calculation performed? What was calculated as 100% splicing? Was the HAC1 splicing in ER-stressed WT cells calculated as 100%? Was HAC1 splicing of mutant cells compared to that value? In addition, unstressed cells with WT Ire1 showed closed to 50% HAC1 mRNA splicing according to the presented values. While the basal HAC1 splicing levels can fluctuate to some degree between YPD and SCD, the splicing values differ significantly (in Fig 1C, the value seems to be <5%). The authors should show the actual quantitative RT/CPR data.

See our answer to point 2 by reviewer 1. Shortly, we report predominantly the level of the spliced HAC1 mRNA and not the degree of HAC1 splicing. For the revised manuscript, we added more data to clearly show the degree of HAC1 mRNA splicing under the given conditions (new Fig. S1D and S3B).

(3) Ire1 foci in ER-stressed cells: Based on previous studies, Ire1 is thought to undergo dimerization and oligomerization in response to ER stress. The sizes of the Ire1 foci shown in this manuscript appear to be heterogeneous (for example, in Fig 1D), consistent with this idea. Why are only dimers considered? In addition, Ire1 oligomerization or dimerization status has been shown to switch the IRE1 RNase between HAC1 RNA splicing and regulated Ire1-dependent degradation (RIDD) of the ER-associated mRNA. The authors should measure the RIDD activity of the mutant Ire1.

Our approach focused predominantly on dimers (without ignoring the contribution and role of oligomers), because cysteine crosslinking can only occur between two molecules. Cysteine crosslinking has very successfully been used in the past to study the structural arrangements of transmembrane helices of receptors and other transmembrane proteins (as references in the manuscript). Our purpose was to study the transmembrane organization of Ire1.

The question if RIDD is a physiologically relevant mechanism in *S. cerevisiae* is a matter of active debate and remains unresolved. We added the following text plus references to the introduction: "A regulated IRE1-dependent decay of mRNA (RIDD) has been suggested as a parallel mechanism to reduce the folding load of the ER. However, RIDD does not seem to play the same important role in *Saccharomyces cerevisiae* as it does in *Saccharomyces pombe* or mammalian cells (Travers et al., 2000; Hollien and Weissman, 2006; Frost et al., 2012; Tam et al., 2014; Li et al., 2018)."

In response to the reviewer's suggestion, we also have studied the impact of ER-stress caused by DTT on four potential RIDD candidates: *FAL1*, *ICL1*, *MCR1*, *PMI40*. These candidates were selected based on transcriptome data from a previous study from the Walter laboratory (PMID: 10847680; Travers et al., 2000). If they were RIDD targets, their mRNA level should be downregulated upon ER stress in both WT cells and in cells lacking *HAC1*, but not in cells lacking *IRE1*. We analyzed the mRNA level of *FAL1*, *ICL1*, *MCR1*, and *PMI40* via qPCR in unstressed and acutely (1h) DTT-stressed cells. None of the four candidates showed the appropriate responses. Thus, we failed to confirm those candidate genes as firm targets of RIDD. Furthermore, recent work on bioarchives (<https://doi.org/10.1101/2021.02.10.430655>) suggests that Ire1

from *S. cerevisiae* is incapable of unspecific mRNA splicing (which is required for RIDD) unless mutations are being introduced to render it more similar to Ire1 from *S. pombe*.

Taken together: The physiological relevance of RIDD remains at least questionable in *S. cerevisiae*. While a possible role of RIDD cannot be formally excluded by our experiments, it is clear, that RIDD does not play the same important role in *S. cerevisiae* as it does in *S. pombe* or mammalian cells. We extended our discussion on a putative role on RIDD for the revised manuscript.

Moreover, Ire1 dimerization and oligomerization are known to be facilitated by the cytoplasmic portion of the Ire1 kinase and RNase domains, in addition to the Ire1 ER luminal and TM domains. Why were only the ER luminal domain interfaces, IF1 and 2, considered? Similar to the ER luminal domain, Ire1 interfaces through the cytoplasmic portion of Ire1.

We agree with the reviewer, that the cytosolic domains of Ire1 are likely to contribute to the overall stability of Ire1 clusters. For the purpose of our experiments, we only needed to destabilize dimers and higher oligomers of Ire1. We used the IF1 and IF2 mutation on the ER-luminal side to lower the degree of dimerization and back-to-back oligomerization, respectively. For our purposes, targeting the ER-luminal domain was sufficient.

(4) Subcellular fractionation: In Figure S3, differential centrifugation is used to fractionate cysteine-less Ire1 in order to confirm that it is similar to WT Ire1. It is unclear why significant levels of Ire1 TM protein are found in the low speed spin supernatant, suggesting that some Ire1 was solubilized. In contrast, re-probing with anti-Dpm1 antibody showed that the majority of Dpm1, another membrane protein, stayed in the low speed spin pellet fraction. Don't these results suggest that cysteine-less Ire1 acts differently than WT Ire1 in vivo?

We assume the reviewer refers to Figure S1. With all due respect, the conclusion that Ire1 was solubilized because it can be detected in the supernatant of a low speed spin is incorrect. ER-derived microsomes do not efficiently pellet upon low speed centrifugation. Some microsomes remain in the supernatant and can only be pelleted by a 100 000 x g step. Dpm1 (bottom panel) shows a similar distribution throughout the fraction as Ire1 (top panel).

Similarly, results in Figure S2C suggest that cysteine-less Ire1 is not solubilized even after TX-100 treatment, while essentially 100% of Dpm1 is in the solubilized fraction upon TX-100 treatment.

C Membrane extraction assay

We assume the reviewer refers to Figure S1C. The solubilization of Ire1 by TX-100 is indeed not complete (while it is complete for Dpm1). An incomplete solubilization of membrane proteins by a particular detergent is not uncommon. In fact, it is known that TX-100 does not solubilize Ire1 completely. It is therefore not unexpected to find some cysteine-less Ire1 remaining in the pellet fraction even after TX-100 treatment and high speed centrifugation (as seen in Figure. S1C). The same behavior has been observed for WT Ire1 by our lab (Halbleib et al., 2017) and, for a similar Ire1 WT construct, by the Kohno lab (Kimata et al. 2007, PMID: 17923530).

We conclude that cysteine-less Ire1 and WT Ire1 show very similar behavior in subcellular fractionations and membrane extraction assays.

Taken together, these results show that property of Ire1 is altered in cysteine-less Ire1, or unfolded/aggregated. On the other hand, it is unlikely that this is the simple unfolding of cysteine-less Ire1 since levels of PDI upon DTT or TM treatments increase significantly at an extent similar to WT Ire1. Since single cysteine Ire1 mutants were generated on the idea that cysteine-less Ire1 behaved similar to WT Ire1, it needs to be clarified.

We have no evidence that cysteine-less Ire1 is unfolded or aggregated. We hope that our answers have resolved any unclarities.

(5) Overall conclusions: The authors should discuss how their conclusions, specifically the significance of F544 within the TM domain of Ire1, fit in with previous results where a complete swap or scramble of Ire1 TM residues remains active. In this case, what happens to the proposed Ire1 TM conformation? Furthermore, in the discussion, the authors argue that "based on our crosslinking data, we suggest that the complex metabolic, transcriptional, and non-transcriptional adaptations to different forms of ER-stress do not reflect distinct functional modes of Ire1 (Fun and Thibault, 2020)." However, the current study does not address any contributions from the kinase or RNase domains of Ire1, which hold enzymatic activities of Ire1. Clearly, metabolic outcomes are largely impacted by the enzymatic activities of Ire1 and, thus, the authors should modify this statement.

Following the reviewer's advice, we have extended our discussion on the role of F544 (see new paragraphs in the results section and the discussion). We also have discussed our data considering the available data on 'swap' and 'scramble' experiments on the Ire1 TMH.

We now state:

"Similarly, when the AH-disrupting mutation F531R was combined with the F544C mutation (at the crossing-point of the X-shaped TMH-dimer), we observed only a very mild, yet significant functional defect (Suppl. Materials Fig. S5A) and a strongly reduced crosslinking propensity (Suppl. Materials Fig. S5B). This robust resistance to DTT is somewhat surprising considering the strongly reduced crosslinking propensity. However, the disruption of the AH changes the placement of the TMH in the membrane and the degree of membrane thinning and water penetration (Halbleib et al., 2017). We speculate that these combined changes would place the polar F544C residue more deeply in the hydrophobic core of the membrane, thereby affecting its propensity to undergo a Cu²⁺-catalyzed crosslinking, but at the same time favoring Ire1 dimerization. Notably, the F544C mutation alone does not lead to an increased UPR activity and ER stress resistance (Fig. 3C, D; Suppl. Materials Fig. S3A). In fact, the primary sequence of Ire1's TMH can be systematically mutated (Fig. 3C; Suppl. Materials Fig. S3A), scrambled (in the case of the mammalian IRE1 α) or exchanged altogether (Halbleib et al., 2017; Volmer et al., 2013) without causing a detectable functional defect. It therefore seems that a suitably placed polar residue in the TMH, here through the F544C mutation, becomes phenotypically relevant only when Ire1 is otherwise compromised. Beyond that, our data suggest that the overall architecture of the TMH region with an intact AH is relevant for normal UPR function."

Furthermore, we have toned down our conclusions on the complex signaling outcome(s) and have extended our discussion based on findings from new experiments (new Fig. 6D,E).

Finally, the authors state that "Based on a systematic cysteine crosslinking approach in native membranes and aided by MD simulations, we show that the neighboring TMHs in clusters of Ire1 are organized in an X-shaped configuration." Beyond the simulation and F554 TM residue involvement, additional configurations of the Ire1 TM domain may be possible.

We agree with the reviewer that the overall organization of the transmembrane helix region allows for some structural dynamics. Indeed, this is also what we observe in MD simulations. However, our cysteine crosslinking approach clearly favors a structural model of Ire1 with an overall X-shaped configuration (which is also stably maintained throughout a 1 μ s long, atomistic MD simulation).

Minor

points:

(1) The authors stated "Nevertheless, seemingly subtle changes in the TMH region, however, can modulate the cellular resistance to ER stress when being placed. In light of these findings, we are convinced that studying UPR transducers at their endogenous level is absolutely required to understand their role in physiology." This has been described in many previous studies. The authors should reference earlier publications.

We would be happy to refer to those publications. Unfortunately, we are not aware of any publication that studies mutant variants Ire1 at the endogenous level. Many elegant studies in the past have used *IRE1* expressed from the endogenous promoter on *CEN*-based plasmids. As explained in our answer to point 1 of reviewer 1, this already can lead to a considerable degree of overexpression. This is why we feel that our knock-in approach is an important step, and why we find our statement important.

(2) In Figures 6C-6D, the authors probed the relationships between the F554 residue within the TM domain and IF2 residues.

In the revised manuscript, we have extended our analysis of the F544C/IF2 double mutation (new Fig. 6D,E).

(3) In Figure 5, the authors stated that a mutation within the AH, F531R, causes unfolding of AH and distorts the X and the placement of F544C for efficient crosslinking. However, Figure 5D shows that this mutation caused only a slight sensitivity change (15% to ~8%).

It is important to realize that the crosslinking efficiency does not report on the sensitivity of Ire1 to ER-stress. The decrease in crosslinking efficiency from 15% to ~8% cannot be taken as an evidence for a reduced oligomerization of Ire1. In fact, many single cysteine variants generated in this study do not show any crosslinking, while still being able to cluster and to mount an active UPR (Figure 3 and Figure S3).

As explained in the revised manuscript, a functional AH becomes most relevant after prolonged ER stress caused by DTT or TM. Acute ER-stress caused by DTT or TM (1 hr for X-linking) activates Ire1 and can 'overrule a defective AH' leading to full activation of the UPR. As explained in the revised manuscript, the AH become physiologically much more relevant during prolonged period of ER stress. We have extended our discussion on the important implications from these findings.

(4) pH concerns: Additionally, throughout the manuscript the authors describe experiments performed both in YPD and SGD. The effect of DTT is altered depending upon the pH of the media. For example, YPD is normally pH 7 unless adjusted, while SGD is pH 4-5. Has the pH of the media been adjusted? If so, this should be mentioned.

Thank you for this important comment. The reviewer's statement helped us a lot with an ongoing project. As for our current study: We did not adjust the pH of the medium. However, we made sure to use consistent protocols and media for all our experiments to assure reproducibility. Following the reviewer's advice, we mention this point in the revised manuscript.

May 13, 2021

RE: JCB Manuscript #202011078R

Prof. Robert Ernst
Saarland University
PZMS and Medical Biochemistry and Molecular Biology
Kirrberger Str. 100 Geb. 61.4
Homburg 66421
Germany

Dear Prof. Ernst:

Thank you for submitting your revised manuscript entitled "Cysteine crosslinking in native membranes establishes the transmembrane architecture of Ire1". We would be happy to publish your paper in JCB pending final revisions necessary to meet our formatting guidelines (see details below). In your final revision, please be sure to address reviewer #2's final minor comments.

A. MANUSCRIPT ORGANIZATION AND FORMATTING:

Full guidelines are available on our Instructions for Authors page, <https://jcb.rupress.org/submission-guidelines#revised>. **Submission of a paper that does not conform to JCB guidelines will delay the acceptance of your manuscript.**

1) Text limits: Character count for Articles is < 40,000, not including spaces. Count includes title page, abstract, introduction, results, discussion, acknowledgments, and figure legends. Count does not include materials and methods, references, tables, or supplemental legends.

2) Figures limits: Articles may have up to 10 main text figures.

3) Figure formatting: Scale bars must be present on all microscopy images, including inset magnifications. Molecular weight or nucleic acid size markers must be included on all gel electrophoresis.

4) Statistical analysis: Error bars on graphic representations of numerical data must be clearly described in the figure legend. The number of independent data points (n) represented in a graph must be indicated in the legend. Statistical methods should be explained in full in the materials and methods. For figures presenting pooled data the statistical measure should be defined in the figure legends. Please also be sure to indicate the statistical tests used in each of your experiments (either in the figure legend itself or in a separate methods section) as well as the parameters of the test (for example, if you ran a t-test, please indicate if it was one- or two-sided, etc.). Also, if you used parametric tests, please indicate if the data distribution was tested for normality (and if so, how). If not, you must state something to the effect that "Data distribution was assumed to be normal but this was not formally tested."

- 5) Abstract and title: The abstract should be no longer than 160 words and should communicate the significance of the paper for a general audience. The title should be less than 100 characters including spaces. Make the title concise but accessible to a general readership.
- 6) Materials and methods: Should be comprehensive and not simply reference a previous publication for details on how an experiment was performed. Please provide full descriptions in the text for readers who may not have access to referenced manuscripts.
- 7) Please be sure to provide the sequences for all of your primers/oligos and RNAi constructs in the materials and methods. You must also indicate in the methods the source, species, and catalog numbers (where appropriate) for all of your antibodies. Please also indicate the acquisition and quantification methods for immunoblotting/western blots.
- 8) Microscope image acquisition: The following information must be provided about the acquisition and processing of images:
 - a. Make and model of microscope
 - b. Type, magnification, and numerical aperture of the objective lenses
 - c. Temperature
 - d. Imaging medium
 - e. Fluorochromes
 - f. Camera make and model
 - g. Acquisition software
 - h. Any software used for image processing subsequent to data acquisition. Please include details and types of operations involved (e.g., type of deconvolution, 3D reconstitutions, surface or volume rendering, gamma adjustments, etc.).
- 9) References: There is no limit to the number of references cited in a manuscript. References should be cited parenthetically in the text by author and year of publication. Abbreviate the names of journals according to PubMed.
- 10) Supplemental materials: There are strict limits on the allowable amount of supplemental data. Articles may have up to 5 supplemental display items (figures and tables). Please consider combining SI panels or moving data to the main figures to reduce the count and be sure to correct the callouts in the text to reflect this change. Please also note that tables, like figures, should be provided as individual, editable files. A summary of all supplemental material should appear at the end of the Materials and methods section.
- 11) eTOC summary: A ~40-50-word summary that describes the context and significance of the findings for a general readership should be included on the title page. The statement should be written in the present tense and refer to the work in the third person.
- 12) Conflict of interest statement: JCB requires inclusion of a statement in the acknowledgements regarding competing financial interests. If no competing financial interests exist, please include the following statement: "The authors declare no competing financial interests." If competing interests are declared, please follow your statement of these competing interests with the following statement: "The authors declare no further competing financial interests."
- 13) ORCID IDs: ORCID IDs are unique identifiers allowing researchers to create a record of their various scholarly contributions in a single place. At resubmission of your final files, please consider

providing an ORCID ID for as many contributing authors as possible.

B. FINAL FILES:

Thank you for this interesting contribution, we look forward to publishing your paper in Journal of Cell Biology.

Sincerely,

Jodi Nunnari, Ph.D.
Editor-in-Chief

Andrea L. Marat, Ph.D.
Senior Scientific Editor

Journal of Cell Biology

Reviewer #1 (Comments to the Authors (Required)):

The authors have addressed most critical concerns of the reviewers. The quality of the manuscript is significantly improved. I have no more major concerns.

Reviewer #2 (Comments to the Authors (Required)):

We were enthusiastic about this paper at the start of the review process, and our concerns were only minor in nature. The authors thoroughly addressed all our comments and suggestions in the revised manuscript. Furthermore, after reading the authors' responses to the other reviewers, we feel that their concerns are also adequately addressed and that the revised manuscript has further improved in quality.

This remains a very strong paper that would be of great interest to the general JCB readership.

No further significant revisions are required, and the manuscript should be published in its current form.

There is a minor error in the revised version that should be corrected: Figure 5 panels are mislabeled; panels C and D are missing.